



# Bridging the gaps between particulate backscattering measurements and modeled particulate organic carbon in the ocean

**Martí Galí[1], Marcus Falls[1], Hervé Claustre[2], Olivier Aumont[3], and Raffaele Bernardello[1]**

[1]Barcelona Supercomputing Center (BSC), Plaça d'Eusebi Güell, 1–3, 08034 Barcelona, Catalonia, Spain
[2]CNRS & Sorbonne Université, Laboratoire d'Océanographie de Villefranche (LOV), 06230 Villefranche-sur-Mer, France
[3]IPSL-LOCEAN, IRD/Sorbonne Université/CNRS/MNHN, Paris, France

**Correspondence:** Martí Galí (marti.gali.tapias@gmail.com)

**Abstract.** Oceanic particulate organic carbon (POC) is a small but dynamic component of the global carbon cycle. Biogeochemical models historically focused on reproducing the sinking flux of POC driven by large fast-sinking particles (LPOC). However, suspended and slow-sinking particles (SPOC, here $< 100\,\mu m$) dominate the total POC (TPOC) stock, support a large fraction of microbial respiration, and can make sizable contributions to vertical fluxes.

Recent developments in the parameterization of POC reactivity in the PISCES (Pelagic Interactions Scheme for Carbon and Ecosystem Studies model; PISCESv2_RC) have improved its ability to capture POC dynamics. Here we evaluated this model by matching a global 3D simulation and 1D simulations at 50 different locations with observations made from biogeochemical (BGC-) Argo floats and satellites. Our evaluation covers globally representative biomes between 0 and 1000 m depth and relies on (1) a refined scheme for converting particulate backscattering at 700 nm ($b_{bp700}$) to POC, based on biome-dependent POC / $b_{bp700}$ ratios in the surface layer that decrease to an asymptotic value at depth; (2) a novel approach for matching annual time series of BGC-Argo vertical profiles to PISCES 1D simulations forced by pre-computed vertical mixing fields; and (3) a critical evaluation of the correspondence between in situ measurements of POC fractions, PISCES model tracers, and SPOC and LPOC estimated from high vertical resolution $b_{bp700}$ profiles through a separation of the baseline and spike signals.

We show that PISCES captures the major features of SPOC and LPOC across a range of spatiotemporal scales, from highly resolved profile time series to biome-aggregated climatological profiles. Model–observation agreement is usually better in the epipelagic (0–200 m) than in the mesopelagic (200–1000 m), with SPOC showing overall higher spatiotemporal correlation and smaller deviation (typically within a factor of 1.5). Still, annual mean LPOC stocks estimated from PISCES and BGC-Argo are highly correlated across biomes, especially in the epipelagic ($r = 0.78$; $n = 50$). Estimates of the SPOC / TPOC fraction converge around a median of 85 % (range 66 %–92 %) globally. Distinct patterns of model–observations misfits are found in subpolar and subtropical gyres, pointing to the need to better resolve the interplay between sinking, remineralization, and SPOC–LPOC interconversion in PISCES. Our analysis also indicates that a widely used satellite algorithm overestimates POC severalfold at high latitudes during the winter. The approaches proposed here can help constrain the stocks, and ultimately budgets, of oceanic POC.

## 1 Introduction

The biological carbon pump (BCP) is the ensemble of processes that transfer the organic matter produced by plankton in the sunlit ocean surface to deeper layers (Volk and Hoffert, 1985). This vertical flux plays a central role in the Earth's climate, as it influences the oceans' capacity to absorb and ultimately store atmospheric $CO_2$ over centennial or millennial timescales (Kwon et al., 2009; Passow and Carlson, 2012). The BCP is also central to biogeochemical functioning of the ocean, as it determines the quality and quantity of organic matter available to the ocean interior (Arístegui et al., 2009; Hernández-León et al., 2020) and the seafloor

ecosystems. The spatiotemporal patterns of organic matter supply and respiration influence the distribution of dissolved oxygen, inorganic carbon, remineralized nutrients and trace metals in the ocean interior (Bianchi et al., 2018; Duteil et al., 2012; Hayes et al., 2015; Oschlies et al., 2018; Weber et al., 2016) and their return pathways to the surface. In consequence, the BCP is intimately linked to, and feeds back on, upper-ocean productivity.

Over the last decades, BCP research has placed emphasis on understanding the ecological and physical factors that control the gravitational export of particulate organic carbon (POC; see Table 1 for abbreviation definitions). This process is often represented as the product of export production (the fraction of net primary production exported below the euphotic layer) and transfer efficiency (Teff; the fraction of export production that reaches a given depth below the euphotic layer). Both variables vary widely across ocean biomes and along the seasonal cycle (Buesseler and Boyd, 2009; Passow and Carlson, 2012; Buesseler et al., 2020), and our understanding of the underlying mechanisms is still limited by the relatively small amount of in situ measurements (Mouw et al., 2016). Biogeochemical models have been built, and their parameters tuned, to be able to reproduce these sparse observations of export production and vertical flux attenuation, sometimes reaching apparently contradictory results (Marsay et al., 2015). In comparison, the models' ability to represent marine particle concentrations has received less attention (Lam et al., 2011).

Marine particles are mainly composed of living microbial plankton cells, living metazoans, and detritus ("marine snow" aggregates, fecal pellets, zooplankton feeding structures, biominerals), whose size ranges from $< 1\,\mu m$ to several millimeters (Bishop et al., 1980; Mullin et al., 1965; Stemmann and Boss, 2012). Indeed, these particles feature wide variations in their physicochemical properties and degree of biological processing (Kharbush et al., 2020; Lam et al., 2015; Passow and Carlson, 2012; Stemmann and Boss, 2012), as well as microbial colonization (Baumas et al., 2021; Duret et al., 2019; Mestre et al., 2018), all of which change during the particles' lifetime. The gravitational sinking speed generally increases with particle size, although observations show a wide scatter around canonical Stokes' law predictions (Cael et al., 2021; Laurenceau-Cornec et al., 2019). Owing to this general relationship, particle populations are often partitioned into a few functional size classes: large particles, typically defined as larger than 50 or $100\,\mu m$, which usually sink at several tens or hundreds of meters per day, and small particles, which usually sink slowly ($< 10\,m\,d^{-1}$) or are suspended in the water column. In this study, POC is divided into small POC (SPOC) and large POC (LPOC), with a nominal cutoff at $100\,\mu m$ (Table 2; Sect. 2.2.2).

The traditional BCP paradigm posits that gravitational sinking of LPOC controls the vertical carbon flux (Sarmiento and Gruber, 2006). However, it has been known for decades

that POC export is shaped by additional processes, such as the physical transport of particles by convection and subduction and the "active" particle flux mediated by the vertical migration of metazoans. Recently, these processes have been collectively termed "particle injection pumps" (Boyd et al., 2019). In parallel, the role of dissolved organic carbon in vertical carbon export has been widely recognized (Jiao et al., 2010; Passow and Carlson, 2012; Legendre et al., 2015). Therefore, the BCP is increasingly seen as a diverse array of interconnected mechanisms.

One aspect that has recently received considerable attention is the role of suspended and slow-sinking particles (Alonso-González et al., 2010; Baker et al., 2017). Owing to their longer residence time, small particles usually dominate the POC stock (Aumont et al., 2017; Baker et al., 2017) and may support a proportional fraction of the respiration (Baltar et al., 2010a, b; Belcher et al., 2016; García-Martín et al., 2021). Convective mixing (Bishop et al., 1986; Dall'Olmo and Mork, 2014; Lacour et al., 2019) and subduction (Llort et al., 2018; Omand et al., 2015; Resplandy et al., 2019) can transport SPOC into the mesopelagic layer, adding to other export mechanisms and potentially making large contributions to total POC export (Alonso-Gonzalez et al., 2010; Henson et al., 2015). Production of SPOC in the mesopelagic and below also results from the fragmentation of LPOC, caused by physical disaggregation (Takeuchi et al., 2019), bacterial solubilization, and zooplankton activity (Briggs et al., 2020; Goldthwait et al., 2004; Mayor et al., 2020; Stemmann et al., 2004b). Moreover, SPOC is also produced through bacterial chemosynthesis in the dark ocean (Arístegui et al., 2009; Herndl and Reinthaler, 2013). Altogether, these findings illustrate how our limited knowledge of POC characteristics and cycling hampers a mechanistic understanding of the BCP and mesopelagic carbon budgets (Giering et al., 2014).

Biogeochemical models designed to capture only gravitational POC sinking fail to represent POC stocks in the ocean interior. Aumont et al. (2017) recently showed that the Pelagic Interactions Scheme for Carbon and Ecosystem Studies model (PISCESv2; Aumont et al., 2015) underestimated POC by 1 order of magnitude or more below the epipelagic layer. This pitfall is likely common to any state-of-the-art model with a similar structure (Laufkötter et al., 2016; Séférian et al., 2020). Aumont's work also showed that the model's fit to observed deep-ocean POC concentrations could be dramatically improved by treating detrital POC, both small and large, as a mixture of particles with different reactivity (or lability) towards bacterial degradation. In this scheme, termed the reactivity continuum (RC) parameterization, detrital POC degradation is computed after dividing it into many reactivity classes that approximately follow a continuous gamma distribution – hence its name. The most labile fractions are rapidly consumed below the upper mixed layer, such that vertically exported POC becomes progressively more refractory. This results in enhanced preservation

**Table 1.** List of abbreviations.

| Type | Abbreviation | Definition |
|---|---|---|
| Region name | NASPG | North Atlantic subpolar gyre |
| | STG | Subtropical gyres of the Atlantic and Pacific oceans |
| Process or concept | BCP | Biological carbon pump |
| | CATS | Coherent annual time series |
| | MLD | Mixed layer depth |
| | Teff | Vertical transfer efficiency of POC |
| Operationally defined chemical compartment[a] | POC | Particulate organic carbon (used here as a generic name) |
| | SPOC | Small particulate organic carbon |
| | LPOC | Large particulate organic carbon |
| | TPOC | Total particulate organic carbon (here equivalent to SPOC + LPOC) |
| Bio-optical variable | $b_{bp700}$ | Particulate backscattering coefficient at 700 nm wavelength |
| | Chl $a$ | Chlorophyll $a$ concentration estimated from fluorescence[b] |
| Numerical model or parameterization | NEMO | Nucleus for European Modelling of the Ocean |
| | PISCES | Pelagic Interactions Scheme for Carbon and Ecosystem Studies |
| | RC | Reactivity continuum parameterization for POC degradation |
| Project name | NAOS | Novel Argo Ocean Observing System |
| | remOCEAN | Remotely-Sensed Biogeochemical Cycles in the Ocean |

[a] These variables may be estimated directly from seawater sampling, indirectly from in situ or remote sensors, and from biogeochemical models. See Table 2. [b] See text Sect. 2.2 for details.

of SPOC in the model and a much more realistic fraction of SPOC with respect to total POC (TPOC) in the ocean interior. In addition, the RC scheme does not appreciably degrade model estimates of the gravitational POC flux.

Despite this breakthrough in the representation of POC fractions in PISCESv2_RC (hereafter "PISCES"), the new parameterization was evaluated using only sparse measurements (Druffel et al., 1992; Lam et al., 2011, 2015) based on large-volume filtration with in situ pumps. This approach enables an accurate determination of the mass and composition of the particulate fraction but cannot afford high-frequency sampling over extended spatiotemporal scales (Bishop, 1999; Boss et al., 2015; Gardner et al., 2006). During the last decade, the launching of the biogeochemical Argo (BGC-Argo) program of robotic observations has ended the historical undersampling of particles in the ocean interior (Claustre et al., 2020). BGC-Argo floats provide vertical profiles of temperature, salinity, bio-optical, and chemical variables between 0–1000 m every 1 to 10 d in near-real time and are thus well-suited to study particles ($\sim 0.5\,\mu$m to $\sim 2$ mm in size; Table 2) in the mesopelagic layer, where the strongest POC gradient occurs. The rapidly growing fleet of BGC-Argo floats equipped with bio-optical sensors enables a comparison between models and observations at global scales with enhanced spatiotemporal resolution. Unfortunately, BGC-Argo floats measure only a bio-optical proxy of POC, the particulate backscattering coefficient (usually at 700 nm, $b_{bp700}$) and empirical conversion factors are needed to estimate POC (Bishop and Wood, 2008; Cetinić et al., 2012; Stramski et al.,

2008). These conversion factors vary in response to several concurrent processes that alter particle abundance, size distribution, shape, composition, and ultimately optical properties (Boss et al., 2015; Giering et al., 2020).

In this study we compare SPOC and LPOC concentrations estimated from BGC-Argo floats to their PISCES-simulated counterparts, as well as satellite-retrieved surface POC concentration. The comparison is enabled by a novel empirical algorithm to convert $b_{bp700}$ to POC. Observations and simulations are matched in 3D (biome-wide climatological scale) and 1D (at defined locations over an annual cycle). These complementary strategies allow us to evaluate the skill of PISCES at simulating POC stocks and fractions in globally representative biomes. We conclude with a list of recommendations to fully exploit the potential of robotic particle observations combined with biogeochemical modeling.

## 2 Methods

### 2.1 Definition of vertical and horizontal domains

Studies of the BCP usually decompose the ocean into vertical domains: a surface layer where autotrophic activities dominate and one or several ocean interior layers where heterotrophic processes dominate. Functional definitions based on light penetration, peak export production, vertical mixing, or long-term carbon sequestration are usually the most appropriate ones for process studies (Buesseler and Boyd, 2009; Buesseler et al., 2020; Guidi et al., 2015;

**Table 2.** Match between BGC-Argo observations, PISCES tracers (Aumont et al., 2017), real-world particulate organic carbon pools, and particle size ranges[a].

| POC fraction | BGC-Argo observation | PISCES tracer (carbon currency) | Closest real-world correspondence |
|---|---|---|---|
| SPOC | $b_{bp700}$ vertical profiles, despiked signal. Most sensitive to 0.5–30 µm particles (Dall'Olmo et al., 2009; Organelli et al., 2018). Calibrated as POC with Eqs. (1) and (2). | *PHY*: "nanophytoplankton", includes calcifiers (default fraction 30 %). Nominally 1–10 µm. | Picocyanobacteria and non-diatom phyto-eukaryotes, generally 0.5–20 µm. |
| | | *PHY2*: silicifying microphytoplankton. Nominally 10–50 µm. | Diatoms, between 1.5 µm cells (Vaulot et al., 2008) and millimeter-scale chains. Mean ESD[b] generally < 50 µm in seawater (Snoejis et al., 2002; Ciotti et al., 2002; Bricaud et al., 2004). |
| | | *POC*: detrital particulate organic carbon. Nominally < 100 µm. In practice, it includes heterotrophic prokaryotes' biomass with current parameter values (see Sect. 4.3). | Detrital particles between ∼ 0.2[c] and 100 µm. In practice, measurements may include particle-attached and free-living organisms, viruses, colloids and adsorbed DOC. |
| | | *ZOO*: microzooplankton. Nominally 10–200 µm. | Microzooplankton. Mostly ciliates and flagellates with size similar to their prey, down to around 2 µm (Calbet, 2008). |
| | | Heterotrophic prokaryotes (*BACT*): currently not a prognostic tracer in PISCES. Not considered explicitly in this study (see Sect. 4.3). | Free-living heterotrophic prokaryotes (bacteria and archaea), < 1 µm. |
| LPOC | $b_{bp700}$ vertical profiles, spike signal. Particle size between ∼ 100 µm ($b_{bp700}$ spike of $2.3 \times 10^{-5}$ m$^{-1}$) and ∼ 2 mm ($b_{bp700}$ spike of $8 \times 10^{-3}$ m$^{-1}$) (Briggs et al., 2020). Calibrated as POC with Eqs. (1) and (2). | *GOC*: detrital particulate organic carbon. Nominally > 100 µm. | Detrital particles > 100 µm (aggregates, fecal pellets). Includes attached microbes. |
| | | *ZOO2*: mesozooplankton. Includes flux feeders[d]. Nominally 0.2–2 mm. | Mesozooplankton[d] |

[a] See Stemmann and Boss (2012) for typical seawater particle size spectra. [b] Equivalent spherical diameter. [c] Particles in the 0.2–0.8 µm size range and DOC are retained with variable efficiency by the filters commonly used to determine POC (Bishop, 1999; Cetinić et al., 2012; Graff et al., 2015; Lee et al., 1995; Morán et al., 1999; Strubinger Sandoval et al., 2021). [d] PISCES mesozooplankton represents mostly copepods in the euphotic layer and flux feeders below it. The fraction of flux feeders is diagnosed in PISCES from the proportion between flux feeding rates and total mesozooplankton ingestion rates. By construction, flux feeding becomes the dominant mode of mesozooplankton feeding below the euphotic layer under productive surface waters in PISCES. In reality, a wide variety of feeding strategies and organisms are found in the mesopelagic (Ikenoue et al., 2019; Kiørboe, 2011; Mayor et al., 2020; Stukel et al., 2019).

Palevsky and Doney, 2018). Because this paper is mainly descriptive and combines observations and simulations, we will refer to layers defined by fixed depths: epipelagic (0–200 m), mesopelagic (200–1000 m), and bathypelagic (1000–4000 m).

Over the horizontal dimensions, our comparisons between observations and model results rely on the ocean biomes defined by Fay and McKinley (2014). These authors subdivided each ocean basin (Atlantic, Pacific, Indian, and Southern Ocean) into different biomes based on observed variables, namely sea-surface temperature, spring–summer chlorophyll *a* concentration (Chl *a*), ice fraction, and maximum mixed layer depth (MLD), all on a 1° × 1° grid. This division resulted in 17 regions ascribed to one of the following five biomes: the ice biome, the subpolar seasonally stratified biome, the subtropical seasonally stratified biome, the subtropical permanently stratified biome, and the equatorial biome. The analyses reported herein focus on the following

four biomes (Fig. 1): the seasonally stratified North Atlantic subpolar gyre (NASPG); the permanently stratified Atlantic and Pacific subtropical gyres, which were grouped together (STG); the seasonally stratified Southern Ocean (subantarctic CE1); and the Mediterranean Sea, which was added here owing to the abundance of BGC-Argo data and represents a seasonally stratified subtropical biome. Fay and McKinley's definition allows biome boundaries to change from one year to another. Here we analyzed only data from the core of each biome, defined as the grid cells that never changed classification during the 1998–2010 satellite observation periods.

## 2.2 BGC-Argo observations

The global dataset acquired by the array of BGC-Argo floats was downloaded from the Global Data Assembly Center hosted by Ifremer (ftp://ftp.ifremer.fr/ifremer/argo/dac/, last access: 14 January 2020) (Argo, 2000). The selected floats were equipped with a Seabird-Wetlabs ECO-Triplet sensor package including a Chl $a$ fluorometer (excitation at 470 nm; emission at 695 nm) and a backscattering sensor at 700 nm ($b_{bp700}$), in addition to the conductivity–temperature–depth (CTD) probe. The downloaded measurements had undergone the standard processing, which includes the application of calibration equations to raw sensor output and the performance of near-real-time quality control to both CTD (Wong et al., 2021) and Chl $a$ measurements (Schmechtig et al., 2018). Since no specific quality control procedure has been established yet for $b_{bp700}$ profiles, the latter were only quality-controlled according to the general criteria (Schmechtig et al., 2016). Thus, we used all $b_{bp700}$ measurements with quality control flag $\leq 3$ (equivalent results were obtained with flag $\leq 2$). Two different processing pipelines were applied to different subsets of the BGC-Argo data, as described below.

### 2.2.1 Global gridded climatologies (3D approach)

The global dataset acquired between 2010 and 2019 was used to produce global gridded monthly and seasonal climatologies for $b_{bp700}$ and Chl $a$. The measurements were binned onto the ORCA2_L31 grid used for NEMO–PISCES simulations (see Sect. 2.4.1), which has a horizontal resolution of about 2° that increases to 0.5° in the meridional direction in the equatorial domain, and 30 oceanic vertical levels between the surface and the ocean bottom. The thickness of the vertical bins increases progressively from 10 m at the surface to 339 m in the 22nd bin (870–1209 m), the deepest one containing BGC-Argo data. In each grid element, the average, median, range, and data counts were computed. Profiles from the CSIRO and INCOIS data assembly centers were not used because, at the time of download, they had not taken into consideration the new calibration files provided by the manufacturer. A total of 72 460 profiles were used to calculate the global gridded climatologies.

### 2.2.2 Profile time series for individual floats sampling at higher resolution (1D approach)

A subset of the floats, deployed mostly by the projects NAOS, remOCEAN, and Bio-Argo France (model NKE PROVOR CTS-4), were programmed to sample at higher temporal and vertical resolution than the Argo defaults (10 d and 10 m). These floats made vertical profiles between 1000 m and the surface every 2, 5, or 10 d with a vertical resolution of 10 m between 1000 and 250 m (or 350), 1 m between 250 (or 350) and 10 m, and 0.2 m between 10 m and the sea surface. We processed this dataset with a dedicated pipeline to extract additional information on POC size fractions and their dynamics. Along each vertical profile we computed depth, conservative temperature, absolute salinity, $\sigma_\theta$, and spiciness (Flament, 2002) from the calibrated pressure, temperature, and salinity using the R package *oce* (Kelley, 2011). The MLD was calculated as the shallowest depth where $\sigma_\theta$ exceeded the surface reference value by 0.03 kg m$^{-3}$ (Bishop and Wood, 2009; Sallée et al., 2021). The surface reference corresponded to the $\sigma_\theta$ at 5 m after applying a five-point running mean to the top 10 m of the profile. The 0.03 kg m$^{-3}$ criterion provided sensible results across biomes and was consistent with the NEMO-simulated turbocline depth (see Sect. 2.5.2). Eleven additional MLD criteria were also calculated to assess the robustness of the approach (Fig. S1).

Following Briggs et al. (2011, 2020), each $b_{bp700}$ vertical profile was smoothed with sequential 11-point running-minimum and running-maximum filters to separate the baseline from the spikes. The baseline signal corresponds to the bulk population of small particles, whose diameter is smaller than 100 μm and mostly between 0.5 and 30 μm (Dall'Olmo et al., 2009; Organelli et al., 2018). Each spike reflects the passage of a particle larger than about 100 μm in front of the sensor window. Previous studies inferred that backscattering spikes are caused mostly by phytodetrital aggregates but also by large zooplankton and phytoplankton (Bishop and Wood, 2008; Briggs et al., 2011; Gardner et al., 2000). Assuming that backscattering sensors sample a volume of 10 mL (Briggs et al., 2020), we estimated that backscattering spike concentration was typically between a few and $< 100$ L$^{-1}$, consistent with previous independent estimates (McDonnell and Buesseler, 2010; Stemmann et al., 2008; Stemmann and Boss, 2012). Backscattering spikes were on average 4–10 times more abundant than chlorophyll fluorescence spikes. The $b_{bp700}$ spikes larger than 0.008 m$^{-1}$, associated with particles larger than $\sim 2$ mm, were removed, with a negligible impact on the total spike signal (Briggs et al., 2020). Unlike Briggs et al. (2020), we did not subtract from the baseline profile the 850–900 m signal, which in that study was attributed to a background of small refractory particles with constant concentration. The baseline and spike signals were converted to SPOC and LPOC, respectively, as described in the next section.

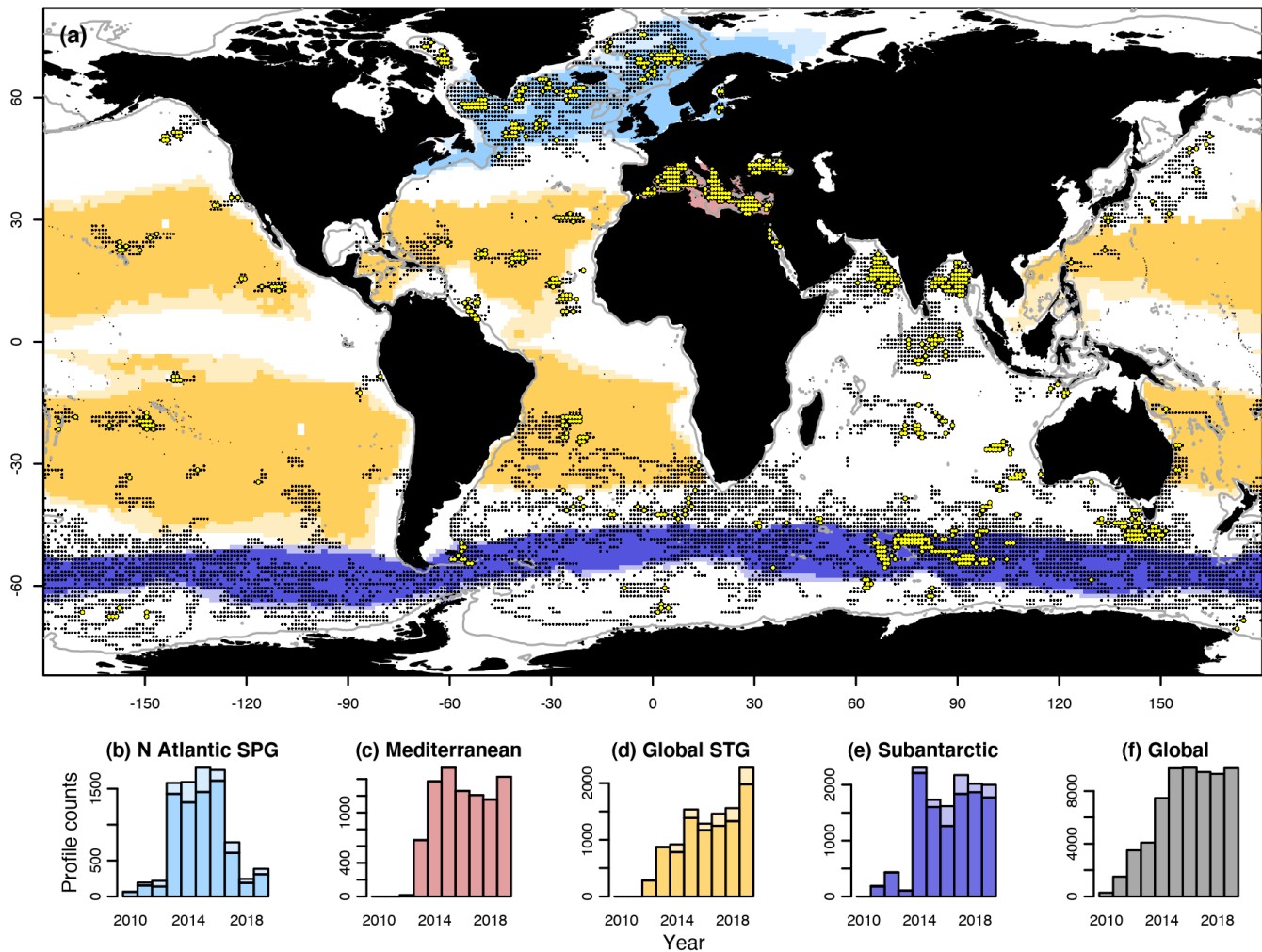

**Figure 1.** Global distribution and abundance of BGC-Argo profiles between 2010 and 2019. Grid cells ($1° \times 1°$) with at least one profile of the backscattering coefficient at 700 nm ($b_{bp700}$) are marked with black dots, and those with at least 20 profiles are marked with yellow-filled circles. The gray contours indicate the 1000 m isobath. Color shading indicates ocean biomes (see text), whose names are indicated on top of the bottom histograms. For each biome, light color indicates its average extent over 13 years of satellite observations (1998–2010), whereas the darker color indicates the "core" grid cells that never changed biome classification during the same period. The bottom panels **(b–f)** show the number of BGC-Argo $b_{bp700}$ CE2 profiles per year in the four selected biomes and in the global ocean.

All measurements were subsequently averaged into 18 vertical bins of progressively increasing thickness, such that the deepest bins contained at least 10 measurements. Finally, each profile was interpolated onto the L75 vertical grid commonly used in NEMO simulations. This grid has 46 bins between the surface (0–1 m) and the deepest layer considered here (901–996 m). The profile time series was temporally binned into 5 d periods.

For the comparison to PISCES 1D simulations, BGC-Argo time series were cut into 1-year periods (shifted by 6 months in the Southern Hemisphere), which we will call coherent annual time series (CATS) hereafter. The CATS fulfilled the following conditions: (1) sampling dates spanned at least between days of year 25 and 340; (2) the float remained in the

same region and did not cross major oceanic fronts according to the vertical–temporal evolution of temperature, salinity, $\sigma_\theta$, and spiciness; (3) bottom depth was > 1000 m for all profiles (bathymetry obtained from the 15 arcsec GEBCO 2019 product; https://www.gebco.net/data_and_products/gridded_bathymetry_data/gebco_2019/gebco_2019_info.html, last access: 12 May 2019); and (4) the Chl $a$ and $b_{bp700}$ sensors were stable according to both the vertical profiles and the continuous measurements acquired during drift at 1000 m between profiles. A total of 50 CATS from 28 different floats were selected, with 10–16 CATS in each biome and 32 (18) in the Northern (Southern) Hemisphere (Table S1).

## 2.3 Conversion of $b_{bp700}$ to POC

To convert the profiles of the backscattering coefficient at 700 nm ($b_{bp700}$) to POC we developed an empirical algorithm building on previous studies (Bol et al., 2018; Evers-King et al., 2017). The behavior of this algorithm is summarized in Fig. 2 and discussed in Sect. 4.2. Further details are provided in the Appendix. The algorithm estimates the POC / $b_{bp700}$ ratio along the vertical profile between 0 and 1000 m and proceeds in two steps. First, the POC / $b_{bp700}$ ratio is calculated by prescribing a POC / $b_{bp700}$ ratio in the surface layer ($z_{surf,biome}$) and an exponential decrease with depth in the underlying water column, which converges asymptotically towards a constant deep value ($c$):

$$\frac{POC}{b_{bp700}}(z) = c + a_{biome} \cdot e^{-0.001 \cdot b \cdot (z - z_{surf,biome})},$$

$$z > z_{surf,biome}. \qquad (1)$$

The $a_{biome}$ coefficient is biome-specific, whereas the asymptote at depth is fixed at $c = 1000 \,\mathrm{mmol\,C\,m^{-3}}$ m. The $z_{surf,biome}$ corresponds to the 5 % quantile of the climatological MLD in summer in a given biome (here ranging between 14 m in the Mediterranean and 41 m in the subantarctic). The POC / $b_{bp,700}$ ratios at $z_{surf,biome}$, corresponding to $a_{biome} + c$ in Eq. (1), are taken from the literature and range between 2600 and 4900 $\mathrm{mmol\,C\,m^{-3}}$ m (Fig. 2; Table A1). Second, the POC / $b_{bp,700}$ profile derived from Eq. (1) is modified by extrapolating a constant POC / $b_{bp,700}$ value, taken from a reference depth, $z_{ref}$, to the sea surface. In each vertical profile, $z_{ref}$ is defined as the deepest of $z_{surf,biome}$ and the MLD:

$$\frac{POC}{b_{bp700}}(z) = \frac{POC}{b_{bp700}}(z_{ref}), \; z \leq z_{ref};$$

$$z_{ref} = \max\left(z_{surf,biome}, \text{MLD}\right). \qquad (2)$$

The exponential decrease prescribed by Eq. (1) is similar to that proposed by Bol et al. (2018), except for the inclusion of the constant term $c$ that prevents the ratio from becoming 0 at depth. The slope of the exponential decrease ($b = -6.57$) is constant in all biomes and based on our fit to the Cetinić et al. (2012) dataset, using the same depth bins as Bol et al. (2018) but additionally forcing the curve towards $c$ at 1000 m.

The uncertainty of regional $b_{bp700}$–POC conversion factors in the epipelagic is typically $< 10\%$ according to the standard error of the POC vs. $b_{bp700}$ linear regression slopes (Table A1). The few available measurements in the mesopelagic suggest a POC / $b_{bp700}$ uncertainty lower than a factor of 2. Through this study, we will assume that model / observation ratios larger (smaller) than 2 (0.5) can safely be regarded as model overestimates (underestimates), which possibly is a conservative criterion for the epipelagic layer.

The conversion of $b_{bp700}$ to POC was done using different MLD data for the global climatologies and the CATS. For

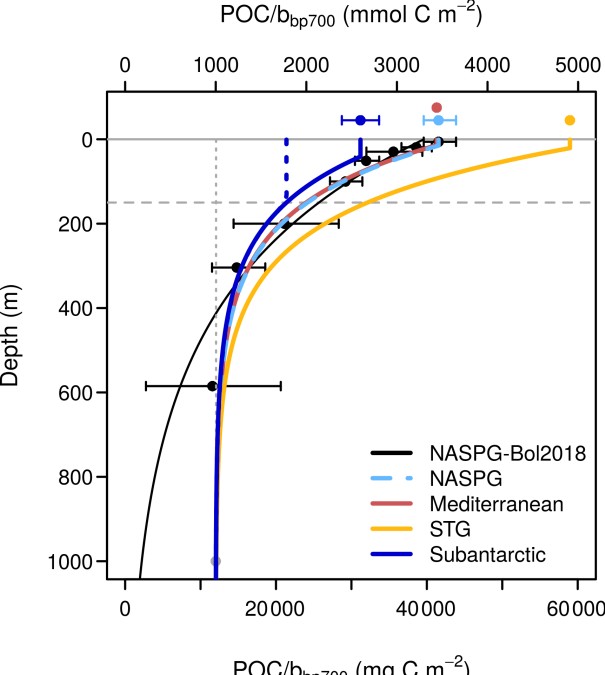

**Figure 2.** Empirical model used to convert the backscattering coefficient at 700 nm ($b_{bp700}$) to particulate organic carbon (POC). Black dots and error bars show the dataset of Cetinić et al. (2012) as binned by Bol et al. (2018). The black curve shows the exponential fit of Bol et al. (2018) to the Cetinić dataset. The dashed light-blue curve shows our fit to the same dataset (NASPG: North Atlantic subpolar gyre), forced to converge to a nonzero minimum value at depth. The remaining curves show similar functions with the same exponential slope ($b = -6.57$) as the NASPG fit, but with different surface values derived from the following studies: Loisel et al. (2001) for the Mediterranean Sea; Stramski et al. (2008) for the ensemble of subtropical and tropical areas excluding equatorial upwellings (STG); and Johnson et al. (2017) for the Southern Ocean (subantarctic). The depth of the homogeneous surface layer, $z_{surf,biome}$ in Eqs. (1) and (2), corresponds to the 5 % quantile of the climatological MLD in summer in a given biome: 15 m in the NASPG, 14 m in the MED, 21 m in the STG, and 41 m in the SO. The dotted green line, shown for the SO case only, illustrates the behavior of the algorithm for a hypothetical MLD of 150 m.

the global climatologies we used the Monthly Isopycnal and Mixed-layer Ocean Climatology (MIMOC) of Schmidtko et al. (2013), downloaded from https://www.pmel.noaa.gov/mimoc/, last access: 24 November 2020, which was reprojected onto the ORCA2 horizontal grid. Although MIMOC is based on an algorithm that evaluates several MLD criteria, it has been shown to be globally consistent with the MLD based on a 0.03 $\mathrm{kg\,m^{-3}}\sigma_\theta$ threshold (Holte and Talley, 2009; Sallée et al., 2021). For the float time series, we used the MLD defined by a 0.03 $\mathrm{kg\,m^{-3}}$ threshold computed for each individual profile.

## 2.4 Ocean color satellite data

Satellite observations for the 1997–2019 period were downloaded from GlobColour (https://www.globcolour.info, last access: 2 March 2020), a merged multi-sensor dataset. Monthly sea-surface POC fields based on the algorithm of Stramski et al. (2008) were used to compute monthly climatologies that were subsequently reprojected onto the ORCA2 grid.

## 2.5 PISCES simulations and matching with observations

Simulations were run using the ocean biogeochemistry model PISCESv2 (Aumont et al., 2015) with the RC parameterization for detrital POC (Aumont et al., 2017). The configuration of PISCES used here has 24 tracers: two classes of phytoplankton ("nanophytoplankton" and diatoms), detrital particles (small and big), and zooplankton (micro- and mesozooplankton), plus 18 additional tracers that comprise dissolved inorganic macronutrients and iron, inorganic carbon chemistry variables, dissolved organic carbon (DOC), and different particulate stocks of iron and silica. Phytoplankton growth depends on light, inorganic nitrogen, phosphorus, and iron, with an additional silicate requirement for diatoms. Microzooplankton and mesozooplankton consume the two classes of phytoplankton and small detrital particles with different preferences, and mesozooplankton also predate on microzooplankton. Detrital particles are produced through the mortality of phytoplankton and zooplankton (which are routed to small and large particles in different proportions), zooplankton sloppy feeding, and DOC coagulation. Production of large detritus also results from enhanced diatom mortality upon bloom collapse, aggregation of small detritus, and zooplankton mortality and fecal pellet production (the latter two derived from a closure term that accounts for unresolved higher trophic levels). Small and large detrital particles are nominally smaller/larger than 100 μm and sink, respectively, at 2 and 50 m d$^{-1}$. Both small and large detritus are remineralized by implicit bacterial activity and consumed by flux feeding mesozooplankton. Remineralization follows first-order kinetics with an initial specific rate "$k$" of 0.035 d$^{-1}$ (at 0 °C) for freshly produced detritus in the upper mixed layer. This $k$ decreases with depth as an emergent result of the RC parameterization. To account for bacterial solubilization of aggregate-binding polymers, 10 % of the degraded large detritus is routed to small detritus (this fraction is hard-coded based on previous calculations). The flux feeding rate depends on the particles' sinking flux and thus attenuates the flux of large particles more strongly than that of small particles. Additionally, a variable fraction of the large detritus intercepted by flux feeders is fragmented into small detritus, attenuating up to around 50 % of the large detritus sinking flux through the top 500 m during intense export events. Phytoplankton growth rates and remineralization rates increase with temperature with a $Q_{10}$ of 1.9, whereas zooplankton growth rates have a $Q_{10}$ of 2.14.

To evaluate PISCES simulations against in situ POC measurements or their proxies, the correspondence between PISCES tracers (in italics) and observed POC fractions must be established. In this study we assumed that SPOC corresponds to the sum of PISCES-simulated nanophytoplankton (*PHY*), diatoms (*PHY2*), small detrital particles (*POC*), and microzooplankton (*ZOO*), whereas LPOC corresponds to the sum of PISCES-simulated large detrital particles (*GOC*) and mesozooplankton (*ZOO2*) (Table 2). Total POC (TPOC) corresponds to the sum of those six PISCES tracers or, which is the same, SPOC + LPOC. Heterotrophic prokaryotes (*BACT*) are not a prognostic tracer in PISCES and are not explicitly included in our analysis. The correspondence between observed and simulated POC fractions, explicit and implicit, is discussed in Sect. 4.3.

### 2.5.1 PISCES 3D simulations vs. biome-aggregated observations

For the global-scale comparison between PISCES outputs and observations from BGC-Argo and satellites, we used the simulation presented by Aumont et al. (2017), which was forced by pre-computed dynamical fields from a pre-industrial run of the ocean circulation model NEMO. Global monthly climatological fields of the PISCES tracers were used to compute seasonal climatologies of modeled SPOC, LPOC, and TPOC. To enable a direct comparison to the BGC-Argo observations, model output was resampled at locations where BGC-Argo profiles were available over the 2010–2019 period. Prior to comparison with modeled fields, BGC-Argo observations were further screened to remove "outliers" in each biome and season. Outliers were defined as grid cells where the mean $b_{bp700}$ in the upper 50 m was above the 95 % percentile or greater than 0.008 m$^{-1}$ (Briggs et al., 2020). The same spatial resampling was applied to satellite-retrieved POC.

### 2.5.2 PISCES 1D simulations vs. BGC-Argo coherent annual time series (CATS)

A more detailed comparison was undertaken by matching each of the CATS from individual BGC-Argo floats with a PISCES water-column ("PISCES 1D") simulation. The match was based on the coherence between the seasonal cycle of MLD observed by the float and the turbulent layer simulated by NEMO. The pre-computed dynamical fields used to evaluate the match-ups were obtained from an ocean-only historical simulation (NEMO v3.6) at 1° resolution with 75 vertical levels (ORCA1_L75 grid) forced with the JRA-55 atmospheric reanalysis that covered the 1958–2018 period (Tsujino et al., 2020) following the OMIP2 protocol (Griffies et al., 2016). The float-observed MLD and the NEMO-simulated turbocline depth (defined by turbulent ver-

tical diffusivity $> 5 \times 10^{-4} \, \mathrm{m^2 \, s^{-1}}$) were compared over the entire annual cycle in all the model grid cells that had been visited by the float on a given year. The best model grid cell was selected based on model–observation correlation, root-mean-square error, and time lag in the onset date of summer stratification. An example of the metrics used to match NEMO dynamical fields and BGC-Argo MLD is provided in Figs. S1–S4. The associated model configuration and datasets are available in public repositories (Galí et al., 2021a and b).

PISCES 1D offline simulations were forced using the dynamical fields from the selected model grid cells. The same annual forcing, corresponding to the year of the BGC-Argo observations, was repeated over 5 simulation years. After 4 years of spin-up, the output from year 5 at a 5 d resolution was used for the comparison to the BGC-Argo CATS. Initial conditions (climatological fields of inorganic nutrients and carbon chemistry variables) and boundary conditions (atmospheric deposition) were the same as used for PISCES 3D (Aumont et al., 2015, 2017). Nutrient fields were restored towards the mean annual profile below 300 m. This procedure avoided drift in nutrient stocks by replenishing the upper ocean with the same amount of nutrients each year, resulting in regular seasonal cycles after 1 year and identical cycles from year 4 onwards.

## 3 Results

### 3.1 Climatological POC fields

This section describes the comparison among TPOC fields estimated from BGC-Argo and satellite observations and PISCES simulations across four ocean biomes. Figure 3 compares monthly climatologies at the sea surface (0–20 m), Fig. 4 compares seasonal climatologies between 0–1000 m, and Fig. 5 displays skill metrics (Pearson's correlation, model / observations ratio, and bias) for the vertical profiles shown in Fig. 4, as well as for the 1D simulations matched to BGC-Argo CATS.

### 3.1.1 Seasonally stratified subpolar biomes

In the subpolar biomes, near-surface TPOC ranged between $\sim 1 \, \mathrm{mmol \, m^{-3}}$ in the winter months and around 5 (subantarctic) or 10 (NASPG) $\mathrm{mmol \, m^{-3}}$ in early summer. In these biomes, PISCES-simulated TPOC was within the 2.5–97.5 % bounds of BGC-Argo observations for most months (Fig. 3). During the apex of the bloom (months 5–7), however, median PISCES estimates exceeded those obtained from BGC-Argo (by $\sim 80 \%$) and satellites (by $\sim 15 \%$). In the subantarctic, this pattern extended through the fall.

Satellite TPOC was in poor agreement with both PISCES and BGC-Argo TPOC outside the apex of the bloom. During the winter semester (months 10–12 and 1–3), and considering only the subset of pixels observed by both satellites

and floats, satellite TPOC exceeded BGC-Argo TPOC by a factor of 6.1 (factor of 3.3) in the NASPG (subantarctic), as discussed in Sect. 4.1.

PISCES reproduced the vertical decrease in TPOC concentration down to 1000 m (Fig. 4) with generally good skill (Fig. 5), but some misfits were observed. In the North Atlantic subpolar gyre (NASPG), PISCES underestimated TPOC through the epipelagic and the upper mesopelagic during the winter by $\sim 40 \%$ (Fig. 4) because of too vigorous convection in the NEMO dynamical fields that kept phytoplankton under insufficient light exposure. In the subantarctic biome, simulated TPOC exceeded BGC-Argo estimates in the upper portion of the mesopelagic layer in spring, and the overestimation pattern propagated downwards through the summer and fall. A similar but smaller overestimation pattern was observed in the NASPG in summer and fall.

### 3.1.2 Permanently and seasonally stratified subtropical biomes

In the oligotrophic biomes, monthly median surface TPOC displayed low seasonal amplitude. Total POC concentrations estimated from BGC-Argo data typically oscillated around $2 \, \mathrm{mmol \, m^{-3}}$, with a maximum / minimum ratio of around 1.6 in the Mediterranean and 1.3 in the Atlantic and Pacific subtropical gyres (STG). In the STG, satellite and BGC-Argo TPOC were in good agreement, which was to be expected because subtropical $b_{\mathrm{bp700}}$–POC conversion factors and satellite POC are based on the same study (Stramski et al., 2008). By contrast, PISCES TPOC exceeded BGC-Argo TPOC around 2-fold in the STG. In the Mediterranean, satellite TPOC exceeded BGC-Argo TPOC by $\sim 80 \%$, pointing to the differences in the respective POC estimation algorithms. PISCES TPOC was nearly fourfold higher than BGC-Argo TPOC at the surface in the Mediterranean, an overestimation that results from unrealistic physics in that basin caused by the too coarse (2°) model grid (Tonani et al., 2008; Lebeaupin Brossier et al., 2011) (see Sects. 3.3 and 4.3). Vertical TPOC profiles evidenced the shortcomings of PISCES simulations in the oligotrophic gyres. Compared to BGC-Argo profiles, PISCES simulations produced too sharp deep POC maxima and underestimated TPOC in the waters above and below (Fig. 4). These mismatch patterns prompted us to examine the seasonal cycles of POC in different biomes in greater detail.

### 3.2 Coherent annual time series of SPOC and LPOC: case studies

In this section we describe two CATS from BGC-Argo floats and their PISCES 1D counterparts (Sect. 3.2.1). The floats, identified by their World Meteorological Organization (WMO) number, are nos. 6901486 in the NASPG (year 2015) and the 6901660 in the South Pacific STG (year 2017). Float 6901486 represents the most productive conditions of

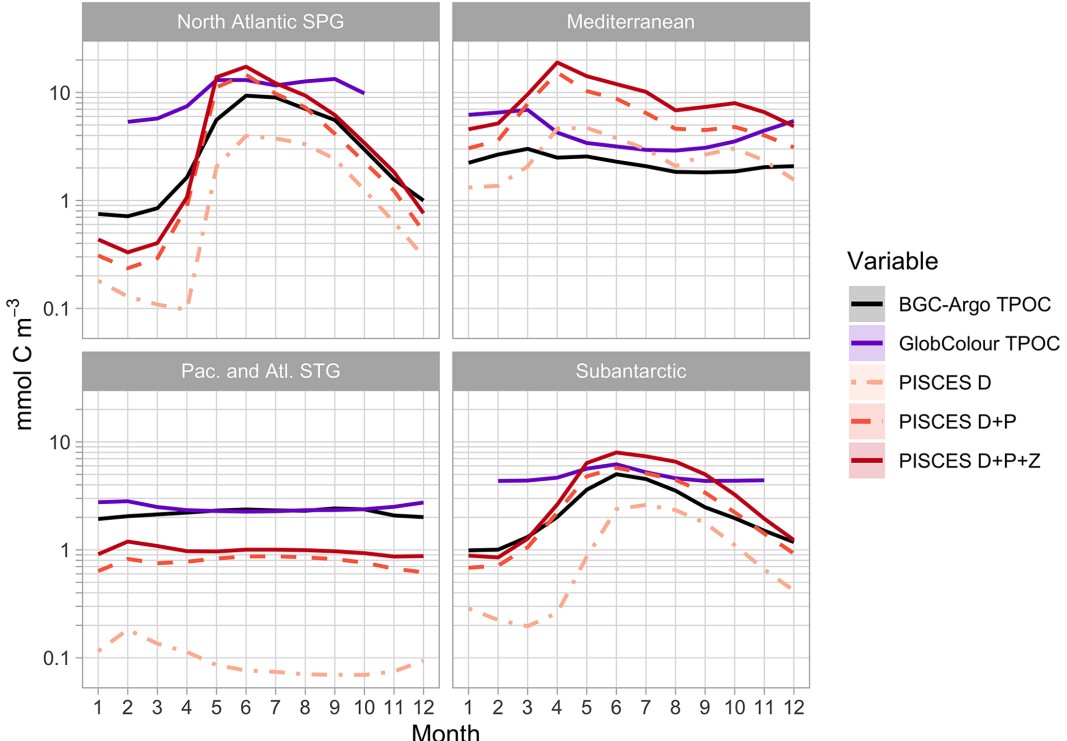

**Figure 3.** Monthly sea-surface particulate organic carbon (POC) concentration at the sea surface. Shown are POC estimates based on BGC-Argo (derived from the backscattering coefficient at 700 nm, $b_{bp700}$), satellite (GlobColour), and PISCES. PISCES TPOC results from the addition of the detritus (D), phytoplankton (P), and zooplankton (Z) tracers (Table 2), which are shown here as cumulative sums. Satellite data not shown for months when more than half of the ocean pixels could not be observed because of low solar elevation at high latitudes.

our dataset, with annual median (maximum) Chl $a$ of 0.60 (10.7) mg m$^{-3}$ and TPOC of 5.5 (16.7) mmol m$^{-3}$ in the near-surface layer (0–20 m). By contrast, float 6901660 represents the most oligotrophic waters, with annual median (maximum) Chl $a$ of 0.011 (0.036) mg m$^{-3}$ and TPOC of 1.8 (2.5) mmol m$^{-3}$ in the near-surface layer. In Figs. S5–S10 we provide additional examples of the CATS–PISCES 1D match-ups in the four biomes and in two subregions within the NASPG.

### 3.2.1 Labrador Sea (North Atlantic subpolar gyre)

Float 6901486 was deployed in June 2013 close to the Reykjanes Ridge in the Irminger Sea, NW Atlantic subpolar gyre. After drifting SW carried by the East Greenland Current, the float was trapped in the Labrador Sea cyclonic circulation between 2014 and July 2017, when it stopped communication after completing 344 profiles (cycles). During its multi-year sampling in the Labrador Sea (56–60° N latitude and 48–54° W longitude), over a bottom depth of around 3500 m, the float showed stable physical and bio-optical records at 1000 m, broken only by winter convection events, and recurring annual patterns of spring–summer phytoplankton blooming and vertical carbon export as depicted by Chl $a$ and POC profiles. Here we describe the

year 2015 (Fig. 6), characterized by deep convection during February and March, when the MLD generally exceeded 1000 m (Fig. S1). Epipelagic TPOC increased rapidly upon water-column re-stratification in mid-April and peaked in mid-May. A secondary bloom peaked in late June after a transient MLD deepening caused by stormy weather. Epipelagic TPOC decreased progressively thereafter until a small bloom was observed in October linked to pycnocline erosion. This bloom terminated rapidly and epipelagic TPOC reached the baseline level in late December. The SPOC fraction dominated epipelagic TPOC all year round, and the highest LPOC fractions of nearly 20 % were recorded at the apex of the spring–summer blooms. Distinct vertical particle export events were observed in May and June, matching the surface phytoplankton blooms, and August, when nutrient limitation likely triggered bloom collapse. These export pulses produced synchronous increases in SPOC and LPOC through the mesopelagic layer, though with different magnitudes. After reaching relative minima in October, mesopelagic SPOC and TPOC increased again in November, but they showed different vertical patterns.

The matching PISCES 1D simulation captured the patterns of SPOC and LPOC with good skill in the epipelagic and, to a lesser extent, the mesopelagic layer (Fig. 6). Excellent correlation ($r = 0.92$) and bias ($-0.5$ %) between BGC-Argo

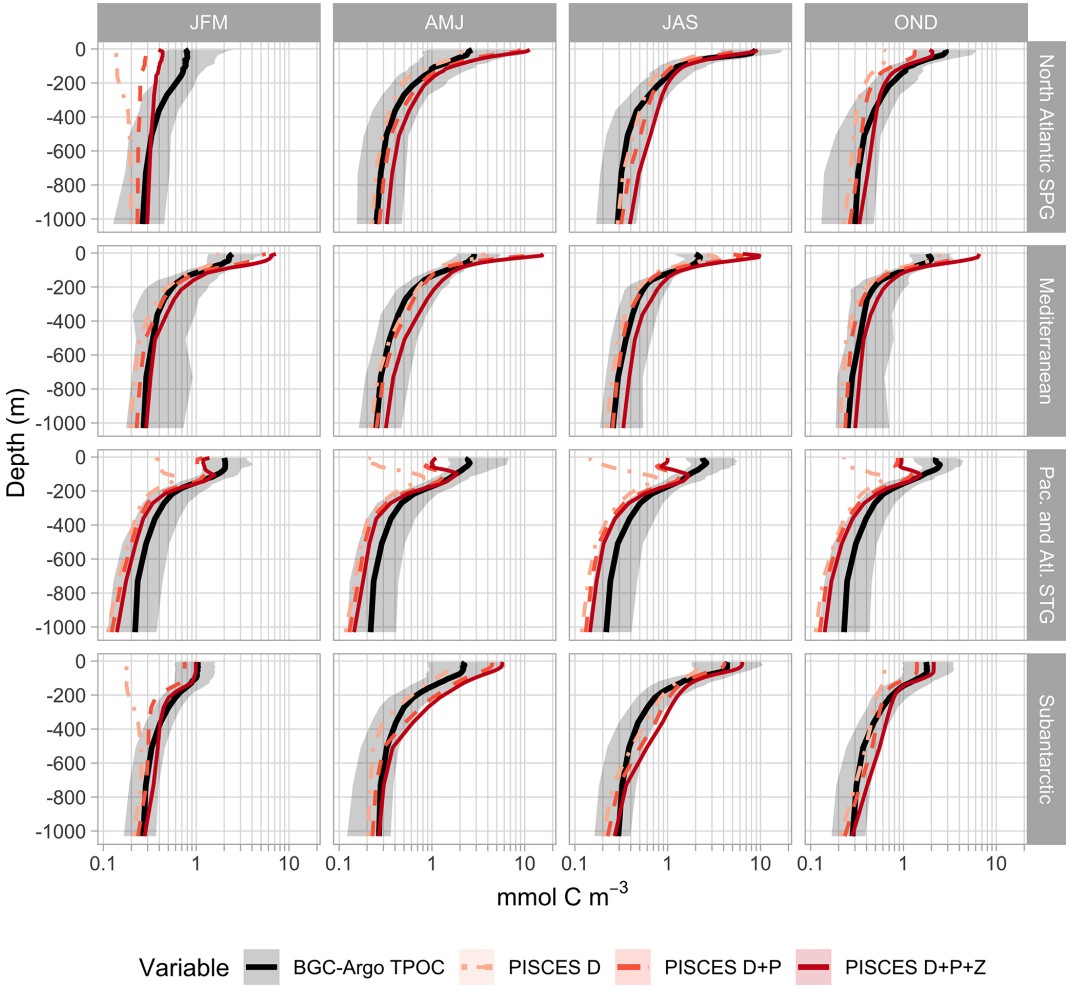

**Figure 4.** Seasonal 0–1000 m profiles of particulate organic carbon (POC). BGC-Argo estimates of TPOC (based on the backscattering coefficient at 700 nm, $b_{bp700}$) are represented with the median and the 0.025–0.975 quantiles within each biome. PISCES TPOC results from the addition of the detritus (D), phytoplankton (P), and zooplankton (Z) tracers (Table 2), which are shown here as cumulative sums.

and PISCES were found for vertically integrated epipelagic TPOC (Fig. 6i). A delayed start of the bloom was observed in the simulation, which can be partly attributed to a delay of around 1 week in the onset of permanent stratification in the model. It is also plausible that modeled phytoplankton reacted too weakly to the cessation of deep convection, which was captured by alternative MLD metrics in BGC-Argo profiles (Fig. 6a and b). Despite the general good agreement between observed and simulated SPOC, the model produced a conspicuous plume of SPOC that sank from the surface spring bloom into the mesopelagic layer, at the prescribed constant rate of $2 \, \mathrm{m \, d^{-1}}$, which was not found in the observations. At the core of this plume, PISCES POC exceeded BGC-Argo SPOC more than 2-fold. A similar model–observation mismatch was observed in all the northern and southern subpolar CATS as well as in some CATS in the Mediterranean (Figs. S5–S8 and S10). On average, simulated LPOC exceeded BGC-Argo LPOC by 36 % and 96 %

in the epi- and mesopelagic layers, respectively. The largest overestimation was observed during the midsummer export event. On the other hand, LPOC underestimation was found during the May bloom between 0–400 m.

PISCES qualitatively reproduced the late summer peak of mesopelagic LPOC, which was observed in all the subpolar North Atlantic CATS. By contrast, it failed to reproduce both the decrease in SPOC and LPOC in fall between 600–800 m and the LPOC increase below 800 m. The latter occurred in 6 out of 11 CATS in the NASPG, all located in the Labrador Sea. The apparent decoupling of deep mesopelagic LPOC from the overlying water column may be related to the insufficient temporal resolution of BGC-Argo profiling during that period compared to LPOC sinking speed or reflect LPOC export events from surface waters not located vertically over the float (Siegel and Deuser, 1997).

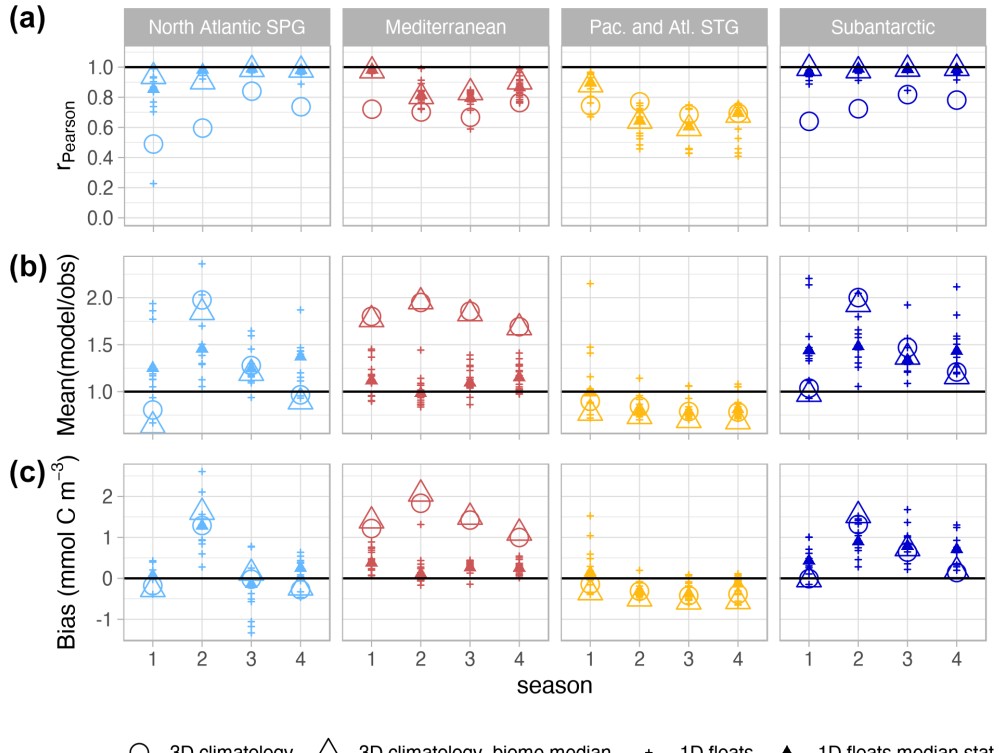

**Figure 5.** Summary of skill metrics for the comparison between PISCES and BGC-Argo particulate organic carbon (POC) profiles in different biomes and seasons. Pearson's correlation coefficient, mean model / data quotient, and mean bias (mmol m$^{-3}$) are computed in linear space (untransformed data). Horizontal black lines show the benchmark values for each statistic. Symbols represent different types of simulations and data aggregation levels. In the case of the PISCES 3D climatological simulation, the statistics are computed at two spatial aggregation levels: using all the grid cells with matching BGC-Argo profiles in a given biome (circles), and after averaging all the documented grid cells within the biome (empty triangles), as displayed in Fig. 4. In the case of the PISCES 1D simulations, statistics are computed for each individual CATS separately (small crosses), and the biome-season median is also shown (small filled triangles).

### 3.2.2 South Pacific subtropical gyre

Float 6901660 was deployed in March 2015 in the western South Pacific STG and drifted westwards until it deflected SW while approaching Tahiti. As of March 2021, the float was still active and had completed 244 cycles with stable continuous records at the 1000 m drift depth. Between July 2017 and June 2018 (18–21° S latitude and 148–157° E CE3 longitude), the period selected for the CATS analysis, the BGC-Argo profiles portrayed a stably stratified water column typical of the core of the subtropical gyres (Fig. 7). Vertical mixing events that reached a depth of around 100 m were observed in July, August, and October. However, their effect on surface SPOC was hardly noticeable, indicating that turbulent entrainment of nutrients was too weak to stimulate new production significantly. A deep Chl *a* maximum was present all year round between 150 and 200 m as identified by the maximum Chl *a* gradient. This Chl *a* maximum did not translate into a deep POC maximum. The fraction LPOC / TPOC was consistently around 6 % in the epipelagic and 12 % in the mesopelagic according to the vertically integrated stocks. The vertical–temporal distribution of LPOC was patchy in the lower mesopelagic (500–1000 m), perhaps reflecting the difficulty of detecting rare aggregates from $b_{bp700}$ spikes.

The epipelagic TPOC stock (driven by SPOC) simulated by PISCES matched the observations well, with a high temporal correlation coefficient ($r = 0.69$; Fig. 7i) despite the low seasonal variability in this tropical setting. PISCES overestimated SPOC between the base of the mixed layer and the deep Chl *a* maximum and underestimated SPOC below it. Thus, the low mean bias of epipelagic SPOC (9 %) resulted from these mutually compensating biases. In the mesopelagic layer, vertically integrated PISCES SPOC was on average 45 % lower than BGC-Argo SPOC and showed low negative temporal correlation to the observations. Regarding LPOC, the simulated stock was nearly twice as large as BGC-Argo estimates in the epipelagic, with the largest overestimation seen in the deep Chl *a* maximum. In the mesopelagic, PISCES LPOC exceeded BGC-Argo estimates by 25 % on average. Despite these deviations, PISCES simulations supported the increase in the LPOC / TPOC fraction

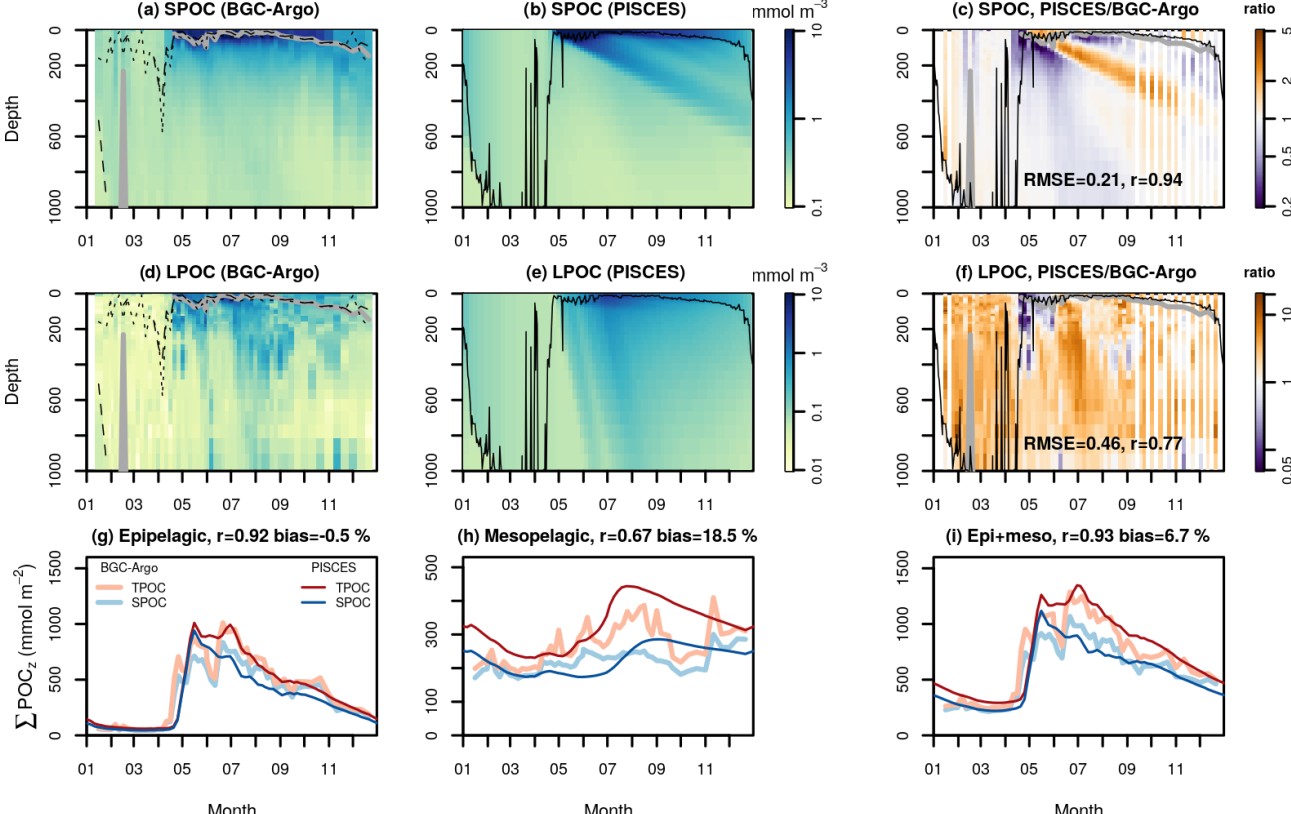

**Figure 6.** Temporal evolution of small and large POC concentrations (SPOC and LPOC, respectively) in the North Atlantic subpolar gyre (NASPG). Panels **(a)** and **(d)** show observations from the BGC-Argo float WMO 6901486, which drifted in the Labrador Sea during the year 2015. Panels **(b)** and **(e)** show the corresponding PISCES 1D simulation. Panels **(c)** and **(f)** show the ratio between the model and the observations and the corresponding correlation coefficient and RMSE in $\log_{10}$ scale. Solid lines depict the observed MLD (**a** and **d**, in gray) and the simulated turbocline depth (**b** and **e**, in black). In panels **a** and **d**, the dotted lines show the depths of the maximum Chl *a* fluorescence gradient and the dashed lines an alternative MLD estimated with a 0.005 °C temperature threshold that captures weak stratification. Panels **(g)**, **(h)**, and **(i)** show the vertically integrated SPOC and LPOC stocks in different layers: 0–200 m or epipelagic **(g)**, 200–1000 m or mesopelagic **(g)**, and 0–1000 m **(i)**. On top of panels **(g)**–**(i)** we show the correlation and mean bias between BGC-Argo and PISCES total POC (TPOC).

between the epipelagic and the mesopelagic layers deduced from BGC-Argo profiles.

### 3.3 Coherent annual time series of SPOC and LPOC: generalized approach

Although each of the 50 CATS included in this study has unique features, some of the misfit patterns between PISCES and BGC-Argo data described in the previous section are common to most CATS in a given biome or, more broadly, in subpolar vs. subtropical biomes. In this section we generalize the quantitative comparison between the 50 BGC-Argo CATS and their PISCES 1D counterparts across the four biomes. We consider separately the epipelagic and mesopelagic domains, focusing on mean annual SPOC and LPOC standing stocks (Fig. 8), the SPOC / TPOC fraction (Fig. 9), and the mesopelagic Teff of SPOC and LPOC (Fig. 10).

Epipelagic SPOC stocks ranged between 193–425 mmol C m$^{-2}$ (1.0–2.1 mmol C m$^{-3}$ in concentration units) according to BGC-Argo observations. The SPOC range in the corresponding PISCES 1D simulations was 282–537 mmol C m$^{-2}$ (1.4–2.7 mmol C m$^{-3}$). Although the ranges of the different biomes overlapped, smaller stocks were usually found in the more oligotrophic (STG and Mediterranean) biomes. The agreement between simulated and observed epipelagic SPOC was better in the STG and NASPG biomes, whereas simulated epipelagic SPOC exceeded observations by around 50 % in the Mediterranean and subantarctic biomes. Simulated and observed stocks were positively correlated ($r = 0.45$, $p = 9 \times 10^{-4}$).

In the mesopelagic domain, observed and simulated SPOC stocks ranged between 145–283 and 105–308 mmol C m$^{-2}$, respectively (corresponding to concentration ranges of 0.18–0.35 and 0.13–0.39 mmol C m$^{-3}$). The correlation between simulated and observed mesopelagic SPOC stocks was not significant ($r = -0.08$, $p = 0.55$). PISCES SPOC was sim-

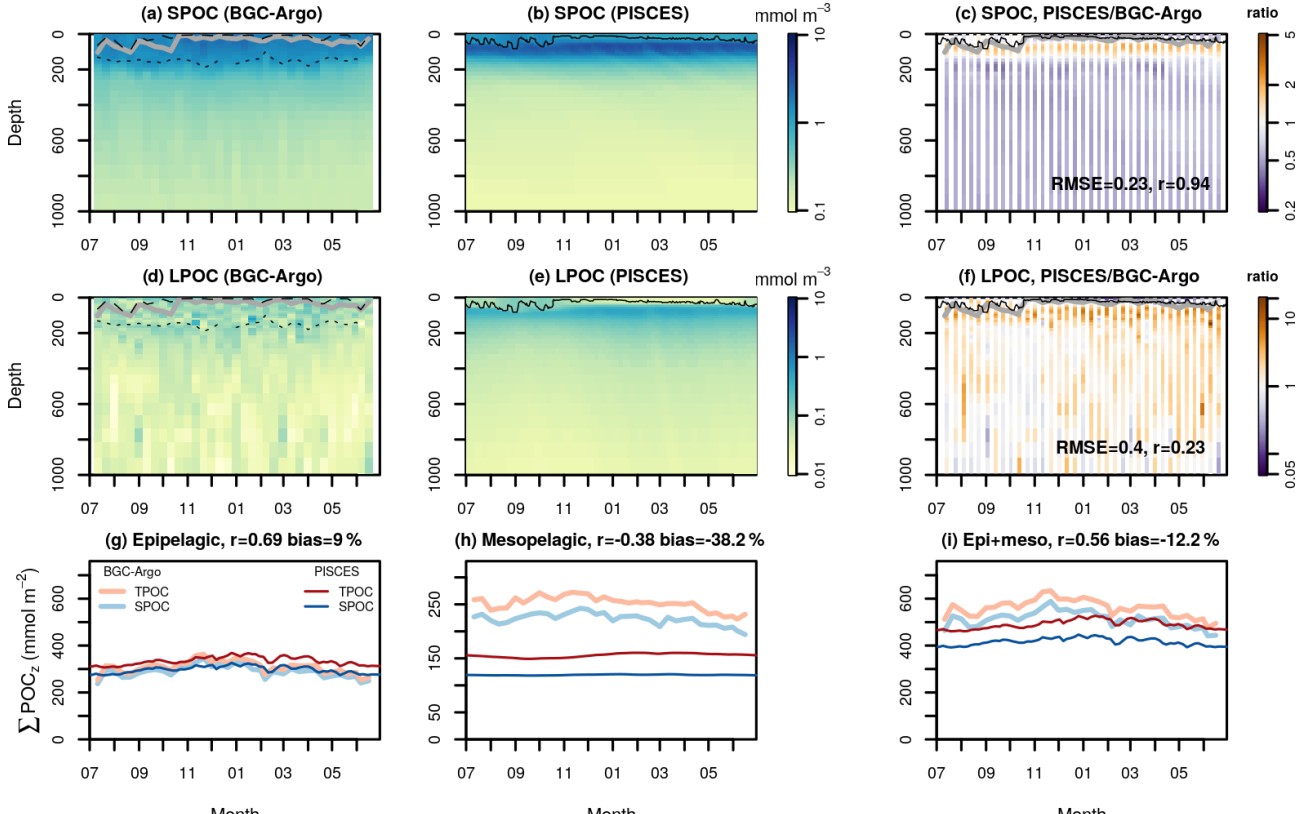

**Figure 7.** Temporal evolution of small and large POC concentrations (SPOC and LPOC, respectively) in the South Pacific subtropical gyre (STG). Panels (**a**) and (**d**) show observations from the BGC-Argo float WMO 6901660, which drifted westwards near Tahiti, during the year 2016. Panels (**b**) and (**e**) show the corresponding PISCES 1D simulation. Panels (**c**) and (**f**) show the ratio between the model and the observations and the corresponding correlation coefficient and RMSE in $\log_{10}$ scale. Solid lines depict the observed MLD (**a** and **d**, in gray) and the simulated turbocline depth (**b** and **e**, in black). In panels (**a**) and (**d**), the dotted lines show the depths of the maximum Chl *a* fluorescence gradient and the dashed lines an alternative MLD estimated with a 0.005 °C temperature threshold that captures weak stratification. The bottom panels show the vertically integrated SPOC and LPOC stocks in different layers: 0–200 m or epipelagic (**g**), 200–1000 m or mesopelagic (**g**), and 0–1000 m (**i**). On top of panels (**g**)–(**i**) we show the correlation and mean bias between BGC-Argo and PISCES total POC (TPOC).

ilar to or slightly higher than BGC-Argo SPOC in the subpolar biomes but up to 2-fold lower in the STG and Mediterranean biomes, as previously shown in Figs. 4 and 7.

Epipelagic LPOC stocks were smaller and showed wider inter-biome variability than SPOC stocks, with around 2-fold higher LPOC in subpolar biomes. The positive correlation between simulated and observed LPOC was highly significant ($r = 0.78$, $p = 2 \times 10^{-11}$). Yet, PISCES LPOC (range 19–83 mmol C m$^{-2}$, 0.10–0.42 mmol C m$^{-3}$) typically exceeded BGC-Argo LPOC (range 27–119 mmol C m$^{-2}$, 0.14–0.60 mmol C m$^{-3}$) by around 50 % and up to 3-fold for some CATS. A lower but still highly significant correlation was found in the mesopelagic ($r = 0.64$, $p = 5 \times 10^{-7}$), where PISCES LPOC was usually within a factor of 1.5 of observations, except for the NASPG biome where it exceeded BGC-Argo LPOC around 2-fold.

The SPOC / TPOC fraction showed low variability in the epipelagic layer (Fig. 9), with a median (range) of 89 %

(84 %–94 %) and 85 % (75 %–91 %) for BGC-Argo and PISCES, respectively. The simulated SPOC / TPOC fraction was within ± 10 % of BGC-Argo estimates for all biomes except the NASPG, where PISCES tended to underestimate the observed SPOC / TPOC fraction. A significant positive correlation ($r = 0.56$, $p = 2 \times 10^{-5}$) was found between simulations and observations in the epipelagic layer. The SPOC / TPOC fraction was lower in the mesopelagic layer according to both BGC-Argo (75 %–90 %) and PISCES (66 %–85 %) estimates. In this case, CATS from the subantarctic biome showed excellent model–observation agreement, whereas PISCES was 20 % lower than BGC-Argo estimates in the other three biomes. No significant correlation was found between the simulated and observed SPOC / TPOC fraction in the mesopelagic.

Transfer efficiency (Teff; Fig. 10) was computed as the ratio between the SPOC or LPOC concentrations in the depth bins centered at 697 m (range 662–734 m) and 200 m (range

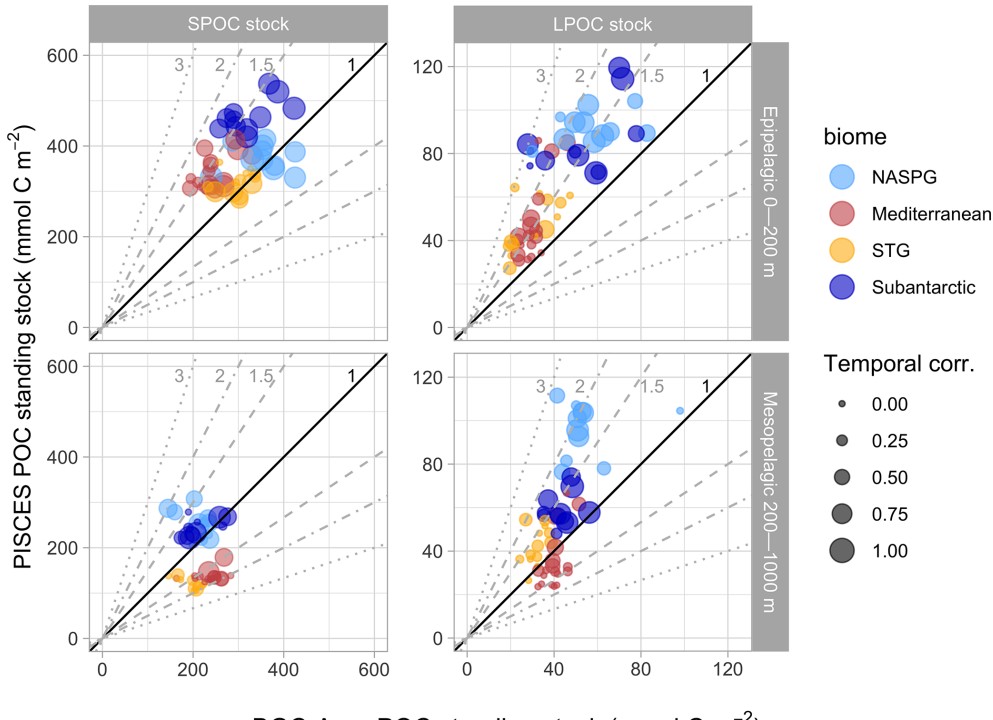

**Figure 8.** Mean annual POC stocks. BGC-Argo versus PISCES scatterplots are shown for the standing stocks (vertical integrals) of small POC (SPOC) and large POC (LPOC) in the epipelagic (0–200 m) and mesopelagic (200–1000 m) layers. Reference lines indicate a range of model / data ratios, from 1 : 1 (perfect correspondence) to a factor of 3 or its inverse. Biomes are distinguished with different colors, and the size of the circles is proportional to the temporal correlation between BGC-Argo and PISCES stocks at each location (as shown in the bottom panels of Figs. 6 and 7).

190–210 m). The shallowest bin corresponds to the bottom of the epipelagic layer, and the 500 m interval was chosen following previous studies (Lam et al., 2011; Dall'Olmo and Mork, 2014). Different depth ranges between 180 and 800 m gave similar Teff patterns. The analysis of Teff yielded some important insights:

1. SPOC and LPOC showed similar Teff, both in the observations (medians of 0.39–0.41) and in the model (medians of 0.27–0.29). Similar patterns were found when each biome was considered separately.

2. BGC-Argo Teff usually exceeded PISCES Teff and spanned a wider range. The best agreement was generally found in the STG and the subantarctic biomes, and the poorest agreement occurred in the Mediterranean where BGC-Argo Teff was typically twice the modeled Teff.

3. Distinct patterns were found for four CATS in the Labrador Sea (NASPG) characterized by high epipelagic TPOC ($> 440$ mmol C m$^{-2}$). In this subset, PISCES and BGC-Argo Teff were in good agreement for SPOC (median 0.40 for both), whereas, for LPOC, PISCES Teff (0.36–0.45) doubled BGC-Argo

Teff (0.12–0.28), which had some of the lowest values of the dataset. These CATS showed a LPOC minimum at 600–800 m in fall, which PISCES could not reproduce (Sect. 3.2.1).

4. PISCES and BGC-Argo Teff showed a weak positive correlation for SPOC ($r = 0.26$, $p = 0.07$), with some improvement when the Labrador Sea CE4 outliers were removed ($r = 0.34$, $p = 0.02$). A negative correlation between simulations and observations was found in the case of LPOC Teff ($r = -0.29$, $p = 0.04$), which became non-significant without the Labrador Sea outliers ($r = 0.04$, $p = 0.81$).

Our observational estimates are sensitive to the choice of $b_{bp700}$–POC conversion factors. These factors are especially uncertain in the mesopelagic due to data scarcity, which prompted us to use a constant global value for "$c$" in Eq. (1) (Fig. 2); $c = 1000$ mmol C m$^{-2}$ is the asymptotic POC–$b_{bp700}$ conversion factor below 1000 m, taken from Cetinić et al. (2012). To address this uncertainty, we conducted sensitivity tests where $c$ was halved or doubled. The resulting range of 500–2000 mmol C m$^{-2}$ is probably generous, as indicated by our indirect estimates based on the studies of Bishop (1999) and Bishop et al. (1999) (Appendix A). As

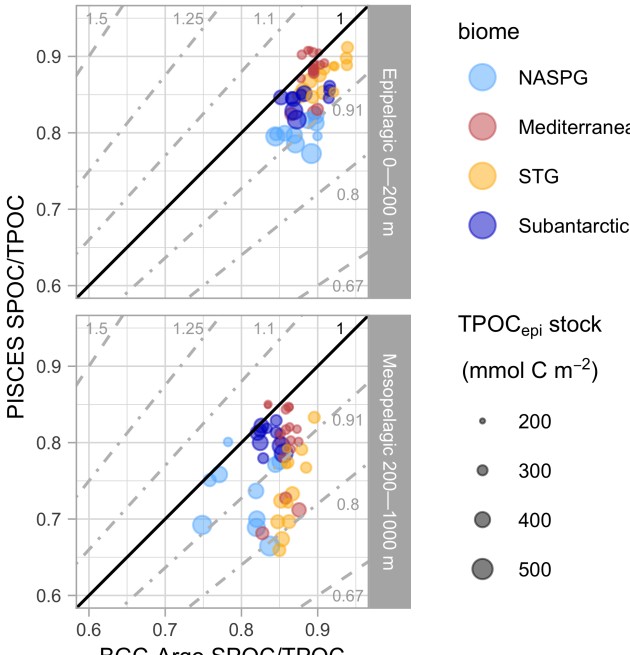

**Figure 9.** Mean annual SPOC / TPOC fractions. BGC-Argo versus PISCES scatterplots are shown for the epipelagic (0–200 m) and mesopelagic (200–1000 m) layers. Reference lines indicate a range of model : data ratios, from 1 : 1 (perfect correspondence) to a factor of 1.5 or its inverse. Biomes are distinguished with different colors, and the size of the circles is proportional to the annual mean TPOC stock in the epipelagic layer, used as an indicator of productivity.

expected, changing $c$ had little effect on epipelagic POC, a larger effect on mesopelagic POC and Teff, and no effect on the SPOC / TPOC fraction (because the same conversion factor is used to estimate SPOC and LPOC). Halving $c$ resulted in steeper POC profiles, which brought mesopelagic SPOC closer to the 1 : 1 line in the STG and Mediterranean, at the expense of increasing the SPOC bias in the subpolar biomes and that of LPOC everywhere (Fig. S11). Doubling $c$ caused a less steep vertical decrease in BGC-Argo POC, which overall worsened the model–observation agreement in the mesopelagic (Fig. S12), except for LPOC stocks in the NASPG.

## 4   Discussion

### 4.1   Towards a globally consistent picture of POC fields in observations and models

Global quantification of POC stocks through the water column has been elusive until recently because of data sparseness and limitations of model parameterizations, both of which are especially severe below the epipelagic. Our joint analysis of PISCES simulations, satellite observations, and over 70 000 BGC-Argo vertical profiles reveals a globally

consistent picture across the epi- and mesopelagic layers (Figs. 3–10). The global ocean stock of TPOC simulated by PISCES amounts to 4 Pg C, shared in a proportion of 39 % (1.6 Pg C), 25 % (1 Pg C), and 36 % (1.4 Pg C) between the epi-, meso-, and bathypelagic layers (Table 3; the bathypelagic is defined here as 1000–5000 m to include the entire model domain in the stock calculation).

The PISCES-simulated TPOC concentration is on average within a factor of 1.56 (1.42) of BGC-Argo estimates for the median (mean) seasonal biome profiles shown in Fig. 4 (Mediterranean excluded). Aumont et al. (2017) reported a similar reliability index of 1.6 for the comparison between PISCES and in situ chemically determined POC profiles. Thus, our evaluation lends further confidence to the POC reactivity continuum parameterization implemented in PISCES, which represents both SPOC and LPOC as a mixture of fractions with different lability (Aumont et al., 2017), in globally representative biomes.

The epipelagic TPOC stock simulated by PISCES, 1.6 Pg C, is comparable to previous observational estimates. Gardner et al. (2006) estimated the global POC stock within the first optical attenuation depth using a compilation of in situ POC and $c_p$ measurements leveraged by ocean color satellite data. They obtained a global stock of 0.43 Pg C, and invoked some scaling arguments to estimate that the total POC stock down to middle-mesopelagic "background levels" would range from 1 to 2 Pg C. Stramska (2009) obtained a larger global epipelagic POC stock of 1.8–2.3 Pg C using the satellite algorithm of Stramski et al. (2008). Our match-up analysis indicates that satellite TPOC exceeds BGC-Argo estimates by up to 7-fold at high latitudes outside of the summer season (Fig. 3). This deviation is well beyond the nominal uncertainty of the satellite POC product (< 30 %) and the range of observed POC / $b_{bp700}$ ratios at the sea surface (Fig. 2). Therefore, we conclude that the algorithm of Stramski et al. (2008) overestimates POC at high latitudes in winter, an issue that deserves further investigation. Potential explanations are the satellite algorithm being calibrated mostly against samples from lower latitudes (50° N–30° S), or its sensitivity to the differential atmospheric attenuation of blue and green wavelengths at low solar elevation.

Recently, Evers-King et al. (2017) calculated the global POC stock in the upper mixed layer using several satellite algorithms (including that of Stramski et al., 2008) and indicated that 0.77–1.3 Pg C was a plausible range. Our PISCES-based estimate for the mixed-layer POC stock, using the same MLD climatology (Schmidtko et al., 2013), is 0.58 Pg C. The lower PISCES-derived estimate may arise from the combination of POC overestimation by some satellite algorithms, discussed above, with PISCES' tendency to underestimate mixed-layer POC in oligotrophic areas (Figs. 4 and 7). This negative bias in PISCES is compensated by a positive POC bias in deep Chl $a$ maxima (below the MLD), resulting in smaller deviations between BGC Argo and PISCES over the entire epipelagic layer in the STG

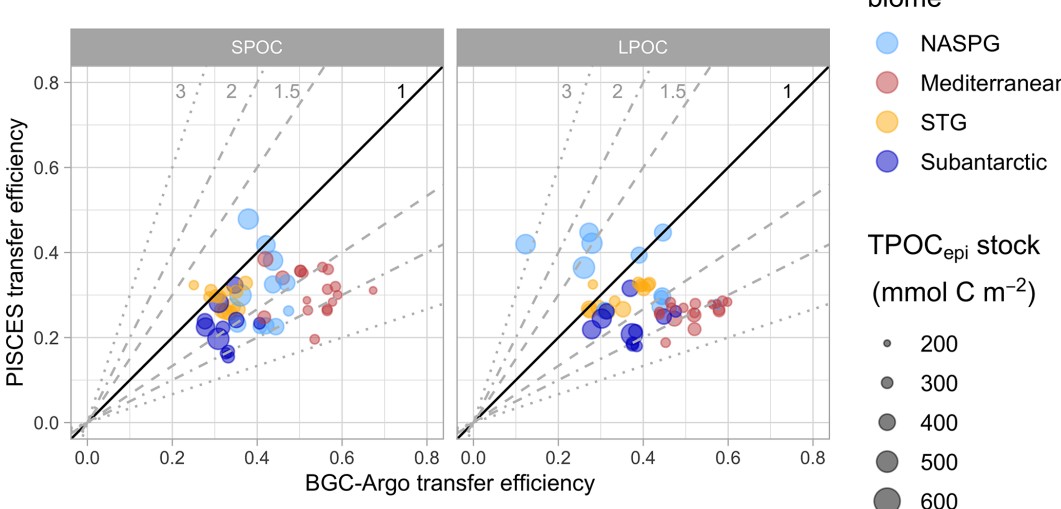

**Figure 10.** Mean annual POC transfer efficiency between 200 and 700 m. BGC-Argo versus PISCES scatterplots are shown for small POC (SPOC) and large POC (LPOC). Reference lines indicate a range of model / data ratios, from 1 : 1 (perfect correspondence) to a factor of 3 or its inverse. Biomes are distinguished with different colors, and the size of the circles is proportional to the annual mean TPOC stock in the epipelagic layer (0–200 m), used as an indicator of productivity.

biome (Figs. 8 and S13). Given the limited coverage of in situ seawater sampling and satellite observations in some regions and seasons (Evers-King et al., 2017), further intercomparison between those observations, BGC-Argo data, and models is needed to better constrain epipelagic POC stocks.

POC in the lower mesopelagic and below has been traditionally treated as a "background" signal (Bellacicco et al., 2019; Gardner et al., 2006; Loisel and Morel, 1998). This approach is convenient for studies that focus solely on upper-ocean processes because POC concentration decreases exponentially with depth. Yet, our global estimates and several previous studies highlight the need to turn our attention to the large POC stocks ($> 2$ Pg C) that reside in the meso- and bathypelagic layers, whose dynamics are still poorly understood. In line with previous studies (Dall'Olmo and Mork, 2014; Poteau et al., 2017), we showed that mesopelagic POC exhibits clear seasonal cycles in productive regions (Figs. 6, S5–S7, and S10), owing to their connection with the upper-ocean through numerous biological and physical processes (Boyd et al., 2019; Briggs et al., 2020). Despite being less reactive on average than upper-ocean POC, meso- and bathypelagic organic particles are microbial hotspots that host key biogeochemical functions, from enzymatic decomposition of macromolecules (Arnosti et al., 2012; Baltar et al., 2010a, b) to aerobic and anaerobic respiration (Bianchi et al., 2018; Karthäuser et al., 2021) and chemosynthesis (Arístegui et al., 2009; Herndl and Reinthaler, 2014; Pachiadaki et al., 2017). Moreover, mesopelagic particles are consumed by upper trophic levels that sustain fisheries (Bode et al., 2021; Woodstock et al., 2021).

In situ bio-optical measurements are poised to play a key role in monitoring marine POC stocks in layers that cannot be accessed by remote sensing. For example, Sauzède et al. (2020) merged BGC-Argo and satellite observations to obtain a dynamic 3D view of particle backscattering. Using a data-driven machine learning approach, they were able to predict the profiles of $\log_{10} b_{\mathrm{bp700}}$ measured by two BGC-Argo floats in the NASPG and STG biomes ($R^2$ of $\sim 0.85$ and mean absolute percentage deviation of $\sim 12\%$) from the sole knowledge of physical properties of the water column and surface ocean color (remote-sensing reflectance). Their estimates were recently extended to POC (Sauzède et al., 2021), which can be of great utility for constraining biogeochemical models. Here we took an entirely different approach, based on converting available $b_{\mathrm{bp700}}$ data to POC with a simple empirical algorithm (Fig. 2) and then comparing it to the outputs of the PISCES model. Our PISCES-based estimates obtained a median $R^2 = 0.86$ and mean absolute percentage deviation of 38 % (5 d depth-binned $\log_{10}$POC; 50 CATS from 28 globally distributed floats). This good skill is remarkable because neither the empirical POC estimates nor PISCES were tuned to maximize their mutual agreement. Still, our study shows that the comparison of bio-optics-derived POC measurements and PISCES is affected by different types of uncertainty that we analyze in the following sections.

## 4.2 Bio-optical underpinnings of POC fields based on BGC-Argo observations

Accurate interconversion between bio-optical variables and concentrations is key for constraining ocean particle dynam-

**Table 3.** Global and regional POC stocks, concentrations, and fractions in different layers as simulated by PISCESv2_RC.

| Variable | Depth range (m) | Open-ocean biomes | | | | | Global[d] |
|---|---|---|---|---|---|---|---|
| | | Ice | SPSS[a] | STSS[b] | STPS[c] | Equatorial | |
| Area (%) | | 6.5 | 14.3 | 12.0 | 46.0 | 9.2 | 100 |
| TPOC (Tg C) | 0–200 | 73 | 281 | 263 | 263 | 154 | 1590 |
| | 200–1000 | 57 | 221 | 173 | 397 | 90 | 1006 |
| | 1000–5000 | 80 | 274 | 237 | 633 | 144 | 1438 |
| TPOC (mmol C m$^{-3}$) | 0–200 | 1.18 | 2.22 | 2.57 | 1.66 | 1.99 | 1.85 |
| | 200–1000 | 0.29 | 0.48 | 0.44 | 0.25 | 0.29 | 0.33 |
| | 1000–5000 | 0.14 | 0.18 | 0.17 | 0.10 | 0.13 | 0.13 |
| SPOC[e] (%) | 0–200 | 72 | 81 | 79 | 89 | 84 | 81 |
| | 200–1000 | 75 | 75 | 70 | 79 | 71 | 75 |
| | 1000–5000 | 81 | 80 | 77 | 79 | 75 | 79 |
| Phyto[f] (%) | 0–20 | 45 | 45 | 41 | 58 | 49 | 50 |
| Diatoms[g] (%) | 0–20 | 28 | 21 | 14 | 3 | 4 | 14 |
| Non-phyto[h] (%) | 0-20 | 55 | 55 | 59 | 42 | 51 | 50 |
| Detritus[i] (%) | 0–20 | 15 | 25 | 31 | 26 | 31 | 24 |
| | 0–200 | 36 | 43 | 48 | 44 | 50 | 42 |
| | 200–1000 | 62 | 67 | 70 | 78 | 78 | 70 |

[a] Subpolar seasonally stratified. [b] Subtropical seasonally stratified. [c] Subtropical permanently stratified. [d] Larger than the sum of biomes because the latter do not include coastal areas. [e] SPOC / TPOC. [f] $(PHY + PHY2)$ / TPOC. [g] $PHY2$ / TPOC. [h] $(POC + GOC + ZOO + ZOO2)$ / TPOC. [i] PISCES detritus $(POC + GOC)$ divided by TPOC. Heterotrophic prokaryotes are de facto included in TPOC.

ics and their model representation (Bishop et al., 2004; Gardner et al., 2006). Here we discuss the processes that appear to drive, to first order, the variability of the POC / $b_{bp700}$ ratios embodied in Eqs. (1) and (2) (Fig. 2; Appendix A) and the main strengths and weaknesses of our scheme.

Changes in the trophic status appear as the primary driver of POC / $b_{bp700}$ variability in the epipelagic layer (Cetinić et al., 2012; Fig. A1). Productive waters host greater absolute and relative abundance of diatoms (Uitz et al., 2006) (see also Table 3), which have lower POC per cell volume (Menden-Deuer and Lessard, 2000) and are covered with silica frustules that may scatter light more efficiently than naked cells (Twardowski et al., 2001), altogether resulting in lower POC content per unit $b_{bp700}$ (Cetinić et al., 2012; Oubelkheir et al., 2005). The proportions of different autotrophic and heterotrophic organisms and detritus are also likely to vary with upper-ocean productivity (see Sect. 4.3). If the mass-specific backscattering coefficients of these components were better known, their systematic variation patterns could be used to develop a continuous formulation for POC / $b_{bp700}$ rather than the regionalized conversion factors used here. However, POC / $b_{bp700}$ is influenced by other seawater constituents whose occurrence is less predictable. Foremost, biogenic calcite (e.g., from coccolithophores) and desert dust (Claustre, 2002; Loisel et al., 2011), both of which enhance $b_{bp700}$ CE5. In our dataset, we detected a CATS (float WMO 6901647,

year 2016) that was strongly affected by coccolith backscattering in the Iceland Basin, an area known for its massive coccolithophore blooms (Moore et al., 2012). This CATS was an obvious outlier in our model–observation scatterplots (Fig. S13), likely because the enhancement of $b_{bp700}$ caused by coccoliths had no translation in TPOC, and was therefore removed from the analysis.

The decrease in POC / $b_{bp700}$ along the vertical axis (Fig. 2) reflects the increase in the particles' index of refraction, hence the backscattering ratio (Cetinić et al., 2012; Nencioli et al., 2010). This change is likely caused by the remineralization of organic materials (Martin et al., 1987) that leaves a higher mineral fraction (Honjo et al., 2008; Lam et al., 2011) and the increase in the structural complexity of aggregates with depth (Organelli et al., 2018). According to our sensitivity analysis (Figs. S11 and S12), the prescribed exponential decrease in POC / $b_{bp700}$ towards a constant POC / $b_{bp700}$ ($c = 1000$ mmol C m$^{-2}$) at depths > 1000 m provides a good compromise globally, given the limited knowledge of mesopelagic POC / $b_{bp700}$ and its variability across regions. However, the comparison between simulated and observed mesopelagic SPOC (and thus TPOC) is more favorable in subpolar than in subtropical biomes. In the latter, better model–observation agreement was found when POC / $b_{bp700}$ was halved ($c = 500$ mmol C m$^{-2}$). A lower $c$ would also bring the simulated Teff for SPOC

Please note the remarks at the end of the manuscript.

and LPOC closer to BGC-Argo estimates. It is tempting to hypothesize that lower latitudes have lower mesopelagic POC / $b_{bp700}$ owing to the greater proportion of calcite (Francois et al., 2002; Honjo et al., 2008; Lam et al., 2011, 2015). These hypotheses need further verification.

Finally, the decrease in surface POC / $b_{bp700}$ with deeper vertical mixing imposed by Eq. (2) partially reflects the dilution of surface particle assemblages by entrainment of deeper waters (Lacour et al., 2019) with lower POC / $b_{bp700}$ (Bol et al., 2018). Modulation of POC / $b_{bp700}$ by vertical mixing improves the agreement between PISCES and BGC-Argo data in regions with wide seasonal MLD amplitude such as the NASPG. Still, our approach should be seen as a simplistic first-order approximation, and alternative formulations should be further evaluated when more data become available. For example, POC / $b_{bp700}$ might be kept constant regardless of the MLD or estimated for each profile as the average within the mixed layer of the pre-computed POC / $b_{bp700}$ profile (Eq. 1). In both cases, however, POC / $b_{bp700}$ would have to decrease abruptly below the mixed layer to meet the low POC / $b_{bp700}$ in the mesopelagic, producing sharp discontinuities that have not been observed. On the other hand, the behavior prescribed by Eq. (2) may not be appropriate for situations when vertical mixing cannot erode the seasonal thermocline or pycnocline because in such cases it will entrain water from the deep Chl $a$ maximum, and not mesopelagic water, to the upper mixed layer.

Our POC–$b_{bp700}$ conversion algorithm omits several other sources of uncertainty. Foremost, it assumes that SPOC and LPOC have the same POC / $b_{bp700}$ ratio, which largely corresponds to that of the more abundant SPOC (Figs. 6–8). In addition, the in situ data used to parameterize Eqs. (1) and (2) are not free of uncertainty (Cetinić et al., 2012; Lam et al., 2011; Morán et al., 1999; Organelli et al., 2018; Strubinger Sandoval et al., 2021). Despite these limitations, the POC–$b_{bp700}$ conversion scheme used here provides POC estimates that are generally consistent with PISCES (Fig. 8) and with previous assessments (Aumont et al., 2017). Moreover, this scheme allowed us to identify potential shortcomings of satellite-based assessments (Evers-King et al., 2017; Stramski et al., 2008). The development of more sophisticated POC–$b_{bp700}$ interconversion schemes is desirable and would greatly benefit from the measurement of mass-specific bio-optical properties of various seawater constituents across different ocean basins and depths.

### 4.3 Correspondence between observed and simulated POC fractions

An additional source of uncertainty in our analysis is the imperfect match between the PISCES tracers and the observable POC fractions (Table 2). A positive aspect of our evaluation is the reasonable agreement between the simulated SPOC / TPOC fraction and the BGC-Argo estimates based on the $b_{bp700}$ spike signal (Fig. 9). Our estimates converge to a global median value of 85 %, with 95 % of the data between 69 %–92 %, fully within the range of size-fractionated chemical POC determination (Aumont et al., 2017, and references therein). Our SPOC / TPOC estimates can also be compared to the suspended POC estimated with the marine snow catcher during spring in subpolar epipelagic and upper mesopelagic waters (Baker et al., 2017). The slow-sinking POC measured by the marine snow catcher should not be included in this comparison because it sinks at around 18 m d$^{-1}$, a range more typical of particles > 100 µm (Cael et al., 2021). The median 94 % of suspended POC reported by Baker et al. (2017) is higher than the 86 % (81 %) SPOC / TPOC estimated here from BGC-Argo (PISCES) in the subpolar biome, suggesting further comparison between different approaches is needed. The tendency of SPOC / TPOC to decrease between the epipelagic and the mesopelagic, found in both PISCES and BGC-Argo data (Fig. 9, Table 3), was also reported in previous studies (Aumont et al., 2017, and references therein; Organelli et al., 2020), lending further confidence to our estimates. On the other hand, the simulated partitioning of POC into different living and detrital compartments is probably less realistic, as discussed in greater detail below.

An aspect that deserves further attention is the partitioning between phytoplanktonic (autotrophic), heterotrophic, and detrital POC in the upper ocean. Unfortunately, this partitioning is far from being well established from observations. Using bio-optical data, Oubelkheir et al. (2005) estimated that detritus accounted for around 60 % of POC across a wide range in ocean productivity, and 70 %–85 % of POC was assigned to non-phytoplanktonic material (detritus + heterotrophs). These percentages accord well with those found by Claustre et al. (1999) in the tropical Pacific (their Table 2; see also Organelli et al., 2020). By contrast, Graff et al. (2015) found wide latitudinal variations in the non-phytoplanktonic POC, increasing between ∼ 20 % and ∼ 80 % from the tropical to the temperate Atlantic. Bellacicco et al. (2019) analyzed the covariation between $b_{bp700}$ and Chl $a$ in the (much larger) BGC-Argo dataset. They concluded that the background $b_{bp700}$, defined as the $b_{bp700}$ that does not covary with Chl $a$, decreases with productivity from > 80 % in subtropical gyres to < 20 % in the NASPG, with a mean contribution of 65 % in oligotrophic areas. Although the background $b_{bp700}$ fraction cannot be entirely assigned to non-phytoplanktonic material, the biogeographic patterns found by Bellacicco et al. (2019) are hardly compatible with those found by Graff et al. (2015) or with a nearly constant detrital fraction across biomes. The detrital and non-phytoplanktonic POC fractions in PISCES near the sea surface range CE6, respectively, between 15 %–31 % and 42 %–59 % (annual mean values by biome; Table 3). Although they are within the full range of observations, these values suggest that PISCES underestimates the percentage of detrital POC, especially in oligotrophic waters. A deeper analysis, properly matching observations and simulations over time

and space, should be undertaken to obtain a mechanistic understanding of upper-ocean POC partitioning, with potential consequences for remotely sensed POC export estimates (e.g., Siegel et al., 2014).

PISCES bias may also arise from the inappropriate representation of some POC reservoirs, such as heterotrophic prokaryotes (bacteria and archaea, *BACT* in PISCES), which have long been recognized as important contributors to the suspended POC (Morel and Ahn, 1990; Gasol et al., 1997). However, *BACT* are not explicitly modeled in PISCES as prognostic tracers, meaning they are not interacting fully with other tracers. Instead, they are diagnosed in the productive surface layer from zooplankton biomass, based on an old model version that had interactive *BACT*. Below this layer, *BACT* biomass is propagated downwards with a power function based on Arístegui et al. (2009), which resembles a Martin curve (Martin et al., 1987), and is therefore very sensitive to the reference depth ($Z_0$) (Buesseler et al., 2020). We find two main arguments against the inclusion of PISCES-estimated *BACT* in our POC estimates with the current model configuration. First, the empirical *BACT* estimation in PISCES has not been validated, to our knowledge, and may introduce noise into the comparisons. Second, PISCES-simulated POC already includes heterotrophic prokaryotes because their biomass was not removed from the in situ POC measurements used to adjust the POC parameters in PISCESv2_RC (Aumont et al., 2017). In consequence, adding *BACT* to SPOC causes overestimation of mesopelagic POC (Figs. S14–S16) and can produce unrealistic temporal patterns (Fig. S15). Nevertheless, we believe that inclusion of prognostic bacteria would enable a more realistic simulation of POC stocks, with the potential side effect of improving the simulation of element fluxes in PISCES.

The comparison between simulated and observed LPOC is a novel contribution of our study. The method of Briggs et al. (2011, 2020), originally developed to study intense POC export events, was here extended to estimate SPOC and LPOC separately over the full annual cycle through the epi- and mesopelagic domains. Our analysis shows that, despite the mismatch in terms of concentration, the LPOC derived from the spikes of high-resolution bio-optical profiles is strongly correlated to the PISCES-simulated LPOC ($r = 0.78$, $r^2 = 0.61$) along a wide trophic gradient (Fig. 8). This result is encouraging and supports the more widespread deployment of instruments that perform high-resolution bio-optical sampling to shed light on the spatiotemporal dynamics of large aggregates and particles (Briggs et al., 2020; Lampitt et al., 1993; Stemmann et al., 2008). On the other hand, it is unclear to what extent the LPOC inferred from the $b_{bp700}$ spike signal is capturing mesozooplankton biomass, in addition to aggregates. The exclusion of PISCES mesozooplankton (*ZOO2*) from the comparison increases model–observation mismatch, with BGC-Argo LPOC exceeding PISCES estimates around 2-fold, although the high correlation remains ($r = 0.76$, $p < 10^{-10}$). Imaging devices

mounted on BGC-Argo floats may provide a more accurate quantification of LPOC, allowing for the separation of detrital LPOC (Trudnowska et al., 2021) from mesozooplankton and micronekton (Haëntjens et al., 2020) and the separation and quantification of particle classes contributing to flux across the complete particle spectrum (Bourne et al., 2021).

## 4.4 Importance of realistic physics and model evaluation across scales

In Fig. 5, PISCES simulations and BGC-Argo observations are compared using an array of skill metrics computed on the seasonal vertical profiles of TPOC between 0–1000 m. Starting with the 3D seasonal climatology, we observed that the correlation between the median (aggregated) profiles was generally better than the correlation between the spatially collocated (non-aggregated) profiles within a given biome and season (Fig. 5a), whereas no differences were found in terms of dispersion metrics (Fig. 5b and c). The difference in correlation was larger in subpolar biomes, suggesting that the model–observation spatial mismatch was magnified in regions with more energetic ocean dynamics and sharper physical and biogeochemical gradients, whose real-world location may not be well reproduced by the ocean circulation model used to force PISCES. For PISCES 1D simulations, their correlation coefficients with their CATS counterparts was usually in the high range of the correlation coefficients obtained by the biome-median PISCES 3D profiles. In terms of dispersion metrics, the ensemble of PISCES 1D simulations showed wider dispersion, but the best 1D simulations clearly outperformed the 3D simulation in a given region and season. The better skill of 1D simulations was more evident during spring, a season characterized by the onset of stable stratification after deep winter vertical mixing in middle and high latitudes. The greatest difference between 3D and 1D simulations was found in the Mediterranean, highlighting the more realistic vertical mixing and upper-ocean productivity in the 1D simulations.

Our cross-scale evaluation indicates how crucial it is to evaluate model physics before extracting conclusions on biogeochemical model performance (Doney et al., 2004; Kriest et al., 2020; Löptien and Dietze, 2019). In our 1D CATS approach, the skill of PISCES simulations was maximized by carefully matching observed and modeled vertical mixing (Figs. S1–S4), which is a key driver of upper-ocean ecosystems. This approach has a subjective component and may also suffer from the idealized assumption that BGC-Argo profiles reflect mostly vertical-scale processes, disregarding horizontal advection (Alonso-González et al., 2009). Yet, the similar misfit patterns encountered for different CATS within a given biome support the robustness of the 1D matching approach (compare Fig. 6 versus Fig. S6 and Fig. 7 versus Fig. S9). To further evaluate this issue, we matched different neighboring NEMO grid cells to the same in situ CATS. Again, this exercise indicated that our main conclu-

sions are not sensitive to the choices made for 1D model–observation matching (compare Figs. 6 and S5). Indeed, alternative matching approaches can be devised, each of which have advantages and pitfalls, for example (1) sampling the outputs of biogeochemical models at the locations visited by BGC-Argo floats, which may require high-resolution models; (2) deploying virtual BGC-Argo floats (van Sebille et al., 2018) and comparing them statistically to observations; or (3) forcing 1D biogeochemical simulations with observed physical fields, e.g., vertical mixing (Llort et al., 2015) or light (Terzić et al., 2019). As a general rule, the good skill of the best PISCES 1D simulations (Fig. 5) indicates that our CATS approach can be used to tease apart model–observation misfits caused by the physical and biogeochemical components, opening up new avenues for parameter optimization (Falls et al., 2022) and model development.

Our cross-scale evaluation is also informative as to the spatiotemporal scales that can be addressed with a given model setup. This matter was recently tackled by Bisson et al. (2019) using the export production model of Siegel et al. (2014), which is forced by satellite-derived primary production. In particular, they showed that this diagnostic model could be optimized to reproduce global climatological patterns but exhibited poor skill when faced with non-climatological datasets, which reflect local snapshots of ecosystem functioning. Model evaluation at climatological scales provides an incomplete picture, especially in productive regions where much POC export can take place during intense but short-lived events (Briggs et al., 2020). Such events are smoothed out when climatologies are computed, and their coherence with the physical forcing is lost. Our work shows that a prognostic model like PISCES can afford both event-scale and climatological scale predictions. This capability is important to test our process-level understanding, which underpins climate change projections.

## 4.5 Joint use of BGC-Argo and models for process-level understanding

Properly representing POC stocks is crucial for constraining epi- and mesopelagic carbon budgets and, ultimately, estimating the strength of the BCP and predicting its future evolution. The mismatch patterns between simulated and observed POC profiles (Figs. 4, 6, and 7) indicate different types of model shortcomings in subpolar and subtropical latitudes. The poorest agreement between PISCES and BGC-Argo data is found when their respective estimates of mesopelagic Teff are compared (Fig. 10), which should prompt further research on this key descriptor of the BCP, both in terms of POC fluxes (Buesseler et al., 2020) and stocks (Lam et al., 2011; this study) and their relationship with the structure and productivity of the upper-ocean ecosystem.

In the subpolar biomes, we find discrepancies between the patterns of SPOC and LPOC vertical export triggered by the

intense surface phytoplankton blooms typical of these waters (Figs. 6, S5–S7, and S10). The observed rapid SPOC increase at depth cannot be explained by the SPOC gravitational sinking, and fragmentation of rapidly sinking LPOC has to be invoked (Briggs et al., 2020). This fragmentation may be caused by zooplankton feeding (Mayor et al., 2020; Stemmann et al., 2004a, b; Stukel et al., 2019) and swimming (Goldthwait et al., 2004), combined with bacterial hydrolysis of aggregate-binding polymers (Arnosti et al., 2012; Baltar et al., 2010a), and turbulence at a high kinetic energy dissipation rate (Takeuchi et al., 2019). In relative terms, mesopelagic SPOC increases less strongly than LPOC during blooms (Fig. 6i), which is consistent with SPOC being a byproduct of the transformation of less abundant LPOC. Fragmentation processes may supply fresher SPOC to the mesopelagic, enhancing the coexistence of suspended particles with variable freshness (Alonso-Gonzalez et al., 2010; Aumont et al., 2017) and overall contributing to POC remineralization (Giering et al., 2014; Mayor et al., 2020).

In the subtropical gyres and the most oligotrophic Mediterranean waters, PISCES underestimates TPOC between 0–1000 m except for the deep Chl $a$ maximum, where it overestimates TPOC. The prominent deep POC maximum simulated by PISCES is generally not found in observations from the STG biome (Figs. 7 and S9), where deep Chl $a$ maxima generally reflect phytoplankton photo-acclimation, not enhanced phytoplankton biomass (Cornec et al., 2021). Thus, PISCES possibly overestimates the productivity of deep Chl $a$ maxima globally, indirectly causing stronger nutrient limitation at the surface. Between the deep Chl $a$ maximum and 200 m, POC pools decrease more steeply in PISCES than in BGC-Argo observations. Between 200 and 700 m, by contrast, simulated SPOC and LPOC Teff are only 10 % lower than observations, well within observation uncertainty. Thus, the mesopelagic SPOC deficit simulated by PISCES in the STG originates mostly through the insufficient vertical POC export at 200 m depth. The Mediterranean represents a different case, whereby the large disagreement between PISCES and BGC-Argo Teff may result from either poorly constrained POC / $b_{bp700}$, simulated POC dynamics, or both. Thus, this region may provide a good test bed for studying the role of the mineral fraction in the BCP, including the controversial ballast hypothesis (François et al., 2002; Klaas and Archer, 2002; Passow, 2004).

In summary, the mismatch with observations suggests the need to improve the representation of SPOC–LPOC interconversion in PISCES. The size distribution of POC along the vertical axis is a key variable for constraining POC budgets because it reflects the interplay between gravitational sinking, remineralization, trophic transfer, and 3D dynamics including horizontal POC advection (Alonso-González et al., 2009; Boyd et al., 2019). In many instances (Fig. 9), our analysis suggests that better model performance in the mesopelagic may be achieved by increasing the net transfer of LPOC to SPOC, e.g., through LPOC fragmentation, and

the Teff of both fractions. The vertical model–observation mismatch patterns observed here emphasize that POC budgets have to be computed with the highest vertical resolution affordable, or otherwise an apparent POC budget balance may result from compensating imbalances in different horizons (Giering et al., 2014; Marsay et al., 2015). The detailed level of information available from BGC-Argo floats may prove to be extremely valuable to help improve the POC schemes embedded in models such as PISCES.

## 5    Conclusions and outlook

In this study we compared globally distributed POC observations between 0–1000 m made by BGC-Argo floats to the predictions made by the PISCES model (PISCESv2_RC). A subset of BGC-Argo floats profiling at high vertical resolution enabled us to analyze small and large POC separately. The comparisons rely on a proposed new scheme for converting a bio-optical measurement ($b_{bp700}$) to POC. Although PISCES recreates the main features observed in subpolar and subtropical biomes with good skill, the comparison is still hampered by (1) spatial and temporal variability in POC / $b_{bp700}$ conversion factors, (2) mismatches in observed and simulated physics, and (3) imperfect correspondence between observed and simulated POC fractions. An evaluation of these uncertainties allowed us to detect limitations of the biogeochemical model parameterizations. Some limitations may be tackled by optimizing model parameters (e.g., particle sinking and remineralization rates), whereas others may require changes in model structure, for example the representation of zooplankton feeding on, and transformation of, mesopelagic particles (Mayor et al., 2020). The descriptive work and model–data matching strategies presented here pave the way towards the use of BGC-Argo observations for data assimilation, parameter optimization (Falls et al., 2022), and, ultimately, model development.

Widespread use of BGC-Argo data for understanding POC budgets and the BCP can complement classical model constraints based on vertical POC fluxes and ocean-interior nutrient remineralization. Merging of BGC-Argo and satellite data streams through data-driven approaches – which allow for great flexibility – and mechanistic models – which provide process-level understanding – can soon provide us with a high-resolution 4D view of the oceanic carbon cycle. Further work is granted to investigate POC dynamics through a combination of PISCES, autonomous observations, and ship-based observation programs (e.g., GEOTRACES; Lam et al., 2015) and data compilations (Mouw et al. ,2016; Evers-King et al., 2017).

Below we list several research priorities, whose implementation would advance the study of the biological carbon pump through the synergies between BGC-Argo observations and modeling.

- Observations

  - High-resolution bio-optical profiles are indispensable for process-level understanding. A combination of low- and high-resolution sampling in separate or individual floats (e.g., burst sampling) may provide a good compromise between float lifetime and observation capabilities.

  - More in situ measurements are needed to constrain POC estimation from bio-optical variables, especially in meso- and bathypelagic waters. The addition of transmissometers may enable more accurate POC quantification (Bishop, 1999) and particle characterization (Boss et al., 2015).

  - The inclusion of new types of sensors (Claustre et al., 2020) and the extension of measurements into the bathypelagic (Deep-Argo; Roemmich et al., 2019) hold high potential for advancing BCP research.

  - Further developments in BGC-Argo data processing are needed, with the final goal of supplying a wide public with user-friendly data products in near real time.

- Models

  - Evaluating model skill at resolving POC stocks, in addition to fluxes, is key to ensure that models reproduce observed fluxes for the right reasons.

  - Evaluation against globally consistent datasets is critical to avoid model overtuning towards small, sparse datasets, such as vertical POC fluxes.

  - Continuous development of schemes representing particle dynamics across the entire size spectrum is needed to constrain ecologically, climatically, and economically relevant element fluxes.

  - Extension of prognostic modeling of bio-optical properties (Dutkiewicz et al., 2019) into the meso- and bathypelagic layers would enable direct matching with measurements made from autonomous platforms, facilitating their assimilation by models.

- Joint planning of field observation and modeling projects, from their very conception and through their entire development, is key to fully exploit the capabilities of each approach.

## Appendix A:    Calculation of POC / $b_{bp700}$ ratios and related optical considerations

POC / $b_{bp700}$ ratios at the sea surface were obtained from the slope of the linear regression between POC and $b_{bp700}$ (Table A1). Our approach essentially follows the literature compilation made by Cetinić et al. (2012). The linear regressions generally yielded small and non-significant $y$ intercepts for sea-surface data. Therefore, we assumed the slopes

were equal to the POC / $b_{bp700}$ ratio. When the $b_{bp}$ measurements were made at another wavelength, we converted them to $b_{bp700}$ assuming an exponent $\eta = 0.41$ (Cetinić et al., 2012), according to the following equation:

$$b_{bp}(\lambda_1) = b_{bp}(\lambda_0) \cdot (\lambda_1/\lambda_0)^{\eta}. \qquad (A1)$$

For measurements taken at 555 nm, which was the wavelength used in two studies (Table A1), this resulted in 10 % lower $b_{bp700}$ compared to $b_{bp555}$ and thus higher POC / $b_{bp700}$ ratios.

The dataset of Cetinić et al. (2012) was the only one showing a significant negative intercept of the POC vs. $b_{bp700}$ linear regression. Unlike the other datasets, where the POC vs. $b_{bp700}$ relationship reflected mostly sea-surface variability in a given biome, this was the only dataset that included data collected between the sea surface and 600 m. Bol et al. (2018) reprocessed this dataset by computing the linear regression between POC and $b_{bp700}$ in different depth bins, forcing the intercept through zero. The slope of the POC vs. $b_{bp700}$ regressions, and therefore the POC / $b_{bp700}$ ratio, decreased more than 2-fold between the surface and the 600 m bins. These results suggest that the POC vs. $b_{bp700}$ relationship may be better modeled with nonlinear equations, as done for surface data in some previous studies (Balch et al., 2010; Johnson et al., 2017; Stramski et al., 1999). However, one must keep in mind that the ecosystem processes that define POC / $b_{bp700}$ ratios in the surface layer may be different from those occurring in the mesopelagic (see Sect. 4.2).

The relationship between POC and particulate beam attenuation at 660 nm, $c_p$, has been analyzed on more occasions than the POC / $b_{bp700}$ relationship. The backscattering ratio, $b_{bp}/b_p$, relates particulate backscattering to the total particulate scattering and is directly related to the refractive index of the particle assemblage (Babin et al., 2003; Dall'Olmo et al., 2009; Loisel et al., 2011; Organelli et al., 2018; Stramski and Kiefer, 1991; Stramski et al., 1999; Twardowski et al., 2001; Ulloa et al., 1994). Light absorption by particles is negligible in the 650–700 nm spectral region, such that total beam attenuation $c_p$ is a good approximation of the scattering coefficient $b_p$. Thus, once the known absorption and scattering coefficients of seawater are removed, $b_{bp}/c_p$ is a good approximation of the backscattering ratio and can be used to compare the relationships between POC vs. $c_p$ and POC vs. $b_{bp700}$.

The observed POC / $b_{bp700}$ variability in the surface layer across biomes, reflected in our POC estimation algorithm, seems analogous to that found for the relationship between POC and $c_p$ by Cetinić et al. (2012) (Fig. A1). The POC content per unit $b_{bp700}$ decreases with the maximum $b_{bp700}$ of each dataset, which may be related to the structure and species composition of upper-ocean ecosystems (see Sect. 4.3). It is also plausible that the fraction of POC (in terms of particle size or type) that contributes to backscatter-

ing varies across sites, further enhancing POC / $b_{bp700}$ variability.

The number of studies that tackled the interconversion between POC and bio-optical proxies in the mesopelagic layer is much smaller than those that focused on the epipelagic layer. Besides Cetinić et al. (2012) in the subpolar North Atlantic, we are only aware of the studies of Bishop (1999) and Bishop et al. (1999). The latter two studies found a POC vs. $c_p$ slope of around 16 mmol C m$^{-2}$ ($=$ mmol C m$^{-3}$ m). Given that modern transmissometers accept more forward scattered light that those used by Bishop (1999), the corresponding slope would be approximately 27 mmol C m$^{-2}$ (Jim K. B. Bishop, personal communication, 2021). Bishop (1999) found this slope to be nearly constant in contrasting areas over the 0–1000 m depth range, as depicted in Fig. A1 with the datasets labeled "Bishop1999eqpac" (Equatorial Pacific) and "Bishop1999all" (Equatorial Pacific, NE Pacific, and North Atlantic together).

The linear relationship between POC and $c_p$ is compatible with the nonlinear relationship between POC and $b_{bp700}$ along the vertical profile if the backscattering ratio also increases with depth, as found by Cetinić et al. (2012). The latter study found an increase in the backscattering ratio ($b_{bp700}/c_{p653}$) from around 1.2 % at the surface to around 1.5 % in the upper mesopelagic. The $b_{bp700}/c_{p653}$ ratio was more variable in deeper layers and values $> 2$ % were not rare (see also Nencioli et al., 2010; Organelli et al., 2018; Twardowski et al., 2001). The POC / $b_{bp700}$ ratio of 1000 mmol C m$^{-2}$ used here for deep waters (Eq. 1), based on the analysis of Bol et al. (2018), would correspond to a POC vs. $c_p$ slope of 27 mmol C m$^{-2}$ if the backscattering ratio was 2.7 % in the lower mesopelagic (600–1000 m, Fig. 2). Analogously, the range of POC / $b_{bp700}$ used in our sensitivity tests, 500 to 2000 mmol m$^{-2}$, would correspond to a backscattering ratio between 5.4 % (likely too high) and 1.35 % (closer to available estimates). More collocated measurements of POC, $b_{bp700}$, and $c_p$ in the mesopelagic and below are needed to reduce these uncertainties.

Beyond the natural variability, the interconversion between POC and bio-optical proxies is also confounded by methodological issues (Cetinić et al., 2012; Strubinger-Sandoval et al., 2021), most of which were not fully addressed in the studies compiled here. In particular, filtration of large sample volumes in the cubic meter range with in situ pumps yielded much stronger POC–$c_p$ relationships than small-volume sampling (Bishop, 1999).

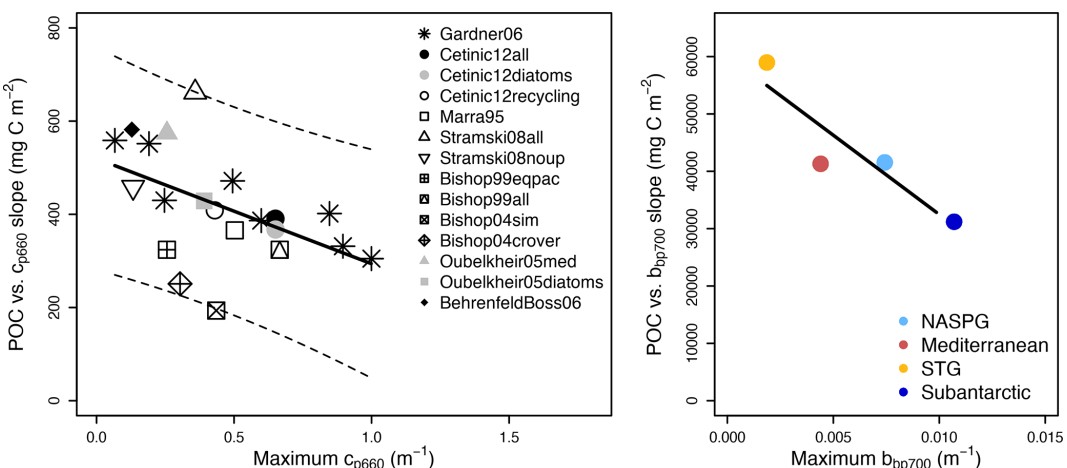

**Figure A1.** Coherent patterns in the conversion factors between POC and bio-optical variables. Left: relationship between the POC versus $c_p$ slope and the maximum $c_p$ in a given dataset, based on the Fig. 8 from Cetinić et al. (2012) with the addition of Bishop (1999) datasets. Right: analogous relationship for $b_{bp700}$, as used in our study. Linear fits are only illustrative.

**Table A1.** Compilation of studies that reported linear regressions between POC and $b_{bp700}$ in the euphotic layer of the oceans.

| Location | Reference | $N$ | Depth range | Slope (mg C m$^{-2}$ m) | Intercept | Comments |
|---|---|---|---|---|---|---|
| North Atlantic subpolar gyre | Cetinić et al. (2012) | 321 | 0–600 | $35\,422 \pm 1754$ | $-14.4 \pm 5.8$ | Downcast |
| | | 321 | 0–600 | $43\,317 \pm 2092$ | $-18.4 \pm 5.8$ | Upcast |
| | Bol et al., 2018 | NA | 0–10 | 41 550 | Forced through 0 | Subset of Cetinić et al. (2012) at the surface |
| Mediterranean | Loisel et al. (2001) | NA | NA | 41 305 | 1.43 | Original measurements at 555 nm |
| Subtropical and tropical Pacific and Atlantic (upwellings excluded) | Stramski et al. (2008) | 54 | 4–8 | 58 968 | 2.75 | Original measurements at 555 nm |
| Subantarctic | Johnson et al. (2017) | 67 | 0–100 | $31\,200 \pm 2470$ | $3.0 \pm 6.8$ | |

NA: not available.

*Code availability.* TS1 The model code used in this paper is available under https://doi.org/10.5281/zenodo.5243343 (Galí et al., 2021a). The authors can provide the code used to process the datasets on reasonable request.

*Data availability.* The simulated and observed datasets analyzed in this article are available at https://doi.org/10.5281/zenodo.5139602 (Galí et al., 2021b). The code and documentation of NEMO and PISCES are available at https://www.nemo-ocean.eu/ ( TS2 Madec and NEMO System Team, 2019; NEMO TOP Working Group, TS3 ).

*Supplement.* The supplement related to this article is available online at: https://doi.org/10.5194/bg-19-1-2022-supplement.

*Author contributions.* MG and RB designed the study. MF produced and/or reprocessed global climatological datasets. RB produced the NEMO dynamical fields used to force PISCES 1D offline simulations. OA provided the global PISCES simulation. MG processed BGC-Argo coherent annual time series, ran PISCES 1D simulations, analyzed data and produced the figures, and wrote the paper with contributions from all coauthors.

*Competing interests.* The contact author has declared that neither they nor their co-authors have any competing interests.

*Acknowledgements.* The authors acknowledge Antoine Poteau for guidance with BGC-Argo data; Margarida Samsó and Pierre-Antoine Bretonnière for downloading and storing the Argo data; and Thomas Arsouze, Vladimir Lapin, and Joan Llort for advice on the PISCES 1D configuration. Argo data were collected and made freely available by the International Argo Program, which is part of the Global Ocean Observing System, and the national programs that contribute to it. The authors thank Jim K. B. Bishop and an anonymous reviewer for their constructive criticisms that improved the paper.

*Financial support.* Martí Galí has received financial support through the Postdoctoral Junior Leader Fellowship Programme from "La Caixa" Banking Foundation (ORCAS project; LCF/BQ/PI18/11630009) and through the OPERA project funded by the Ministerio de Ciencia, Innovación y Universidades (PID2019-107952GA-I00). Raffaele Bernardello received support from the Ministerio de Ciencia, Innovación y Universidades as part of the DeCUSO project (CGL2017-84493-R).

*Review statement.* This paper was edited by Carolin Löscher and reviewed by Jim K. B. Bishop and one anonymous referee.

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

**Remarks from the language copy-editor**

**Remarks from the typesetter**