# Peer review of "Bridging the gaps between particulate backscattering measurements and modeled particulate organic carbon in the ocean"

_Biogeosciences, 2021_

## Author Comment (AC1)

**General response to reviewers**

We thank both reviewers (R1 and Jim Bishop) for their constructive criticisms and encouraging comments. Their recommendations will be useful to improve the article and make it a more valuable resource for both the modelling and observational communities.

Below we provide an overview of the main changes we propose:

- Revise the abstract to make it more informative and concise.
- Sharpen the writing of some sentences found to be cryptic by the reviewers, and provide more specific and quantitative information whenever possible.
- Add the references suggested by the reviewers, or cite them again in the place suggested by the reviewers if they were already cited in the previous manuscript. In a few cases we declined to add the citation and provided a reasoning.
- Add a table to define acronyms (in addition to in-text definitions).
- Change the acronyms of small and big POC (sPOC and bPOC, respectively) to SPOC and LPOC.
- We will
- Add brief pieces of text to document or discuss more in depth some aspects highlighted by R1 (see detailed responses), related to particle remineralization, fragmentation, and sinking, and their representation in models, as well as the role of chemolithotrophic microbes. Note, however, that in-depth discussion of POC cycling processes in the mesopelagic is beyond the scope of this paper. These relevant questions are being addressed by our group in ongoing studies and will be the matter of future papers.
- Refine the information on POC estimation from bio-optics (beam attenuation and backscattering) in the main text and Appendix based on the studies of Bishop and Wood (2008), Bishop 1999, and Boss et al. (2015).
- Improve the critical assessment of the correspondence between POC fractions simulated by PISCES and observed by backscattering sensors.
- Make the requested improvements to figures 4, 7 and 8.

In the following sections we provide our point-by-point responses, showing how we plan to address each of the referees' concerns. In most instances, we also proposed the revised text (with new text underlined). At the end, we listed the references cited here that were not in the original text. All referee comments are in blue, author responses are in black.

**Response to reviewer 1**

This study compares POC concentrations simulated by a biogeochemical model (PISCES) to high resolution field estimations from autonomous platforms and satellites. The authors show large discrepancies between models and observations. However, model and observations both agree on the behaviour of slow and fast sinking particles. The authors conclude that uncertainties in the POC conversion factors, imperfection between available observations and model points and in the representation of the physics in the model are sources of mismatch. While I feel like some of the processes invoked for explaining the discrepancies are not deeply discussed, I do not have any major issues with this work. I believe that some points should be clarified, better justified, and better documented before publication. These are listed below:

We thank R1 for his/her positive evaluation. In the new version, we will discuss or better document the aspects highlighted by R1, as described in detail below.

**Minor remarks :**

L24 : So all known factors driving sinking POC cycling in the mesopelagic basically...

Please see the response below.

L24-26: The abstract mentions that inconsistencies were identified. The rest of the sentence is somehow cryptic, and I feel like such inconsistencies should be detailed a bit more.

We will revise the abstract to address these comments. In particular, we propose to (i) present our novel approach and the results in a more logical order, (ii) provide quantitative results, and (iii) specify the major model-observations mismatch patterns, as well as the inconsistencies between BGC-Argo and satellite observations.

L46: None of these papers report actual measured flux attenuation, only (Marsay et al., 2015) does please revise.

Instead of these references, we will cite Mouw et al. 2(016), who presented the most up-to-date compilation of gravitational POC flux measurements in the global ocean, to our knowledge.

L53: And wide variations in the attached microbial communities (Baumas et al., 2021)

We will modify the sentence as follows: "Indeed, these particles also feature wide variations in shape, density, chemical composition and reactivity towards microbial degradation (Kharbush et al., 2020), as well as wide variation in the attached microbial communities (Baumas et al., 2021; Duret et al., 2019; Mestre et al., 2018)."

We propose to add citations to the articles of Mestre et al. (2018) and Duret et al. (2019), which preceded Baumas et al. (2021) and give complementary views.

L55: Rephrase to "large" instead of big, here and throughout

We will reword throughout.

L56: This is not always true, please see work by (McDonnell M., P. & Buesseler, 2010)

We thank the reviewer for calling our attention to this interesting reference, as well as Laurenceau-Cornec et al. (2019) (comment on L569). However, in this particular sentence of the Introduction we suggest to cite Cael et al. (2021), who provided the most comprehensive intercomparison to date, to our knowledge, of particle size vs. sinking speed relationships measured with a wide diversity of devices/methods and in various ocean settings. The study of McDonnell and Buesseler (2010) represents a particular region and season studied with a method (drifting sediment traps) that may select for/against certain particle types, as the authors acknowledge.

We propose to modify the sentence as follows:

"Gravitational sinking speed generally increases with particle size, although observations show a wide scatter around, and deviations from, canonical Stokes' law predictions (Laurenceau-Cornec et al., 2019; Cael et al., 2021). Owing to this general relationship, particle populations are often partitioned into a few functional size classes...".

L62: Please include results by (Alonso-Gonzalez J. et al., 2010; Riley et al., 2012).

We will add these citations here. Please note, however, that Alonso-Gonález (2010) was already cited (eg, line 72, 682). Instead of Riley et al. (2012) we had cited Baker et al. (2017), because the latter study expanded Riley's one using a similar methodology (Marine Snow Catcher sampling).

L89: How was this represented in previous versions of the model?

In previous versions and up to PISCESv2 (Aumont et al., 2015), the default model configuration had two POC size classes, small and large, that sank at different speeds. Both small and large POC were remineralized with a fixed rate constant at a given temperature. An alternative configuration was also available with PISCESv2, which used the parameterization of Kriest and Evans (1999). Despite resolving the particles' size and sinking speed spectra, the latter parameterization was not sufficient

to reproduce observed POC concentration profiles, likely because slow-sinking and suspended particles were remineralized too rapidly through the entire water column.

See also the responses to reviewer 1 comments' on L236 and L239, as well as the response to reviewer 2, L222.

L92: which ones? from what methods? You may want to state the pitfalls of such methods to connect with the subsequent part of the text.

We propose to rewrite as follows (new text underlined): "Despite this breakthrough in the representation of POC fractions in PISCESv2\_RC (hereafter "PISCES"), the new parameterization was evaluated using only sparse measurements (Druffel et al., 1992; Lam et al., 2011; 2015) based on large-volume filtration with in-situ pumps. Although this method enables accurate determination of the mass and composition of the particulate fraction, bar fast-swimming zooplankton (Bishop and Wood, 2008), it cannot afford high-resolution sampling".

L95. Please state the size range of particles observed with such devices.

Will do. Also, please refer to Table 1.

L103 : surface POC flux or concentration ?

Concentration.

**L106 : Which of the « drivers » are specifically being evaluated here?**

The processes that control POC cycling ("drivers") are not specifically evaluated in this paper, so we propose to remove the sentence enclosed in commas in L106.

**L236: what is this based on? justify**

In most biogeochemical models, POC remineralization follows first-order reaction kinetics, with POC remineralization rate constants typically ranging between 0.02–0.06 d-1 at 0 degrees (Laufkötter et al., 2016). This "k0" is then modulated as a power-law function of temperature in most models. In PISCES\_v2,  $k(T) = k0 \cdot (1.064)^T$ , resulting in a Q10 of 1.9. That is, remineralization rate constants for "fresh" POC (see response below, comment on L239) approximately double every 10 degrees, which is in reasonable agreement with experimental estimates (see for example the data compiled by Belcher et al., 2016; section 4.3; note the data are reported at in situ temperature).

The exact values used in each model result from the history of model development, usually based on 1D model configurations (see for example the classical model of Fasham et al., 1990), and the global-scale tuning applied in order to balance export production and remineralization while simulating realistic oxygen and nutrient fields.

**L238: More strongly than what?**

More strongly than that of slow-sinking particles. Fixed.

L239: (Briggs et al., 2020) recently found that fragmentation could be responsible for up to half what's observed in flux attenuation. How does the "small" fragmentation may compare to (Briggs et al., 2020) here.

We thank the reviewer for suggesting more specific wording here, because the expression "small fraction", which we used to avoid excessive detail, was in fact inappropriate. We propose to rephrase the last part of the paragraph as follows (new text underlined):

"Remineralization follows first-order kinetics with an initial specific rate "k" of 0.035 d-1 (at 0°C) for freshly produced detritus in the upper mixed layer. This "k" decreases with depth as an emergent result of the RC parameterization. To account for bacterial solubilization of aggregate-binding polymers, 10% of the degraded large detritus are routed to small detritus (this fraction is hard-coded based on previous calculations). The flux feeding rate depends on the particles' sinking flux, and thus attenuates the flux of large particles more strongly than that of small particles. A variable fraction of the large detritus intercepted by flux feeders is not assimilated but fragmented into small detritus, attenuating up to around 50% of the large detritus sinking flux through the top 500 m during intense export events."

To provide further context: in PISCES, the fraction of the large POC vertical flux that is fragmented into small POC is not constant, and results from two main biological processes: bacterial GOC solubilization and mesozooplankton flux feeding on the "rain" of aggregates (large POC). Physical disaggregation of large POC into small POC is not considered.

Fragmentation by zooplankton increases with the proportion of mesozooplankton that are flux feeders and with the proportion of phytodetritus aggregates within the large POC. The latter two variables are computed internally in the model. The proportion of mesozooplankton flux feeders depends on the proportion of food available from sinking POC interception vs. that ingested through active suspension feeding. The proportion of large POC in the form of phytodetritus aggregates increases with diatom mortality upon nutrient limitation (the remaining large POC arising mostly from fecal pellets). In consequence, the role of fragmentation is expected to be most important during strong export events following the termination of diatom blooms.

Our estimates indicate that, in PISCES, mesozooplankton fragmentation removes around 50% of the sinking flux of large POC through the upper mesopelagic (100-500 m) in productive ecosystems (e.g., the subpolar North Atlantic). This figure is in good agreement with the findings of Briggs et al. (2020). In subtropical gyres, the relative flux attenuation due to fragmentation is smaller, partly because a larger proportion of the sinking flux is carried by small POC, which is not susceptible to fragmentation. Solubilization represents a more constant background process and becomes relatively more important at low latitudes. We refer the reviewer to the presentation we made at EGU 2020, available here

https://presentations.copernicus.org/EGU2020/EGU2020-16602 presentation.pdf (slides 16–18).

**L280: Any error associated with such climatologies (PISCES results excluded)?**

Estimation of the uncertainty associated with these datasets is an arduous task, and measurement errors may easily be confounded with natural variability. Unfortunately, measurement uncertainties were not provided with these observational datasets on a grid cell basis. Work is ongoing to include space- and time-resolved uncertainty estimates in future versions of observational datasets (for example, ESA's ocean colour climate change initiative), which will enable more accurate calculation of model skill metrics taking into account those uncertainties. In the case of our POC climatology derived from BGC-Argo bbp700, our publicly available datasets include not only the means and the medians but the range and N of measurements in each grid cell, which can be used to some extent to gauge the uncertainty in climatological data. Recent assessments of the intercomparability between bio-optical measurements of POC and other quantities have been made by Lombard et al. (2019), Giering et al. (2020) and Liu et al. (2021).

We would like to note that, sometimes, the nominal uncertainties are smaller than the real uncertainties when sensors or retrieval algorithms are faced with a wider range of conditions. This seems to be the case, for example, with the Stramski et al. (2008) satellite POC algorithm, which our study found to severely underestimate POC during the high-latitude winter. A recent evaluation

**L307: Please specify, what are the physical processes that are unrealistically represented in this region?**

There are no specific processes that are not correctly represented per se in the model. The main limitation in that area (and in most coastal areas and semi-enclosed seas) is the resolution of the model. It is roughly equivalent to about 1° resolution which is very coarse. Several studies have

shown that a proper representation of ocean dynamics in the med sea requires a resolution of at least 1/12° (Lebeaupin Brossier et al., 2011; Tonani et al., 2008) corresponding roughly to the first internal Rossby radius of deformation. As a consequence, our achieved resolution in that sea is insufficient to correctly simulate ocean dynamics there. Therefore, ocean biogeochemistry should be quite crudely simulated.

We will add the two references cited above in the new version.

**L 369: what do mean by rare aggregates?**

"Rare" is used here meaning that "aggregates are found at low concentrations". "Rare" is usually defined as "common or frequent; very unusual" (Cambridge), or "seldom occurring or found" (Merriam-Webster).

L450 See work by adam Maritny'group https://www.nature.com/articles/sdata201448 This presents of substantial amount of data including mesopelagic data that could be potentially used here.

Thanks for the reference. We will use this dataset in ongoing research and subsequent publications.

L490: Add (Bianchi et al., 2018)

**Thanks, will do.**

L530: does this translate into an increase in sinking speed (for a given size) in your model?

**See response to comment on L670**

L569: Size vs sinking velocities relationships depends mostly on their composition (Laurenceau-Cornec et al., 2019), some large unballasted aggregates will not sink as fast as Stokes predicts.

Please see the response on L56. We will add the citation and mention exceptions to the general assumption.

**L585: What about chemoautolithotrophy (CO2 dark fixation by particles associated bacterias) (Herndl & Reinthaler, 2013)**

We are well aware of the potential importance of chemoautolithotrophy, as shown in lines 491-492 and references cited. We will add the suggested reference and indicate that chemoautolithotrophy can alter the proportion of autotrophs/heterotrophs.

**LL595-608: Silly question, why are BACT not explicitly represented in such models?**

Most global biogeochemical models do not explicitly represent carbon oxidizing bacteria. Instead, they simulate degradation of organic matter according to rather simple and quite crude parameterizations based on a first order kinetics with a remineralization constant which may depend on temperature and some other factors (such as lability). We are not aware of a systematic justification of such an approach apart that historically it has been the first to be used. Some models do include free living carbon oxidizing bacteria which mainly consume labile DOM and hydrolyse semi-labile DOM. This increases quite significantly the complexity of the models which means that they are more expensive to run. Yet, to our knowledge, the skill of such detailed representation with respect to simpler parameterizations has not been documented in global scale ocean biogeochemical models. No models do include an explicit representation of particle attached bacteria, except to our knowledge the model of Anderson and Tang (2010). Probable reasons for that are: the complex structure of marine particles and their colonization by bacteria, the specificity of the ecosystem in the particle microenvironment, exchanges between the particles and the surrounding sea water, ...

L620: Down to what particle size?

Please see Table 1.

**L670: How are particles sinking velocities represented in PISCES? Do they increase with depth as observed in (Villa Alfageme et al., 2016)?**

Depends on the model configuration. PISCES does not include an explicit mineral ballast mechanism, as this did not bring clear improvement over other parameterizations. However, it is possible to prescribe an increase in LPOC (bPOC) sinking speed between the base of the mixed layer and 5000 m (Aumont et al., 2015). Following Aumont et al. (2017), we prescribed a constant LPOC sinking speed of 50 m/d. Evaluation of LPOC sinking speeds is beyond the scope of this study. Yet, in the companion paper (https://gmd.copernicus.org/preprints/gmd-2021-222/) we did evaluate the feasibility of optimizing LPOC sinking speed in PISCES using BGC-Argo observations in the mesopelagic. The results indicate that observations in the mesopelagic layer are not sufficient to adjust LPOC sinking speed in PISCES, likely because a wider depth range (with wider variation in sinking speeds) is needed.

L676: Is fragmentation caused only by biological factors or could non-biological factors such as microturbulence play a role? Please discuss.

We propose to amend as follows:

"This fragmentation may be caused by zooplankton feeding (Mayor et al., 2020; Stemmann et al., 2004a, 2004b; Stukel et al., 2019) and swimming (Goldthwait et al., 2004), bacterial hydrolysis of aggregate-binding polymers (Arnosti et al., 2012; Baltar et al., 2010a), and turbulence at high kinetic energy dissipation rate (Takeuchi et al., 2019)."

The study of Takeuchi et al. (2019) showed that turbulence results in new aggregate formation until a "critical turbulent kinetic energy dissipation rate of  $10^{-6}$  (W kg-1), above which the smallest turbulent eddies limit aggregate size".

**L704, please explicit**

We propose to rephrase as follows (new text underlined):

"In many instances (Fig. 9), our analysis suggests that better model performance in the mesopelagic may be achieved by increasing the net transfer of LPOC to SPOC, e.g. through LPOC fragmentation, and the Teff of both fractions"

and to remove the last part of the sentence

", which may require further balancing the interplay between various mechanisms of POC export and flux attenuation."

L717: cryptic sentence, please revise.

We propose to replace this part of the sentence:

"which may arise from both suboptimal model parameters and model structure."

with the following one:

"Some limitations may be tackled by optimizing model parameters (e.g., particle sinking and remineralization rates), whereas others may require changes in model structure (i.e. equations), for example the representation of zooplankton feeding on, and transformation of, mesopelagic particles (Mayor et al., 2020).

Figure 6: This clearly show that at high latitudes PISCES produces a pool of small slow sinking particles that doesn't seem to be remineralised quickly enough as they do not appear on the sPOC BGC-ARGO profile. The occurrence of sPOC seen from the BGC-ARGO profile could well be from b-POC fragmentation into s-POC rather than from the original s-POC fraction sinking out on its own.

Thanks, we agree with R1, as shown by L701-704.

**Response to reviewer 2 (Jim Bishop)**

**Overview**. The authors compare PISCES model simulations and observed in-situ proxies for particulate organic carbon (POC) concentration in small (sPOC) and large size (bPOC) particle fractions in the 0-200 m epipelagic and 200m-1000m mesopelagic zone at several biomes in the world oceans. The observed in-situ sPOC and bPOC are derived from backscattering sensor data from biogeochemical ARGO floats (BGC-ARGO). bbp700 data are filtered to remove spikes – yielding a background profile representative of the small particle fraction (sbbp) and large particle spike anomaly (bbbp). The resultant sbbp data is scaled by a formula that includes a biome dependent multiplier and a depth decreasing ratio of POC/scattering with an assumed asymptotic value. The bPOC data are calculated from the anomaly contributed to by spikes and the same conversion factor is applied.

To my knowledge, this is the first successful attempt to reproduce subsurface particle concentrations in a model. This paper is very well written and logical in its development and summarizes areas where model and observations agree well and where there is disagreement. It highlights the value of optical sensors on floats. In the conclusions, there are concrete recommendations for added observations that will enhance insight into model parameterizations.

This is perhaps the most informative paper I have reviewed. The authors did a great job. The length of the following text reflects my excitement about the product. The recommendation of this reviewer is to publish after revision and a quick rereview. There is some more work with referencing. This should take little time.

There are some areas that could be clarified. ACRONYMS often appear without definition. Please add a table to ACRONYMS and clarify important ones in the text. Furthermore the representation of stocks (units mmol POC m-2) in a 200 m epipelagic layer and 800 m thick mesopelagic layer in the figures confuses the presentation of the paper. Please use mmol POC m-3. Another point of confusion is the use of bPOC to describe the large particle size fraction. Most of the particle literature uses "S" small and "L" large; it would be helpful to use this terminology. In the conclusions it might be important to mention that the international program GEOTRACES will produce comprehensive information on size fractionated particle chemistry (after Lam et al. 2011 - 2015), many of these data sets are growing in availability. Finally, as summarized, there are a number of float based optical systems that could be productively co-deployed within the BIO-ARGO framework that would illuminate particle flux processes. I think the flux float described by Bourne et al. 2021 (Biogeosciences, 18, 3053–3086, 2021) is a worthy addition to the list, such instruments were envisioned by Claustre et al. (2009) and will be ready. I also think the community should convene a review of the developing suite of sensors that could contribute to an evolved BIO-ARGO that answers key questions regarding modelling needs.

Reference to observations of Bishop and Wood (2009, in the southern ocean) using floats is relevant. They talked about varied criteria for MLD as well as transient stratification, and the concept of burst sampling. They deployed transmission and scattering sensors and had metrics for export and timing for exported material to reach 800m. (doi:10.1029/2008GB003206).

We thank Jim Bishop for his very encouraging comments, and appreciate his suggestions regarding relevant references that we had missed. In particular, we realized that Bishop (1999) and Bishop and Wood (2008) had addressed several questions that are highly relevant to our study, such as the types of particles causing spikes in different optical sensors, or the different type of relationship between POC and cp (linear) or bbp700 (nonlinear) along vertical profiles.

**Detailed comments.**

L8: Not sure where the 4Pg C number comes from. Needs a reference. Most of the numbers in the text are in the range of 0.5-2. You could say is dwarfed by the 1000 Pg C in DIC pool (above 100 m). Or is this an estimate for the entire water column.

The 4 Pg C is the PISCES estimate for the entire water column (Table 2), which appears reasonable to us because PISCES output compares well to observations (Aumont et al., 2017; our study). We will specify this information to avoid confusion. In addition, we will consider mentioning in the Discussion (e.g. 4.1) that the global POC stock is dwarfed by the DIC pool, as suggested by the reviewer, while stressing that the turnover of POC is much faster than that of DIC.

**L11-12: 80-90% Reference? Lam et al. 2011?**

The 80-90% range approximately brackets the majority of observed and modelled estimates provided in our study (median 85%); L565), and is also central amongst other POC datasets obtained with different fractionation approaches. For example, marine snow catcher measurements (Riley et al., 2012; Baker et al., 2017; García-Martín et al., 2019) tend to give SPOC fractions >90% (L569-571). Fractions based on filtration with in situ pumps (MULVFS; Lam et al., 2011; Lam et al., 2015) and particle collection on filters (cutoff usually at 53  $\mu$ m) give a wider SPOC/TPOC range of around 50% to virtually 100%, with most measurements clustering around 80% (see figure 5 in Aumont et al., 2017). Given that references are not allowed in the abstract unless strictly necessary, we propose to mention again the 80-90% range in the third paragraph of the Introduction (L55) and provide the same supporting references cited here.

**L13: Define ACRONYM PISCES.**

Pelagic Interactions Scheme for Carbon and Ecosystem Studies model. Defined at L8 in the first version of the manuscript.

L16: bPOC is not defined. Above... most literature refers to "L" POC (large).

We will add the definition of bPOC here if necessary (note that its first occurrence is in the abstract, L11), and change the acronym to LPOC. This and other definitions will be added to a table of acronyms.

L51-54: Stemmann and Boss (2012), Cael and White, 2020; Stemmann et al., 2004a), particle populations are usually partitioned into a few.... Other references? Certainly this has been described in literature back to the 1970's.

We thank the reviewer for this comment and will add citations to will add more references: Mullin (1965) and Bishop et al. (1980).

L70: Small POC can drive vertical POC fluxes across the mesopelagic layer. Needs better wording of "Small POC can drive..." Small POC does not drive... but convective mixing or subduction of small POC can transport POC below the euphotic layer (or epi pelagic layer) into the mesopelagic layer, adding to POC export...

We thank the reviewer for this correction and propose to rephrase following his suggestion:

"Convective mixing (Bishop et al., 1986; Dall'Olmo and Mork, 2014; Lacour et al., 2019) and subduction (Llort et al., 2018; Omand et al., 2015; Resplandy et al., 2019) can transport SPOC into the mesopelagic layer, adding to other export mechanisms and potentially making large contributions to total POC export (Alonso-Gonzalez et al., 2010; Henson et al., 2015)".

L 132: a fluorometer... optical backscattering. Name manufacturer. Use specific language.

We will add the following information: "The selected floats were equipped with a Seabird-Wetlabs ECO-Triplet sensor package including a Chla fluorometer (excitation at 470 nm; emission at 695 nm), and a backscattering sensor at 700 nm."

not azL136: ... the latter were flagged... need more information... Provide an example how data were flagged.

All BGC-Argo data undergo near real-time quality control at the data assembly center (DAC) level. This procedure includes a total of 30 tests on different sensors, such that each measurement is finally flagged with numbers ranging from 4 ("bad data") to 1 ("good data"). In the case of bbp700,

the only postprocessing applied at the DAC level (besides application of calibration factors to raw digital counts) is the removal of negative spikes. Therefore, the entire bbp700 profiles typically get QC flags 2 or 3 depending on the DAC, and data that look perfectly good may routinely get a "3" until further delayed-mode processing can be applied. That's why we specified in L137 that "we used all bbp700 measurements with quality control flag  $\leq$  3 (equivalent results were obtained with flag  $\leq$  2)". Additionally, all bbp time series used in this study were visually checked which confirmed that no additional treatment or profile removal was required.

The case of bbp700 is different from that of chlorophyll a fluorescence, for example, which is further post-processed according to a protocol all DACs have agreed on, which includes application of a constant correction factor and further correction for non-photochemical quenching (NPQ), FDOM interference, bottom-of-profile signal, etc. (Schmechtig et al., 2018).

In summary, rather than providing too many details in the main text, we prefer to direct readers to the relevant documents already cited in the text (Schmechtig et al., 2016 and 2018).

**L141: Define ACRONYMS**

L149: NAOS, remOcean and other ACRONYMS should be tabulated.

We will add a table with acronyms.

L156: Reference needed...  $\sigma\theta$  exceeded the surface reference value by 0.03 kg m-3. Bishop and Wood (2009) seems a good one.

We will add citations to Bishop and Wood (2009) and Sallée et al. (2021) at the end of L156 to support our choice, followed by the following sentence: "The 0.03 kg m-3 criterion provided sensible results across biomes and was consistent with the NEMO-simulated turbocline depth."

Note that in L215 we had already given some justification for using this MLD criterion:

"Although MIMOC is based on an algorithm that evaluates several MLD criteria, it has been shown to be globally consistent with the MLD based on a 0.03 kg m-3  $\sigma\theta$  threshold (Holte and Talley, 2009; Sallée et al., 2021). For the float time series, we used the MLD defined by a 0.03 kg m-3 threshold computed for each individual profile."

As done by Bishop and Wood (2009), we tested several MLD criteria (explained in L157; Fig. S1), which can capture vertical mixing on different time scales, and with different skill depending on the strength of thermal and haline stratification in a given region/season.

L 164: "The baseline and spike signals were converted to sPOC and bPOC...". Are there observations that confirm this correspondence? Please calculate the volume of water seen by the scattering sensor and the likelihood of the sensor seeing large particles. The only study I am aware of is Bishop and Wood (2008 Deep-Sea Research I55, 1684–1706) who concluded: "Those [referring to spikes] of the scattering sensor mostly appear to reflect abundances of larger and chain forming phytoplankton, protist grazers and possibly large aggregates. Nearly transparent particles like marine snow are not readily detected in transmitted light but are easily detected using reflected light. "

We propose to add the following specification in L162 (new text underlined):

"Each spike reflects the passage of a particle (organism or aggregate) larger than about 100  $\mu$ m in front of the sensor window. Previous studies inferred that backscattering spikes arise mostly from phytodetritus aggregates, but also from large zooplankton and phytoplankton (Bishop and Wood, 2008; Briggs et al., 2011; Gardner et al., 2000). Assuming that backscattering sensors sample a volume of 10 mL (Briggs et al., 2020), we estimated that backscattering spikes concentration was typically between a few and <100 L-1, consistent with other independent estimates of large particle concentration in the open ocean (McDonnell and Buesseler, 2010; Stemmann et al., 2008; Stemmann and Boss, 2012). Backscattering spikes were on average 4–10 times more abundant than

chlorophyll fluorescence spikes. The  $b_{bp700}$  spikes larger than 0.008 m-1, associated with particles larger than around 2 mm (very large aggregates, vertically migrating zooplankton) were removed, with a negligible impact on the total spike signal (Briggs et al., 2020)".

To justify this added text, we provide an extended argumentation herewith. The processing of raw backscattering profiles to separate the baseline and spike signals was addressed in detail by Briggs et al. (2020) (see the Supplementary Materials). The processing applied in our study followed in most aspects their approach (L150-165), and was restricted to the same subset of floats sampling at high vertical resolution (L150).

As described by Briggs and coauthors, the signal processing applied "removes positive spikes due to large individual particles that are rare relative to the sample volume of ~10 ml (taking into account ~10 cm of movement during the sensors' 1 s integration time)". They also highlighted that "high vertical resolution (~1 m or higher) measurements are required in order to obtain useful, depth-resolved concentrations from a single profile".

Particles larger than 100  $\mu$ m, the approximate minimum size of particles causing spikes in our approach, are usually found at low concentrations in the top 1000 m of the ocean, typically between a few and <100 particles/L (e.g., see total particle counts or particle size spectra in Stemmann and Boss, 2012, figure 2; Stemmann et al., 2008, figure 5 and 6; McDonnell & Buesseler, 2010, figure 3; McKenzie et al., 2020). Smaller particles occur at much greater abundance owing to power-law (log-log) slopes of the particle size distribution typically ranging between -4 and -3 . Therefore, the likelihood of detecting aggregates when sampling a volume of 10 mL is low, considering that the concentration of large particles is usually <1 particle/mL. Given that we are binning the data into depth intervals with no less than discrete 10 measurements per bin (and around 20 on average), the minimum water volume sampled in our approach is of around 100 mL per bin (but usually 200 mL). Theoretically, this should allow for the detection of aggregates in the 10/L concentration range.

Fortunately, our processing pipeline included the calculation of the mean spike frequency in each depth bin. Dividing it by the mean sample volume (10 mL = 0.01 L) gives the mean spike concentration. The table below summarizes the mean spike concentration (#/L) for four representative floats that sampled over >2 years, identified by their WMO number and biome, for depths >50 m (this is to avoid the smoothing effect of chlorophyll fluorescence correction on the profile, which effectively removes chlorophyll spikes; see also Briggs et al., 2011). The absolute range of bbp700 spikes concentration in these floats was 7–108 spikes/L.

| Variable                         | 6901486
(NASPG) | 6901579
(Subantarctic) | 6901472
(SPG) | 6901491
(Mediterranean) |
|----------------------------------|--------------------|---------------------------|------------------|----------------------------|
| bbp700 spikes, # L -1 | 59±9               | 63±11                     | 55±8             | 61±7                       |
| chl spikes, # L -1    | 15±19              | 16±24                     | 8±12             | 7±13                       |

This analysis indicates that:

- bbp700 spikes are more abundant and less variable than chl spikes, which can be expected given that chl spikes only detect large living cells or aggregates, whereas bbp700 can also detect living zooplankton and their fecal pellets. Differences in the vertical-temporal distribution of bbp700 and fluorescence spikes were already noted by Briggs et al., 2011.
- chl spikes are relatively more abundant in the more productive subpolar biomes, consistent with the notion of a larger proportion of the export production channelled by phytodetritus aggregates in productive biomes.

- inclusion of deep chl maxima (and/or bbp700 maxima), located well-below 50 m in the SPG and Mediterranean biomes, may explain the relatively high abundance of large particles in those biomes compared to subpolar ones (as found at ALOHA station by Bishop and Wood, 2008) despite their lower productivity.

Visual examination indicates that vertically binned bbp700 spike profiles are noisy but usually have defined shapes, i.e., aggregates do not occur randomly through the water column, especially in productive biomes (Fig. 6). Such features allow for the detection of export events from temporal sequences of profiles (Briggs et al. 2011, 2020). Indeed, further binning may be needed to reduce noise and obtain more accurate quantitative estimates of large POC concentration. This is precisely what we attempted by integrating the data between the 0-200 m (epipelagic) and 200-1000 m (mesopelagic) depth horizons, obtaining high correlation with PISCES-modelled LPOC (Fig. 8). Taking into account the vertical resolution of the profiles (L152), total epi- and mesopelagic sample volumes are around 3L and 2L, respectively.

Finally, we agree with the reviewer that the nature of the particles causing spike events cannot be deduced solely from their backscattering at 700 nm. Thus, we also explored whether additional information could be obtained by computing the fraction of bbp700 spikes that co-occurred with chlorophyll fluorescence spikes. Our analysis indicated that bbp700 spikes co-occurring with fluorescence spikes accounted for between 15% (subtropical gyres) and 45% (subpolar biomes) of the large POC mass. This estimate is only a lower bound because the backscattering and chlorophyll fluorescence sensors do not sense exactly the same volume of water. In fact, most bbp770 spikes did not co-occur with fluorescence ones, meaning that co-occurring spikes were larger than the average spikes. It is logical to hypothesize that co-occurring bbp700 and fluorescence spikes are more likely when aggregates are larger. In summary, we estimate that a sizable fraction of the LPOC, of at least 15-45% depending on the biomes and the seasons, may have corresponded to phytodetritus aggregates in our study.

Although these calculations are admittedly rough, our conclusions are fully compatible with the inferences made by Bishop and Wood (2008). Note, however, that the Seapoint turbidity sensor used by Bishop and Wood (2008) was different to the Seabird-Wetlabs ECO-Triplet sensors mounted on BGC-Argo floats. In particular, the Seapoint sensor measured scattering at a broad angle between 45-135° (880 nm wavelength), whereas ECO-Triplet sensors measure bbp (700 nm wavelength) at 124°, with full width at half maximum of 20° to 30°. Cetinic et al. (2012) showed that these measurements are not equivalent (see their Appendix, figure A1).

**Line 181. Reference to Fig. 2. Should not the green dotted line in the deep mixed layer reflect the average of POC/bbp over the interval, not the value at its deepest limit? Please explain more clearly why or why not.**

The algorithm we used to estimate POC/bbp700 is correctly represented in Fig. 2. The modulation of POC/bbp700 was already critically assessed in L623-630 of the Discussion, but we admit that the effect of diluting mixed-layer waters with waters from below would be more correctly represented by taking the average across the entire depth interval homogenized by mixing. However, this solution does not come without problems. We propose to rewrite as follows to incorporate this criticism, providing also a justification of our approach:

"Modulation of POC/bbp700 by vertical mixing improves the agreement between PISCES and BGC-Argo data in regions with wide seasonal MLD amplitude such as the NASPG. Still, our approach should be seen as a simplistic first-order approximation, and alternative formulations should be further evaluated when more data become available. In the mixed layer, for example, POC/bbp700 might be kept constant regardless of the MLD, or estimated for each profile as the average of the pre-computed POC/bbp700 profile (eq. 1) between the sea surface and the observed MLD. In both cases, however, POC/bbp700 would have to decrease abruptly below the mixed layer to meet the

low POC/bbp700 in the mesopelagic, producing sharp discontinuities for which observations are lacking."

Unfortunately, we are not aware of studies that measured the change of POC/bbp700 during convective mixing.

L 199. Equation 2. zref = max ("X","Y"). Reads like computer code . Try to clarify in the text what is meant.

**Will fix.**

L 222. Please explain briefly terms like "RC parameterization".

We propose to expand a little bit the description of the RC parameterization in the Introduction, rather than here. We propose the following new text in L84:

"Aumont's work also showed that the model's fit to observed deep-ocean POC concentrations could be dramatically improved by treating detrital POC, both small and large, as a mixture of particles with different reactivity (or lability) towards bacterial degradation. In this scheme, termed the reactivity continuum (RC) parameterization, detrital POC degradation is computing after dividing it into many reactivity classes that approximately follow a continuous gamma distribution –hence its name. The most labile fractions are rapidly consumed below the upper mixed layer, such that vertically exported POC becomes progressively more refractory. This results in enhanced preservation of SPOC in the model and a much more realistic fraction of SPOC with respect to total POC (TPOC) in the ocean interior. In addition, the RC scheme does not appreciably degrade model estimates of the gravitational POC flux."

Please define all ACRONYMS somewhere.

Will do.

L310. Fig. 4. Add log tick marks. Rather than regular grid on x axis. Otherwise... nice figure.

Thanks. Will do.

L349. Fig. 6 (and corresponding similar figures). When printed. The various blue/cyan lines in panel (a) and (d) are completely lost.

We will improve the figure to make these lines more visible.

L440-441. ... "Bishop et al. (1999) suggest a plausible range of 815–1630 mmol C m-2 (Appendix A)".

In the Appendix, the authors overlooked Bishop (1999) "Transmissometer measurement of POC". Their Figure 2 shows tight correlation of c vs POC below 500 nM. Deep-Sea Research I, 46 (1999) 353-369. Bishop (1999) found that the slope of correlations is very similar in almost all environments described – including the Atlantic – all samples were collected by in-situ filtration and compared to the same co-deployed 1-m pathlength transmissometer. Modern transmissometers accept more forward scattered light so the slope changed from 16 to 27. Never-the-less the correlations with beam c or beam cp remains robust in the upper kilometer. The same cannot be said of scattering (e.g. Bishop and Wood, 2008, Deep-Sea Research I 55, 1684–1706). See also Boss et al. 2015. Progress in Oceanography 133 (2015) 43–54. There will always be more assumptions for conversion of bbp to POC. I almost think that BIO-ARGO should consider bringing transmissometers back.

We thank the reviewer for calling our attention to these references. We agree that POC vs. cp is expected to be more linear than POC vs. bbp700. The nonlinearity of the POC vs. bbp700 relationship is clearly conveyed by our Fig. 2.

We propose to modify L440 as follows (new text underlined):

"The resulting range of 500–2000 mmol C m-2 is probably wide enough, as indicated by our indirect estimates based on the studies of Bishop (1999) and Bishop et al. (1999) (Appendix A)"

In addition, we will add to the Conclusions the recommendation to "incorporate transmissometers in the standard BGC-Argo float payload, which may enable more accurate POC quantification and provide valuable information on particle properties". In fact, transmissometers were already present in some floats deployed recently.

In the Appendix, we will amend lines 809-821 to clearly state that:

- Bishop 1999 found a consistent linear relationship between POC and cp in the top 1000 m.
- The correct conversion factor, taking into account the change in sensor design, is 27 mmol C m-2 rather than 16 mmol C m-2. This corresponds to ~324 mg C m-2.
- With this updated POC/cp estimate, the range of POC/bbp700 used in our sensitivity tests (500 to 2000 mmol m-2) would correspond to a backscattering ratio (bbp/cp) between 5% (seemingly too high) and 1.5% (closer to to most available estimates).
- Largest uncertainty in deep POC/bbp700 estimates at 1000 m results from insufficient knowledge of the bbp/cp at that depth, as most of the studies cited in our Appendix sampled shallower depths.

We will also modify Figure A1 to include the studies of Bishop 1999 and Bishop et al. (1999), and warn readers in the figure caption that there is no agreement on whether the POC vs. cp relationship can be considered constant in the open ocean or shows systematic variation with ecosystem structure.

L 616-617. On the other hand, it is unclear to what extent the bPOC inferred from the bbp700 spike signal is capturing mesozooplankton biomass... See. Comments above based on Bishop and Wood (2008; Deep-Sea Research I, 55, 1684–1706). The other point to make is that spikes seen even if they are all particles, don't adequately predict the complete particle spectrum of particles that dominate the sinking flux.

**Please see the response below.**

L 619-621. "Imaging devices mounted on BGC-Argo floats may provide more accurate quantification of bPOC, allowing for the separation of detrital bPOC (Trudnowska et al.,2021) from mesozooplankton and micronekton (Haëntjens et al., 2020)". As UVP methodology has not yet been proven on a float... an appropriate additional reference is Bourne et al., 2021 (Biogeosciences, 18, 3053–3086). These authors have documented an imaging carbon flux measuring float with the ability to separate and quantify particle classes contributing to flux. The instrument will be able to perform missions lasting seasons to years. It would be great to co-deploy this system with a UVP-modified float as the authors suggest.

We agree with the reviewer, and refer to our reply on L164. Following his suggestion, we propose to add this new text (underlined):

"Imaging devices mounted on BGC-Argo floats may provide more accurate quantification of LPOC, allowing for the separation of detrital LPOC (Trudnowska et al., 2021) from mesozooplankton and micronekton (Haëntjens et al., 2020) and the separation and quantification of particle classes contributing to flux across the complete particle spectrum (Bourne et al., 2021)."

In the Summary, There should be mention of international programs like GEOTRACES (Lam et al. 2011,2015) that are currently at sea in the global ocean and are doing an excellent job of quantifying the large and small particle abundances and chemistry in the ocean water column. The product from GEOTRACES is an asset to such synthesis efforts.

We will add a reference to GEOTRACES in L728:

"Further work is granted to investigate POC dynamics through the combination of PISCES2, autonomous observations and ship-based observation programs such as GEOTRACES (Lam et al., 2015)"

**Additional references cited in the responses**

Bishop, J. K. B., R. W. Collier, D. R. Kettens, and J. M. Edmond (1980), The chemistry, biology, and vertical flux of particulate matter from the upper 1500 m of the Panama Basin, Deep Sea Res., Part A, 27(8), 615–640, doi:10.1016/0198-0149(80)90077-1.

Druffel, E. R., Williams, P. M., Bauer, J. E., & Ertel, J. R. (1992). Cycling of dissolved and particulate organic matter in the open ocean. *Journal of Geophysical Research: Oceans*, *97*(C10), 15639-15659.

Duret, M. T., Lampitt, R. S., & Lam, P. (2019). Prokaryotic niche partitioning between suspended and sinking marine particles. *Environmental microbiology reports*, *11*(3), 386-400.

Fasham, M. J., Ducklow, H. W., & McKelvie, S. M. (1990). A nitrogen-based model of plankton dynamics in the oceanic mixed layer. Journal of Marine Research, 48(3), 591-639.

García-Martín, E. E., Davidson, K., Davis, C. E., Mahaffey, C., Mcneill, S., Purdie, D. A., & Robinson, C. (2021). Low contribution of the fast-sinking particle fraction to total plankton metabolism in a temperate shelf sea. *Global Biogeochemical Cycles*, *35*(9), e2021GB007015.

Giering, S. L. C., Cavan, E. L., Basedow, S. L., Briggs, N., Burd, A. B., Darroch, L. J., ... & Waite, A. M. (2020). Sinking organic particles in the ocean—flux estimates from in situ optical devices. *Frontiers in Marine Science*, *6*, 834.

Kriest, I., & Evans, G. T. (1999). Representing phytoplankton aggregates in biogeochemical models. *Deep Sea Research Part I: Oceanographic Research Papers*, *46*(11), 1841-1859.

Lebeaupin Brossier, C., Béranger, K., Deltel, C., & Drobinski, P. (2011). The Mediterranean response to different space-time resolution atmospheric forcings using perpetual mode sensitivity simulations. *Ocean Modelling*, *36*(1-2), 1-25.

Liu, H., Li, Q., Bai, Y., Yang, C., Wang, J., Zhou, Q., ... & Wu, G. (2021). Improving satellite retrieval of oceanic particulate organic carbon concentrations using machine learning methods. *Remote Sensing of Environment*, *256*, 112316.

Lombard, F., Boss, E., Waite, A. M., Vogt, M., Uitz, J., Stemmann, L., ... & Appeltans, W. (2019). Globally consistent quantitative observations of planktonic ecosystems. *Frontiers in Marine Science*, *6*, 196.

McKenzie, T., Twardowski, M., Briggs, N., Nayak, A. R., Boswell, K. M., & Dalgleish, F. (2020). Three-dimensional imaging lidar for characterizing particle fields and organisms in the mesopelagic zone. *Frontiers in Marine Science*, *7*, 1021.

Mestre, M., Ruiz-González, C., Logares, R., Duarte, C. M., Gasol, J. M., & Sala, M. M. (2018). Sinking particles promote vertical connectivity in the ocean microbiome. *Proceedings of the National Academy of Sciences*, *115*(29), E6799-E6807.

Mouw, C. B., Barnett, A., McKinley, G. A., Gloege, L., & Pilcher, D. (2016). Global ocean particulate organic carbon flux merged with satellite parameters. *Earth System Science Data*, 8(2), 531-541.

Mullin, M. M. (1965), Size fractionation of particulate organic carbon in the surface waters of the western Indian Ocean, Limnol. Oceanogr., 10(3), 459–462, doi:10.4319/lo.1965.10.3.0459.

Takeuchi, M., Doubell, M. J., Jackson, G. A., Yukawa, M., Sagara, Y., & Yamazaki, H. (2019). Turbulence mediates marine aggregate formation and destruction in the upper ocean. *Scientific reports*, *9*(1), 1-8.

Tonani, M., Pinardi, N., Dobricic, S., Pujol, I., & Fratianni, C. (2008). A high-resolution free-surface model of the Mediterranean Sea. *Ocean Science*, *4*(1), 1-14.

---

## Author Response (AR1)

**General response to reviewers**

We thank both reviewers (R1 and Jim Bishop) for their constructive criticisms and encouraging comments. Their recommendations have been useful to improve the article and make it a more valuable resource for both the modelling and observational communities.

Below we provide an overview of the main changes:

- We revised the abstract to make it more informative, precise and quantitative.
- We sharpened the writing of some sentences found to be cryptic by the reviewers, and provided more specific and quantitative information whenever possible.
- We added the references suggested by the reviewers, or cited them again in the place suggested by the reviewers if they were already cited elsewhere in the manuscript. In a few cases we declined to add the citation and provided a reasoning.
- We added a table to define acronyms, Table 1, in addition to in-text definitions.
- We changed the acronyms of small and big POC (sPOC and bPOC, respectively) to SPOC and LPOC.
- We added brief pieces of text to document or discuss more in depth some aspects highlighted by R1 (see detailed responses), related to particle remineralization, fragmentation, and sinking, and their representation in models, as well as the role of chemolithotrophic microbes. Note, however, that in-depth discussion of POC cycling processes in the mesopelagic is beyond the scope of this paper. These relevant questions are being addressed by our group in ongoing studies and will be the matter of future papers.
- We refined the information on POC estimation from bio-optics (beam attenuation and backscattering) in the main text and Appendix 1 based on the studies of Bishop and Wood (2008), Bishop (1999) and Boss et al. (2015).
- We improved the critical assessment of the correspondence between POC fractions simulated by PISCES and observed by backscattering sensors.
- We made the requested improvements to figures 4, 6, 7 and 8 (and their SM analogs).

In the following sections we provide our point-by-point responses, showing how we addressed each of the referees' concerns (with new text underlined). At the end, we listed the references cited here that were not in the original text. All referee comments are in blue, author responses are in black. Line numbers refer to the old version unless otherwise noted.

**Response to reviewer 1**

This study compares POC concentrations simulated by a biogeochemical model (PISCES) to high resolution field estimations from autonomous platforms and satellites. The authors show large discrepancies between models and observations. However, model and observations both agree on the behaviour of slow and fast sinking particles. The authors conclude that uncertainties in the POC conversion factors, imperfection between available observations and model points and in the representation of the physics in the model are sources of mismatch. While I feel like some of the processes invoked for explaining the discrepancies are not deeply discussed, I do not have any major issues with this work. I believe that some points should be clarified, better justified, and better documented before publication. These are listed below:

We thank R1 for his/her positive evaluation. In the new version, we will discuss or better document the aspects highlighted by R1, as described in detail below.

Minor remarks :

L24 : So all known factors driving sinking POC cycling in the mesopelagic basically…

Please see the response below.

L24-26: The abstract mentions that inconsistencies were identified. The rest of the sentence is somehow cryptic, and I feel like such inconsistencies should be detailed a bit more.

We have revised the abstract to address these comments. In the revised version we (i) present our novel approach and the results in a more logical order, (ii) provide quantitative results, and (iii) specify the major model-observations mismatch patterns, as well as the inconsistencies between BGC-Argo and satellite observations.

L46: None of these papers report actual measured flux attenuation, only (Marsay et al., 2015) does please revise.

Instead of these references, we will cite Mouw et al. 2(016), who presented the most up-to-date compilation of gravitational POC flux measurements in  the global ocean, to our knowledge.

L53: And wide variations in the attached microbial communities (Baumas et al., 2021)

We will modify the sentence as follows: "*Indeed, these particles also feature wide variations in shape, density, chemical composition and reactivity towards microbial degradation (Kharbush et al., 2020; Lam et al., 2015; Passow and Carlson, 2012; Stemmann and Boss, 2012), as well as wide variation in the attached microbial communities (Baumas et al., 2021; Duret et al., 2019; Mestre et al., 2018).*"

We added citations to the articles of Mestre et al. (2018) and Duret et al. (2019), which preceded Baumas et al. (2021) and give complementary views.

L55: Rephrase to "large" instead of big, here and throughout

Reworded throughout.

L56: This is not always true, please see work by (McDonnell  M., P. & Buesseler, 2010)

We thank the reviewer for calling our attention to this interesting reference, as well as Laurenceau-Cornec et al. (2019) (comment on L569). However, in this particular sentence of the Introduction we suggest to cite Cael et al. (2021), who provided the most comprehensive intercomparison to date, to our knowledge, of particle size vs. sinking speed relationships measured with a wide diversity of devices/methods and in various ocean settings. The study of McDonnell and Buesseler (2010) represents a particular region and season studied with a method (drifting sediment traps) that may select for/against certain particle types, as the authors acknowledge.

We modified the sentence as follows:

"Gravitational sinking speed generally increases with particle size, although observations show a wide scatter around, and deviations from, canonical Stokes' law predictions (Laurenceau-Cornec et al., 2019; Cael et al., 2021). Owing to this general relationship, particle populations are often partitioned into a few functional size classes...".

L62: Please include results by (Alonso-Gonzalez  J. et al., 2010; Riley et al., 2012).

We added these citations here. Please note, however, that Alonso-Gonález (2010) was already cited (eg, line 72, 682). Instead of Riley et al. (2012) we had cited Baker et al. (2017), because the latter study expanded Riley's one using a similar methodology (Marine Snow Catcher sampling).

L89: How was this represented in previous versions of the model?

In previous versions and up to PISCESv2 (Aumont et al., 2015), the default model configuration had two POC size classes, small and large, that sank at different speeds. Both small and large POC were remineralized with a fixed rate constant at a given temperature. An alternative configuration was also available with PISCESv2, which used the parameterization of Kriest and Evans (1999). Despite resolving the particles' size and sinking speed spectra, the latter parameterization was not sufficient

to reproduce observed POC concentration profiles, likely because slow-sinking and suspended particles were remineralized too rapidly through the entire water column.

See also the responses to reviewer 1 comments' on L236 and L239, as well as the response to reviewer 2, L222.

L92: which ones? from what methods? You may want to state the pitfalls of such methods to connect with the subsequent part of the text.

Rewritten as follows (new text underlined): "Despite this breakthrough in the representation of POC fractions in PISCESv2_RC (hereafter "PISCES"), the new parameterization was evaluated using only sparse measurements (Druffel et al., 1992; Lam et al., 2011; 2015) based on large-volume filtration with in-situ pumps. This approach enables accurate determination of the mass and composition of the particulate fraction but cannot afford high-frequency sampling over extended spatiotemporal scales (Bishop, 1999; Boss et al., 2015; Gardner et al., 2006)".

L95. Please state the size range of particles observed with such devices.

Done. Also, please refer to Table 1.

L103 : surface POC flux or concentration ?

Concentration. Fixed.

L106 : Which of the « drivers » are specifically being evaluated here?

The processes that control POC cycling ("drivers") are not specifically evaluated in this paper, so we remove the part of the sentence enclosed in commas in L106.

L236: what is this based on? justify

In most biogeochemical models, POC remineralization follows first-order reaction kinetics, with POC remineralization rate constants typically ranging between 0.02–0.06 $d^{-1}$ at 0 degrees (Laufkötter et al., 2016). This "k0" is then modulated as a power-law function of temperature in most models. In PISCES_v2, $k(T) = k0\cdot(1.064)^{T}$, resulting in a Q10 of 1.9. That is, remineralization rate constants for "fresh" POC (see response below, comment on L239) approximately double every 10 degrees, which is in reasonable agreement with experimental estimates (see for example the data compiled by Belcher et al., 2016; section 4.3; note the data are reported at in situ temperature).

The exact values used in each model result from the history of model development, usually based on 1D model configurations (see for example the classical model of Fasham et al., 1990), and the global-scale tuning applied in order to balance export production and remineralization while simulating realistic oxygen and nutrient fields.

L238: More strongly than what?

More strongly than that of slow-sinking particles. Fixed.

L239: (Briggs et al., 2020) recently found that fragmentation could be responsible for up to half what's observed in flux attenuation. How does the "small" fragmentation may compare to (Briggs et al., 2020) here.

We thank the reviewer for suggesting more specific wording here, because the expression "small fraction", which we used to avoid excessive detail, was in fact inappropriate. We propose to rephrase the last part of the paragraph as follows (new text underlined):

"Remineralization follows first-order kinetics with an initial specific rate "k" of 0.035 d-1 (at 0°C) for freshly produced detritus in the upper mixed layer. This "k" decreases with depth as an emergent result of the RC parameterization. To account for bacterial solubilization of aggregate-binding polymers, 10% of the degraded large detritus are routed to small detritus (this fraction is hard-coded based on previous calculations). The flux feeding rate depends on the particles' sinking flux, and thus attenuates the flux of large particles more strongly than that of small particles. A variable

fraction of the large detritus intercepted by flux feeders is not assimilated but fragmented into small detritus, attenuating up to around 50% of the large detritus sinking flux through the top 500 m during intense export events."

To provide further context: in PISCES, the fraction of the large POC vertical flux that is fragmented into small POC is not constant, and results from two main biological processes: bacterial GOC solubilization and mesozooplankton flux feeding on the "rain" of aggregates (large POC). Physical disaggregation of large POC into small POC is not considered.

Fragmentation by zooplankton increases with the proportion of mesozooplankton that are flux feeders and with the proportion of phytodetritus aggregates within the large POC. The latter two variables are computed internally in the model. The proportion of mesozooplankton flux feeders depends on the proportion of food available from sinking POC interception vs. that ingested through active suspension feeding. The proportion of large POC in the form of phytodetritus aggregates increases with diatom mortality upon nutrient limitation (the remaining large POC arising mostly from fecal pellets). In consequence, the role of fragmentation is expected to be most important during strong export events following the termination of diatom blooms.

Our estimates indicate that, in PISCES, mesozooplankton fragmentation removes around 50% of the sinking flux of large POC through the upper mesopelagic (100-500 m) in productive ecosystems (e.g., the subpolar North Atlantic). This figure is in good agreement with the findings of Briggs et al. (2020). In subtropical gyres, the relative flux attenuation due to fragmentation is smaller, partly because a larger proportion of the sinking flux is carried by small POC, which is not susceptible to fragmentation. Solubilization represents a more constant background process and becomes relatively more important at low latitudes. We refer the reviewer to the presentation we made at EGU 2020, available here

 https://presentations.copernicus.org/EGU2020/EGU2020-16602_presentation.pdf (slides 16–18).

L280: Any error associated with such climatologies (PISCES results excluded)?

Estimation of the uncertainty associated with these datasets is an arduous task, and measurement errors may easily be confounded with natural variability. Unfortunately, measurement uncertainties were not provided with these observational datasets on a grid cell basis. Work is ongoing to include space- and time-resolved uncertainty estimates in future versions of observational datasets (for example, ESA's ocean colour climate change initiative), which will enable more accurate calculation of model skill metrics taking into account those uncertainties. In the case of our POC climatology derived from BGC-Argo bbp700, our publicly available datasets include not only the means and the medians but the range and N of measurements in each grid cell, which can be used to some extent to gauge the uncertainty in climatological data. Recent assessments of the intercomparability between bio-optical measurements of POC and other quantities have been made by Lombard et al. (2019), Giering et al. (2020) and Liu et al. (2021).

We would like to note that, sometimes, the nominal uncertainties are smaller than the real uncertainties when sensors or retrieval algorithms are faced with a wider range of conditions. This seems to be the case, for example, with the Stramski et al. (2008) satellite POC algorithm. Therefore, we added the following text in section 4.1, L492 of the next version:

"This deviation is well beyond the nominal uncertainty of the satellite POC product (<30%) and the range of observed POC/bbp700 ratios at the sea surface (Fig. 2). Therefore, we conclude that the algorithm of Stramski et al. (2008) overestimates POC at high latitudes in winter, an issue that deserves further investigation".

L307: Please specify, what are the physical processes that are unrealistically represented in this region?

There are no specific processes that are not correctly represented per se in the model. The main limitation in that area (and in most coastal areas and semi-enclosed seas) is the resolution of the model. It is roughly equivalent to about 1° resolution which is very coarse. Several studies have shown that a proper representation of ocean dynamics in the med sea requires a resolution of at least 1/12° (Lebeaupin Brossier et al., 2011; Tonani et al., 2008) corresponding roughly to the first internal Rossby radius of deformation. As a consequence, our achieved resolution in that sea is insufficient to correctly simulate ocean dynamics there. Therefore, ocean biogeochemistry should be quite crudely simulated.

The text now reads (L330 of the new version): "PISCES TPOC was nearly fourfold higher than BGC-Argo TPOC at the surface in the Mediterranean, an overestimation that results from unrealistic physics in that basin caused by the too-coarse (2°) model grid (Tonani et al., 2008; Lebeaupin Brossier et al., 2011)".

L 369: what do mean by rare aggregates?

"Rare" is used here meaning that "aggregates are found at low concentrations". "Rare" is usually defined as "common or frequent; very unusual" (Cambridge), or "seldom occurring or found" (Merriam-Webster).

L450 See work by adam Maritny'group https://www.nature.com/articles/sdata201448 This presents of substantial amount of data including mesopelagic data that could be potentially used here.

Thanks for the reference. This dataset is part of the more recent compilation made by Ecvers-King et al. (2019), already cited in the manuscript..

L490: Add (Bianchi et al., 2018)

Done.

L530: does this translate into an increase in sinking speed (for a given size) in your model?

See response to comment on L670

L569: Size vs sinking velocities relationships depends mostly on their composition (Laurenceau-Cornec et al., 2019), some large unballasted aggregates will not sink as fast as Stokes predicts.

Please see the response on L56. We added the citation and mentioned uncertainty around the general assumption.

L585: What about chemoautolithotrophy (CO2 dark fixation by particles associated bacterias) (Herndl & Reinthaler, 2013)

We are well aware of the potential importance of chemoautolithotrophy, as shown in lines 491-492 and references cited. We will add the suggested reference and indicate that chemoautolithotrophy can alter the proportion of autotrophs/heterotrophs.

LL595-608: Silly question, why are BACT not explicitly represented in such models?

Most global biogeochemical models do not explicitly represent carbon oxidizing bacteria. Instead, they simulate degradation of organic matter according to rather simple and quite crude parameterizations based on a first order kinetics with a remineralization constant which may depend on temperature and some other factors (such as lability). We are not aware of a systematic justification of such an approach apart that historically it has been the first to be used. Some models do include free living carbon oxidizing bacteria which mainly consume labile DOM and hydrolyse semi-labile DOM. This increases quite significantly the complexity of the models which means that they are more expensive to run. Yet, to our knowledge, the skill of such detailed representation with respect to simpler parameterizations has not been documented in global scale ocean biogeochemical models. No models do include an explicit representation of particle attached bacteria, except to our knowledge the model of Anderson and Tang (2010). Probable reasons for

that are: the complex structure of marine particles and their colonization by bacteria, the specificity of the ecosystem in the particle microenvironment, exchanges between the particles and the surrounding sea water, ...

Please see Table 1.

L670: How are particles sinking velocities represented in PISCES? Do they increase with depth as observed in (Villa Alfageme et al., 2016)?

Depends on the model configuration. PISCES does not include an explicit mineral ballast mechanism, as this did not bring clear improvement over other parameterizations. However, it is possible to prescribe an increase in LPOC (bPOC) sinking speed between the base of the mixed layer and 5000 m (Aumont et al., 2015). Following Aumont et al. (2017), we prescribed a constant LPOC sinking speed of 50 m/d. Evaluation of LPOC sinking speeds is beyond the scope of this study. Yet, in the companion paper (https://gmd.copernicus.org/preprints/gmd-2021-222/) we did evaluate the feasibility of optimizing LPOC sinking speed in PISCES using BGC-Argo observations in the mesopelagic. The results indicate that observations in the mesopelagic layer are not sufficient to adjust LPOC sinking speed in PISCES, likely because a wider depth range (with wider variation in sinking speeds) is needed.

L676: Is fragmentation caused only by biological factors or could non-biological factors such as microturbulence play a role? Please discuss.

Amended as follows:

"This fragmentation may be caused by zooplankton feeding (Mayor et al., 2020; Stemmann et al., 2004a, 2004b; Stukel et al., 2019) and swimming (Goldthwait et al., 2004), bacterial hydrolysis of aggregate-binding polymers (Arnosti et al., 2012; Baltar et al., 2010a), and turbulence at high kinetic energy dissipation rate (Takeuchi et al., 2019)."

The study of Takeuchi et al. (2019) showed that turbulence results in new aggregate formation until a "critical turbulent kinetic energy dissipation rate of $10^{-6}$ (W kg$^{-1}$), above which the smallest turbulent eddies limit aggregate size".

We rephrased as follows:

"In many instances (Fig. 9), our analysis suggests that better model performance in the mesopelagic may be achieved by increasing the net transfer of LPOC to SPOC, e.g. through LPOC fragmentation, and the Teff of both fractions"

and removed the last part of the sentence:

", which may require further balancing the interplay between various mechanisms of POC export and flux attenuation."

We replaced this part of the sentence:

"which may arise from both suboptimal model parameters and model structure."

with the following one:

"Some limitations may be tackled by optimizing model parameters (e.g., particle sinking and remineralization rates), whereas others may require changes in model structure (i.e. equations), for example the representation of zooplankton feeding on, and transformation of, mesopelagic particles (Mayor et al., 2020).

Figure 6: This clearly show that at high latitudes PISCES produces a pool of small slow sinking particles that doesn't seem to be remineralised quickly enough as they do not appear on the sPOC BGC-ARGO profile. The occurrence of sPOC seen from the BGC-ARGO profile could well be from b-POC fragmentation into s-POC rather than from the original s-POC fraction sinking out on its own.

Thanks, we agree with R1, as shown by L701-704.

**Response to reviewer 2 (Jim Bishop)**

**Overview**. The authors compare PISCES model simulations and observed in-situ proxies for particulate organic carbon (POC) concentration in small (sPOC) and large size (bPOC) particle fractions in the 0-200 m epipelagic and 200m-1000m mesopelagic zone at several biomes in the world oceans. The observed in-situ sPOC and bPOC are derived from backscattering sensor data from biogeochemical ARGO floats (BGC-ARGO). bbp700 data are filtered to remove spikes – yielding a background profile representative of the small particle fraction (sbbp) and large particle spike anomaly (bbbp). The resultant sbbp data is scaled by a formula that includes a biome dependent multiplier and a depth decreasing ratio of POC/scattering with an assumed asymptotic value. The bPOC data are calculated from the anomaly contributed to by spikes and the same conversion factor is applied.

To my knowledge, this is the first successful attempt to reproduce subsurface particle concentrations in a model. This paper is very well written and logical in its development and summarizes areas where model and observations agree well and where there is disagreement. It highlights the value of optical sensors on floats. In the conclusions, there are concrete recommendations for added observations that will enhance insight into model parameterizations.

This is perhaps the most informative paper I have reviewed. The authors did a great job. The length of the following text reflects my excitement about the product. The recommendation of this reviewer is to publish after revision and a quick rereview. There is some more work with referencing. This should take little time.

There are some areas that could be clarified. ACRONYMS often appear without definition. Please add a table to ACRONYMS and clarify important ones in the text. Furthermore the representation of stocks (units mmol POC m-2) in a 200 m epipelagic layer and 800 m thick mesopelagic layer in the figures confuses the presentation of the paper. Please use mmol POC m-3. Another point of confusion is the use of bPOC to describe the large particle size fraction. Most of the particle literature uses "S" small and "L" large; it would be helpful to use this terminology. In the conclusions it might be important to mention that the international program GEOTRACES will produce comprehensive information on size fractionated particle chemistry (after Lam et al. 2011 – 2015), many of these data sets are growing in availability. Finally, as summarized, there are a number of float based optical systems that could be productively co-deployed within the BIO-ARGO framework that would illuminate particle flux processes. I think the flux float described by Bourne et al. 2021 (Biogeosciences, 18, 3053–3086, 2021) is a worthy addition to the list, such instruments were envisioned by Claustre et al. (2009) and will be ready. I also think the community should convene a review of the developing suite of sensors that could contribute to an evolved BIO-ARGO that answers key questions regarding modelling needs.

Reference to observations of Bishop and Wood (2009, in the southern ocean) using floats is relevant. They talked about varied criteria for MLD as well as transient stratification, and the concept of burst sampling. They deployed transmission and scattering sensors and had metrics for export and timing for exported material to reach 800m. (doi:10.1029/2008GB003206).

We thank Jim Bishop for his very encouraging comments, and appreciate his suggestions regarding relevant references that we had missed. In particular, we realized that Bishop (1999) and Bishop and Wood (2008) had addressed several questions that are highly relevant to our study, such as the

types of particles causing spikes in different optical sensors, or the different type of relationship between POC and cp (linear) or bbp700 (nonlinear) along vertical profiles.

We refer the reviewer to the bottom of the Detailed comments for the response about stock vs. concentration units in the figures.

**Detailed comments.**

L8: Not sure where the 4Pg C number comes from. Needs a reference. Most of the numbers in the text are in the range of 0.5-2. You could say is dwarfed by the 1000 Pg C in DIC pool (above 100 m). Or is this an estimate for the entire water column.

The 4 Pg C is the PISCES estimate for the entire water column (Table 2), which appears reasonable to us because PISCES output compares well to observations (Aumont et al., 2017; our study). We will specify this information to avoid confusion. In addition, we will consider mentioning in the Discussion (e.g. 4.1) that the global POC stock is dwarfed by the DIC pool, as suggested by the reviewer, while stressing that the turnover of POC is much faster than that of DIC.

L11-12: 80-90% Reference? Lam et al. 2011?

The 80-90% range approximately brackets the majority of observed and modelled estimates provided in our study (median 85%; L565), and is also central amongst other POC datasets obtained with different fractionation approaches. For example, marine snow catcher measurements (Riley et al., 2012; Baker et al., 2017; García-Martín et al., 2019) tend to give SPOC fractions >90% (L569-571). Fractions based on filtration with in situ pumps (MULVFS; Lam et al., 2011; Lam et al., 2015) and particle collection on filters (cutoff usually at 53 μm) give a wider SPOC/TPOC range of around 50% to virtually 100%, with most measurements clustering around 80% (see figure 5 in Aumont et al., 2017). Given that references are not allowed in the abstract unless strictly necessary, we propose to mention again the 80-90% range in the third paragraph of the Introduction (L55) and provide the same supporting references cited here.

L13: Define ACRONYM PISCES.

Pelagic Interactions Scheme for Carbon and Ecosystem Studies model. Defined at L8 in the first version of the manuscript.

L16: bPOC is not defined. Above… most literature refers to "L" POC (large).

We added the definition of bPOC and changed the acronym to LPOC. This and other definitions were added to a table of acronyms.

L51-54: Stemmann and Boss (2012), Cael and White, 2020; Stemmann et al., 2004a), particle populations are usually partitioned into a few.... Other references? Certainly this has been described in literature back to the 1970's.

We thank the reviewer for this comment. We added more references: Mullin (1965) and Bishop et al. (1980).

L70: Small POC can drive vertical POC fluxes across the mesopelagic layer. Needs better wording of "Small POC can drive…"  Small POC does not drive... but convective mixing or subduction of small POC can transport POC below the euphotic layer (or epi pelagic layer) into the mesopelagic layer, adding to POC export...

We thank the reviewer for this correction and rephrased following his suggestion:

"Convective mixing (Bishop et al., 1986; Dall'Olmo and Mork, 2014; Lacour et al., 2019) and subduction (Llort et al., 2018; Omand et al., 2015; Resplandy et al., 2019) can transport SPOC into the mesopelagic layer, adding to other export mechanisms and potentially making large contributions to total POC export (Alonso-Gonzalez et al., 2010; Henson et al., 2015)".

L 132: a fluorometer... optical backscattering. Name manufacturer. Use specific language.

We added the following information: "The selected floats were equipped with a Seabird-Wetlabs ECO-Triplet sensor package including a Chla fluorometer (excitation at 470 nm; emission at 695 nm), and a backscattering sensor at 700 nm."

not azL136: … the latter were flagged... need more information… Provide an example how data were flagged.

All BGC-Argo data undergo near real-time quality control at the data assembly center (DAC) level. This procedure includes a total of 30 tests on different sensors, such that each measurement is finally flagged with numbers ranging from 4 ("bad data") to 1 ("good data"). In the case of bbp700, the only postprocessing applied at the DAC level (besides application of calibration factors to raw digital counts) is the removal of negative spikes. Therefore, the entire bbp700 profiles typically get QC flags 2 or 3 depending on the DAC, and data that look perfectly good may routinely get a "3" until further delayed-mode processing can be applied. That's why we specified in L137 that "we used all bbp700 measurements with quality control flag ≤ 3 (equivalent results were obtained with flag ≤ 2)". Additionally, all bbp time series used in this study were visually checked which confirmed that no additional treatment or profile removal was required.

The case of bbp700 is different from that of chlorophyll a fluorescence, for example, which is further post-processed according to a protocol all DACs have agreed on, which includes application of a constant correction factor and further correction for non-photochemical quenching (NPQ), FDOM interference, bottom-of-profile signal, etc. (Schmechtig et al., 2018).

In summary, rather than providing too many details in the main text, we prefer to direct readers to the relevant documents already cited in the text (Schmechtig et al., 2016 and 2018).

L141: Define ACRONYMS

L149: NAOS, remOcean and other ACRONYMS should be tabulated.

Done.

L156: Reference needed… $\sigma\theta$ exceeded the surface reference value by 0.03 kg m-3. Bishop and Wood (2009) seems a good one.

We added citations to Bishop and Wood (2009) and Sallée et al. (2021) at the end of L156 to support our choice, followed by this sentence: "The 0.03 kg m$^{-3}$ criterion provided sensible results across biomes and was consistent with the NEMO-simulated turbocline depth."

Note that in L215 we had already given some justification for using this MLD criterion:

"Although MIMOC is based on an algorithm that evaluates several MLD criteria, it has been shown to be globally consistent with the MLD based on a 0.03 kg m-3 $\sigma\theta$ threshold (Holte and Talley, 2009; Sallée et al., 2021). For the float time series, we used the MLD defined by a 0.03 kg m-3 threshold computed for each individual profile."

As done by Bishop and Wood (2009), we tested several MLD criteria (explained in L157; Fig. S1), which can capture vertical mixing on different time scales, and with different skill depending on the strength of thermal and haline stratification in a given region/season.

L 164: "The baseline and spike signals were converted to sPOC and bPOC...".  Are there observations that confirm this correspondence? Please calculate the volume of water seen by the scattering sensor and the likelihood of the sensor seeing large particles. The only study I am aware of is Bishop and Wood (2008 Deep-Sea Research I55, 1684–1706) who concluded: "Those [referring to spikes] of the scattering sensor mostly appear to reflect abundances of larger and chain forming phytoplankton, protist grazers and possibly large aggregates. Nearly transparent particles like marine snow are not readily detected in transmitted light but are easily detected using reflected light. "

We added the following specification in L162:

"Each spike reflects the passage of a particle (organism or aggregate) larger than about 100 μm in front of the sensor window. Previous studies inferred that backscattering spikes arise mostly from phytodetritus aggregates, but also from large zooplankton and phytoplankton (Bishop and Wood, 2008; Briggs et al., 2011; Gardner et al., 2000). Assuming that backscattering sensors sample a volume of 10 mL (Briggs et al., 2020), we estimated that backscattering spikes concentration was typically between a few and <100 $L^{-1}$, consistent with other independent estimates of large particle concentration in the open ocean (McDonnell and Buesseler, 2010; Stemmann et al., 2008; Stemmann and Boss, 2012). Backscattering spikes were on average 4–10 times more abundant than chlorophyll fluorescence spikes. The $b_{bp700}$ spikes larger than 0.008 $m^{-1}$, associated with particles larger than around 2 mm (very large aggregates, vertically migrating zooplankton) were removed, with a negligible impact on the total spike signal (Briggs et al., 2020)".

To justify this added text, we provide an extended argumentation herewith. The processing of raw backscattering profiles to separate the baseline and spike signals was addressed in detail by Briggs et al. (2020) (see the Supplementary Materials). The processing applied in our study followed in most aspects their approach (L150-165), and was restricted to the same subset of floats sampling at high vertical resolution (L150).

As described by Briggs and coauthors, the signal processing applied "removes positive spikes due to large individual particles that are rare relative to the sample volume of ~10 ml (taking into account ~10 cm of movement during the sensors' 1 s integration time)". They also highlighted that "high vertical resolution (~1 m or higher) measurements are required in order to obtain useful, depth-resolved concentrations from a single profile".

Particles larger than 100 μm, the approximate minimum size of particles causing spikes in our approach, are usually found at low concentrations in the top 1000 m of the ocean, typically between a few and <100 particles/L (e.g., see total particle counts or particle size spectra in Stemmann and Boss, 2012, figure 2; Stemmann et al., 2008, figure 5 and 6; McDonnell & Buesseler, 2010, figure 3; McKenzie et al., 2020). Smaller particles occur at much greater abundance owing to power-law (log-log) slopes of the particle size distribution typically ranging between -4 and -3 . Therefore, the likelihood of detecting aggregates when sampling a volume of 10 mL is low, considering that the concentration of large particles is usually <1 particle/mL. Given that we are binning the data into depth intervals with no less than discrete 10 measurements per bin (and around 20 on average), the minimum water volume sampled in our approach is of around 100 mL per bin (but usually 200 mL). Theoretically, this should allow for the detection of aggregates in the 10/L concentration range.

Fortunately, our processing pipeline included the calculation of the mean spike frequency in each depth bin. Dividing it by the mean sample volume (10 mL = 0.01 L) gives the mean spike concentration. The table below summarizes the mean spike concentration (#/L) for four representative floats that sampled over >2 years, identified by their WMO number and biome, for depths >50 m (this is to avoid the smoothing effect of chlorophyll fluorescence correction on the profile, which effectively removes chlorophyll spikes; see also Briggs et al., 2011). The absolute range of bbp700 spikes concentration in these floats was 7–108 spikes/L.

| Variable | 6901486 (NASPG) | 6901579 (Subantarctic) | 6901472 (SPG) | 6901491 (Mediterranean) |
|---|---|---|---|---|
| bbp700 spikes, # $L^{-1}$ | 59±9 | 63±11 | 55±8 | 61±7 |
| chl spikes, # $L^{-1}$ | 15±19 | 16±24 | 8±12 | 7±13 |

This analysis indicates that:

- bbp700 spikes are more abundant and less variable than chl spikes, which can be expected given that chl spikes only detect large living cells or aggregates, whereas bbp700 can also detect living zooplankton and their fecal pellets. Differences in the vertical-temporal distribution of bbp700 and fluorescence spikes were already noted by Briggs et al., 2011.
- chl spikes are relatively more abundant in the more productive subpolar biomes, consistent with the notion of a larger proportion of the export production channelled by phytodetritus aggregates in productive biomes.
- inclusion of deep chl maxima (and/or bbp700 maxima), located well-below 50 m in the SPG and Mediterranean biomes, may explain the relatively high abundance of large particles in those biomes compared to subpolar ones (as found at ALOHA station by Bishop and Wood, 2008) despite their lower productivity.

Visual examination indicates that vertically binned bbp700 spike profiles are noisy but usually have defined shapes, i.e., aggregates do not occur randomly through the water column, especially in productive biomes (Fig. 6). Such features allow for the detection of export events from temporal sequences of profiles (Briggs et al. 2011, 2020). Indeed, further binning may be needed to reduce noise and obtain more accurate quantitative estimates of large POC concentration. This is precisely what we attempted by integrating the data between the 0-200 m (epipelagic) and 200-1000 m (mesopelagic) depth horizons, obtaining high correlation with PISCES-modelled LPOC (Fig. 8). Taking into account the vertical resolution of the profiles (L152),  total epi- and mesopelagic sample volumes are around 3L and 2L, respectively.

Finally, we agree with the reviewer that the nature of the particles causing spike events cannot be deduced solely from their backscattering at 700 nm. Thus, we also explored whether additional information could be obtained by computing the fraction of bbp700 spikes that co-occurred with chlorophyll fluorescence spikes. Our analysis indicated that bbp700 spikes co-occurring with fluorescence spikes accounted for between 15% (subtropical gyres) and 45% (subpolar biomes) of the large POC mass. This estimate is only a lower bound because the backscattering and chlorophyll fluorescence sensors do not sense exactly the same volume of water. In fact, most bbp770 spikes did not co-occur with fluorescence ones, meaning that co-occurring spikes were larger than the average spikes. It is logical to hypothesize that co-occurring bbp700 and fluorescence spikes are more likely when aggregates are larger. In summary, we estimate that a sizable fraction of the LPOC, of at least 15-45% depending on the biomes and the seasons,  may have corresponded to phytodetritus aggregates in our study.

Although these calculations are admittedly rough, our conclusions are fully compatible with the inferences made by Bishop and Wood (2008). Note, however, that the Seapoint turbidity sensor used by Bishop and Wood (2008) was different to the Seabird-Wetlabs ECO-Triplet sensors mounted on BGC-Argo floats. In particular, the Seapoint sensor measured scattering at a broad angle between 45-135° (880 nm wavelength), whereas ECO-Triplet sensors measure bbp (700 nm wavelength) at 124°, with full width at half maximum of 20° to 30°. Cetinic et al. (2012) showed that these measurements are not equivalent (see their Appendix, figure A1).

Line 181. Reference to Fig. 2. Should not the green dotted line in the deep mixed layer reflect the average of POC/bbp over the interval, not the value at its deepest limit? Please explain more clearly why or why not.

The algorithm we used to estimate POC/bbp700 is correctly represented in Fig. 2. The modulation of POC/bbp700 was already critically assessed in L623-630 of the Discussion, but we admit that the effect of diluting mixed-layer waters with waters from below would be more correctly represented by taking the average across the entire depth interval homogenized by mixing. However, this solution does not come without problems. We rewrote L570 of the new version (section 4.2) as follows to incorporate this criticism, providing also a justification of our approach:

"Modulation of POC/bbp700 by vertical mixing improves the agreement between PISCES and BGC-Argo data in regions with wide seasonal MLD amplitude such as the NASPG. Still, our approach should be seen as a simplistic first-order approximation, and alternative formulations should be further evaluated when more data become available. For example, POC/bbp700 might be kept constant regardless of the MLD, or estimated for each profile as the average within the mixed layer of the pre-computed POC/bbp700 profile (eq. 1). In both cases, however, POC/bbp700 would have to decrease abruptly below the mixed layer to meet the low POC/bbp700 in the mesopelagic, producing sharp discontinuities that have not been observed."

Unfortunately, we are not aware of studies that measured the change of POC/bbp700 during convective mixing.

L 199. Equation 2.  zref = max ("X","Y"). Reads like computer code . Try to clarify in the text what is meant.

Fixed.

L 222. Please explain briefly terms like "RC parameterization".

We propose to expand a little bit the description of the RC parameterization in the Introduction, rather than here. We added the following new text:

"Aumont's work also showed that the model's fit to observed deep-ocean POC concentrations could be dramatically improved by treating detrital POC, both small and large, as a mixture of particles with different reactivity (or lability) towards bacterial degradation. In this scheme, termed the reactivity continuum (RC) parameterization, detrital POC degradation is computing after dividing it into many reactivity classes that approximately follow a continuous gamma distribution –hence its name. The most labile fractions are rapidly consumed below the upper mixed layer, such that vertically exported POC becomes progressively more refractory. This results in enhanced preservation of SPOC in the model and a much more realistic fraction of SPOC with respect to total POC (TPOC) in the ocean interior. In addition, the RC scheme does not appear to harm model estimates of the gravitational POC flux."

Please define all ACRONYMS somewhere.

Done (Table 1).

L310. Fig. 4. Add log tick marks. Rather than regular grid on x axis. Otherwise… nice figure.

Thanks. Fixed.

L349. Fig. 6 (and corresponding similar figures). When printed. The various blue/cyan lines in panel (a) and (d) are completely lost.

Done, changed to black lines with different patterns.

L440-441. …"Bishop et al. (1999) suggest a plausible range of 815–1630 mmol C m-2 (Appendix A)".

In the Appendix, the authors overlooked Bishop (1999) "Transmissometer measurement of POC". Their Figure 2 shows tight correlation of c vs POC below 500 nM.  Deep-Sea Research I, 46 (1999) 353-369.  Bishop (1999) found that the slope of correlations is very similar in almost all environments described – including the Atlantic – all samples were collected by in-situ filtration and compared to the same co-deployed 1-m pathlength transmissometer.  Modern transmissometers accept more forward scattered light so the slope changed from 16 to 27. Never-the-less the correlations with beam c or beam cp remains robust in the upper kilometer. The same cannot be said of scattering (e.g. Bishop and Wood, 2008, Deep-Sea Research I 55, 1684–1706). See also Boss et al. 2015. Progress in Oceanography 133 (2015) 43–54. There will always be more assumptions for conversion of bbp to POC. I almost think that BIO-ARGO should consider bringing transmissometers back.

We thank the reviewer for calling our attention to these references. We agree that POC vs. cp is expected to be more linear than POC vs. bbp700. The nonlinearity of the POC vs. bbp700 relationship is clearly conveyed by our Fig. 2.

We modified L440 as follows (new text underlined):

"The resulting range of 500–2000 mmol C m$^{-2}$ is probably wide enough, as indicated by our indirect estimates based on the studies of Bishop (1999) and Bishop et al. (1999) (Appendix A)"

In addition, we added to the Conclusions: "Addition of transmissometers may enable more accurate POC quantification (Bishop, 1999) and particle characterization (Boss et al., 2015)". In fact, transmissometers were already present in some floats deployed recently.

In the Appendix, we amended L839-846 as follows:

"The number of studies that tackled the interconversion between POC and bio-optical proxies in the mesopelagic layer is much smaller than those that focused on the epipelagic layer. Besides Cetinić et al. (2012) in the subpolar North Atlantic, we are only aware of the studies of Bishop (1999) and Bishop et al. (1999). The latter two studies found a POC vs. cp slope of around 16 mmol C m-2 (= mmol C m-3 / m-1). Given that modern transmissometers accept more forward scattered light that those used by Bishop (1999), the corresponding slope would be approximately 27 mmol C m-2 (J. Bishop, personal communication). Bishop (1999) found this slope to be nearly constant in contrasting areas over the 0–1000 m depth range, as depicted in Fig. A1 with the datasets labelled "Bishop1999eqpac" (Equatorial Pacific) and "Bishop1999all" (Equatorial Pacific, NE Pacific and North Atlantic together)."

We also modified Figure A1 to include the studies of Bishop 1999 and Bishop et al. (1999).

L 616-617. On the other hand, it is unclear to what extent the bPOC inferred from the bbp700 spike signal is capturing mesozooplankton biomass… See. Comments above based on Bishop and Wood (2008; Deep-Sea Research I, 55, 1684–1706). The other point to make is that spikes seen even if they are all particles, don't adequately predict the complete particle spectrum of particles that dominate the sinking flux.

Please see the response below.

L 619-621. "Imaging devices mounted on BGC-Argo floats may provide more accurate quantification of bPOC, allowing for the separation of detrital bPOC (Trudnowska et al.,2021) from mesozooplankton and micronekton (Haëntjens et al., 2020)". As UVP methodology has not yet been proven on a float… an appropriate additional reference is Bourne et al., 2021 (Biogeosciences, 18, 3053–3086). These authors have documented an imaging carbon flux measuring float with the ability to separate and quantify particle classes contributing to flux. The instrument will be able to perform missions lasting seasons to years. It would be great to co-deploy this system with a UVP-modified float as the authors suggest.

We agree with the reviewer, and refer to our reply on L164. Following his suggestion, we added:

"Imaging devices mounted on BGC-Argo floats may provide more accurate quantification of LPOC, allowing for the separation of detrital LPOC (Trudnowska et al., 2021) from mesozooplankton and micronekton (Haëntjens et al., 2020) and the separation and quantification of particle classes contributing to flux across the complete particle spectrum (Bourne et al., 2021)."

In the Summary, There should be mention of international programs like GEOTRACES (Lam et al. 2011,2015) that are currently at sea in the global ocean and are doing an excellent job of quantifying the large and small particle abundances and chemistry in the ocean water column. The product from GEOTRACES is an asset to such synthesis efforts.

We will add a reference to GEOTRACES in L728:

"Further work is granted to investigate POC dynamics through the combination of PISCES2, autonomous observations and ship-based observation programs (e.g., GEOTRACES, Lam et al. (2015)) and data compilations (Mouw et al. ,2016; Evers-King et al., 2017)."

Reviewer 2: "the representation of stocks (units mmol POC m-2) in a 200 m epipelagic layer and 800 m thick mesopelagic layer in the figures confuses the presentation of the paper. Please use mmol POC m-3."

This comment affects mostly figure 8 (plus S11–S14 of the SM, with similar design) and the bottom panels of figures 6 and 7 (plus figures S5–S10 and S15–S16). We thank the reviewer for calling our attention to this aspect, but we disagree that displaying both stocks and concentrations can confuse the reader. We prefer to keep displaying POC stocks for the following reasons:

- Considering that previous estimates of the oceanic POC stocks have largely focused on the epipelagic layer (see section 4.1), we find it inherently interesting to show that the mesopelagic layer holds a similar amount of POC.
- The distinction between both types of variable and their respective units is always clear in the figures (axes, labels, captions) and in the main text.
- The magnitude of POC concentrations from the surface to 1000 m depth can be easily appreciated in figure 4 (which has also been improved), in the top and middle panels of figures 6 and 7 (plus figures S5–S10 and S15–S16), and in Table 3. Steep vertical gradients of POC concentration are usually found within each layer, such that "mean" concentrations are rarely representative of the entire layer. Therefore, vertically averaged concentrations do not add relevant new information, whereas vertically integrated stocks add a new layer of information.
- In Figure 8 and its analogous SM figures, the depth range (0–200 or 200–1000 m) used to compute the vertical integrals is indicated in the panel labels. The conversion from standing stocks (mmol C m$^{-2}$) to average vertical concentrations (mmol C m$^{-3}$) can be easily accomplished by dividing standing stocks by the thickness of each layer (200 or 800 m).
- Finally, in Fig. 8 and figures with an analogous design, the use of standing stocks facilitates the display of data points with the same scale in the left and right panels, halving the amount of axis scales that have to be read.

To address the reviewer's criticism, we made the following changes:

1. In section 3.3, second and third paragraphs, we added the ranges of SPOC and LPOC concentrations after the ranges of the stocks.

2. Although the figures are self explanatory, we added further information in the caption of figure 8:

"Figure 8. Mean annual POC stocks. BGC-Argo versus PISCES scatterplots are shown for the standing stocks (vertical integrals) of small POC (SPOC) and large POC (LPOC) in the epipelagic (0–200 m) and mesopelagic (200–1000 m) layers…".

**Additional references cited in the responses**

Bishop, J. K. B., R. W. Collier, D. R. Kettens, and J. M. Edmond (1980), The chemistry, biology, and vertical flux of particulate matter from the upper 1500 m of the Panama Basin, Deep Sea Res., Part A, 27(8), 615–640, doi:10.1016/0198-0149(80)90077-1.

Druffel, E. R., Williams, P. M., Bauer, J. E., & Ertel, J. R. (1992). Cycling of dissolved and particulate organic matter in the open ocean. *Journal of Geophysical Research: Oceans*, *97*(C10), 15639-15659.

Duret, M. T., Lampitt, R. S., & Lam, P. (2019). Prokaryotic niche partitioning between suspended and sinking marine particles. *Environmental microbiology reports*, *11*(3), 386-400.

Fasham, M. J., Ducklow, H. W., & McKelvie, S. M. (1990). A nitrogen-based model of plankton dynamics in the oceanic mixed layer. Journal of Marine Research, 48(3), 591-639.

García-Martín, E. E., Davidson, K., Davis, C. E., Mahaffey, C., Mcneill, S., Purdie, D. A., & Robinson, C. (2021). Low contribution of the fast-sinking particle fraction to total plankton metabolism in a temperate shelf sea. *Global Biogeochemical Cycles*, *35*(9), e2021GB007015.

Giering, S. L. C., Cavan, E. L., Basedow, S. L., Briggs, N., Burd, A. B., Darroch, L. J., ... & Waite, A. M. (2020). Sinking organic particles in the ocean—flux estimates from in situ optical devices. *Frontiers in Marine Science*, *6*, 834.

Kriest, I., & Evans, G. T. (1999). Representing phytoplankton aggregates in biogeochemical models. *Deep Sea Research Part I: Oceanographic Research Papers*, *46*(11), 1841-1859.

Lebeaupin Brossier, C., Béranger, K., Deltel, C., & Drobinski, P. (2011). The Mediterranean response to different space–time resolution atmospheric forcings using perpetual mode sensitivity simulations. *Ocean Modelling*, *36*(1-2), 1-25.

Liu, H., Li, Q., Bai, Y., Yang, C., Wang, J., Zhou, Q., ... & Wu, G. (2021). Improving satellite retrieval of oceanic particulate organic carbon concentrations using machine learning methods. *Remote Sensing of Environment*, *256*, 112316.

Lombard, F., Boss, E., Waite, A. M., Vogt, M., Uitz, J., Stemmann, L., ... & Appeltans, W. (2019). Globally consistent quantitative observations of planktonic ecosystems. *Frontiers in Marine Science*, *6*, 196.

McKenzie, T., Twardowski, M., Briggs, N., Nayak, A. R., Boswell, K. M., & Dalgleish, F. (2020). Three-dimensional imaging lidar for characterizing particle fields and organisms in the mesopelagic zone. *Frontiers in Marine Science*, *7*, 1021.

Mestre, M., Ruiz-González, C., Logares, R., Duarte, C. M., Gasol, J. M., & Sala, M. M. (2018). Sinking particles promote vertical connectivity in the ocean microbiome. *Proceedings of the National Academy of Sciences*, *115*(29), E6799-E6807.

Mouw, C. B., Barnett, A., McKinley, G. A., Gloege, L., & Pilcher, D. (2016). Global ocean particulate organic carbon flux merged with satellite parameters. *Earth System Science Data*, *8*(2), 531-541.

Mullin, M. M. (1965), Size fractionation of particulate organic carbon in the surface waters of the western Indian Ocean, Limnol. Oceanogr., 10(3), 459–462, doi:10.4319/lo.1965.10.3.0459.

Takeuchi, M., Doubell, M. J., Jackson, G. A., Yukawa, M., Sagara, Y., & Yamazaki, H. (2019). Turbulence mediates marine aggregate formation and destruction in the upper ocean. *Scientific reports*, *9*(1), 1-8.

Tonani, M., Pinardi, N., Dobricic, S., Pujol, I., & Fratianni, C. (2008). A high-resolution free-surface model of the Mediterranean Sea. *Ocean Science*, *4*(1), 1-14.

---

## Author Response (AR2)

Dear Editor,

Please find enclosed the manuscript files with the corrections you suggested. To make the figures more readable by color-blind people, we modified:

- The continuous color scale in panels a, b, d and e of figures 6 and 7 (plus similar SM figures), replacing the "Spectral" palette by the "YlGnBu" palette (from the R package Color Brewer).
- The discrete color scale in figures 1, 2, 5 and 8–10 (plus SM figures), used to distinguish ocean biomes. In this case, we have chosen two shades of blue plus red and golden tones.

We followed the recommendations exposed here and checked all the figures using Color Blindness Simulator. We are thankful for your suggestion, which clearly improved the figures, and made them more harmonious in the case of fig. 6 and 7.

We also modified Fig. 3 and edited one sentence (L315) to better describe the comparison between POC estimated from BGC-Argo and satellite data. The underlying calculations and datasets, as well as the message of the paper, remain unchanged. These slight modifications are described below:

**Figure 3**: We added in the caption: "*Satellite data not shown for months when more than half of the ocean pixels could not be observed because of low solar elevation at high latitudes.*"

It is well known that ocean color remote sensing is limited by incident sunlight at low solar elevation (discussed in 4.1). In consequence, satellites cannot "see" the full domain sampled by BGC-Argo floats or simulated by the model during some months at high latitudes. For this reason, and to avoid misleading visual comparison, in the previous version of Fig. 3 we had removed the satellite POC data points for the months 1 and 12 in the NASPG region, which were deemed less representative of the full domain. Here we applied a more stringent criterion and additionally removed satellite POC data points for month 11 in the NASPG and months 1 and 12 in the Subantarctic biomes, as explained in the caption. Note that satellites see 100% of the domain in the Mediterranean and STG biomes in all months, and 95–100% of the domain during months 3–9 and 2–10, respectively, in the NASPG and Subantarctic biomes. Finally, note that this issue affects only the display of data. Our quantitative comparisons between observed and modeled datasets are based, in all cases, on fully-coincident domains, i.e. on the same number of grid cells (see next point).

**L315**: The sentence

"*Satellite TPOC was in poor agreement with both PISCES and BGC-Argo TPOC outside the apex of the bloom, exceeding BGC-Argo TPOC by up to seven-fold (fourfold) in the NASPG (Subantarctic), as discussed in section 4.1.*"

was replaced by

"*Satellite TPOC was in poor agreement with both PISCES and BGC-Argo TPOC outside the apex of the bloom. During the winter semester (months 10-12 and 1-3), and considering only the subset of pixels observed by both satellites and floats, satellite TPOC exceeded BGC-Argo TPOC by a factor of 6.1 (3.3) in the NASPG (Subantarctic), as discussed in section 4.1.*"

Thus, the text now provides more precise quantities, representative of all the pixels concurrently observed by satellites and BGC-Argo over a longer period. The new text better supports our claim that available satellite products strongly overestimate sea-surface POC under certain conditions.

We thank you for your positive evaluation of the paper, and hope that this new version will fulfill all the requirements for publication in *Biogeosciences*.

Martí Galí

on behalf of all coauthors